# Towards a Unified Framework for Uncertainty-aware Nonlinear Variable Selection with Theoretical Guarantees

**Wenying Deng**
Harvard University
wdeng@g.harvard.edu

**Beau Coker**
Harvard University
beaucoker@g.harvard.edu

**Rajarshi Mukherjee**
Harvard University
ram521@mail.harvard.edu

**Jeremiah Zhe Liu**[*]
Harvard University & Google Research
jereliu@google.com

**Brent A. Coull**[*]
Harvard University
bcoull@hsph.harvard.edu

## Abstract

We develop a simple and unified framework for nonlinear variable importance estimation that incorporates uncertainty in the prediction function and is compatible with a wide range of machine learning models (e.g., tree ensembles, kernel methods, neural networks, etc). In particular, for a learned nonlinear model $f(\mathbf{x})$, we consider quantifying the importance of an input variable $\mathbf{x}^j$ using the integrated partial derivative $\Psi_j = \|\frac{\partial}{\partial \mathbf{x}^j} f(\mathbf{x})\|^2_{P_{\mathcal{X}}}$. We then (1) provide a principled approach for quantifying uncertainty in variable importance by deriving its posterior distribution, and (2) show that the approach is generalizable even to non-differentiable models such as tree ensembles. Rigorous Bayesian nonparametric theorems are derived to guarantee the posterior consistency and asymptotic uncertainty of the proposed approach. Extensive simulations and experiments on healthcare benchmark datasets confirm that the proposed algorithm outperforms existing classical and recent variable selection methods.

## 1 Introduction

Variable selection is often of fundamental interest in many data science applications, providing benefits in prediction error, interpretability, and computation by excluding unnecessary variables. As datasets grow in complexity and size, it is crucial that variable importance estimation methods can account for complex dependencies among variables while remaining computationally feasible. Furthermore, as the number of approaches to model such datasets has increased, it is crucial that the importance of each variable can be compared across model classes and extended to new ones as they are developed.

While there are established approaches for quantifying variable importance in linear models (e.g., LASSO regression Hastie et al. [2015]), there is little consensus as to the preferred methodology or theory for variable importance in nonlinear models. Generalized additive models Hastie and Tibshirani [1990] use similar methods as their linear counterparts Wang et al. [2014], but the additivity assumption for nonlinear functions of the variables is too restrictive in many applications. Random Forests (RF) Breiman [2001] measure variable importance using an impurity measure, which is based on the average reduction of the loss function were a given variable removed from the model. Friedman [2001] extended this method to boosting, where the definition of variable importance is generalized by considering the average over all of the decision trees. Deep neural networks (DNNs) are widely-used for many artificial intelligence applications, and a substantial effort has been invested

---

[*]Co-senior author. Work done at Harvard University.

36th Conference on Neural Information Processing Systems (NeurIPS 2022).

into developing DNNs with variable selection capabilities. Typically, this class of models involves manipulating the input layer, for example by imposing an $L_1$ penalty Castellano and Fanelli [2000], Feng and Simon [2019], using backward selection Castellano and Fanelli [2000], or knockoffs Lu et al. [2018]. Unfortunately, each model class based on DNNs requires a tailored procedure, which limits comparability across different model formulations.

Bayesian variable selection methods provide principled uncertainty quantification in variable importance estimates as well as a complete characterization of their dependency structure. These methods allow the variable importance estimation procedure to tailor its decision rule with respect to the correlation structure Liu [2021]. Yet, as in frequentist models, each method has a different definition of a variable's importance. For example, in Bayesian additive regression trees (BART), a variable's importance can be measured by the proportion of trees that use it Chipman et al. [2010], while in Gaussian process (GP) models, a variable's importance can be measured by the frequency of the fluctuations of the estimated outcome-predictor function (e.g., the length-scale parameter as controlled by the automatic relevance determination) in the direction of the variable Neal [1996], Wipf and Nagarajan [2007]. Recently, a closely-related line of work uses the norm of the kernel gradient to quantify variable importance under classical GP models [He et al., 2021] or deep Bayesian neural networks [Liu, 2021]. However, these work either do not incorporate uncertainty, or are restricted to a particular model class (see Appendix J). Furthermore, the traditional Bayesian modeling procedures tend to be computationally burdensome, making them less feasible for large-scale applications [Andrieu et al., 2003].

Our work starts with the observation that many machine learning models can be written as kernel methods by constructing a corresponding feature map. For example, random forests can be written as kernel methods by partitions Davies and Ghahramani [2014], and deep neural networks can be written as kernel methods by using the last hidden layer as the feature map Snoek et al. [2015], Hinton and Salakhutdinov [2007], Calandra et al. [2016]. Each of these feature maps can be constructed before Bayesian learning of the GP (e.g., by pre-training on the same or a separate dataset), providing additional modeling expressiveness and representational capacity. Then, the GP learning is equivalent to performing Bayesian inference with respect to the (linear) weighting parameters of the feature-map basis functions and the posterior inference proceeds analogously to that of a Bayesian linear regression (see Section 2.1 for details). The ability of a GP model to incorporate these adaptive feature maps becomes especially important in high-dimensional applications, where effective dimension reduction is necessary to circumvent the curse of dimensionality and ensure good finite-sample performance [Bach, 2016].

**Contributions**. We propose a unified variable importance estimation framework that is compatible with a wide range of machine learning models and can be defined by, or be closely approximated by, a differentiable feature map. Notable members include neural networks and random forests (Appendix B). Our approach defines variable importance as the norm of the function's partial derivative, as was previously studied in the context of frequentist nonparametric regression Rosasco et al. [2013]. We extend it to a much wider class of models than previously considered (Section 2), propose a principled Bayesian approach to quantify the variable importance uncertainty in finite data (Section 3.1), and derive rigorous Bayesian nonparametric theorems to guarantee the method's consistency and asymptotic optimality (Section 3.2). To incorporate powerful non-differentiable models into our framework, we also show how to apply this approach to partition-based methods (e.g., decision trees) by leveraging their (soft) feature representation (Appendix F.1). This leads to the first derivative-based Bayesian variable importance estimation approach for tree-type models that is both theoretically grounded and empirically powerful. This method strongly outperforms other variable importance estimation approaches tailor-designed for random forests (e.g., impurity or random-forest knockoff [Breiman et al., 1984, Candes et al., 2017]). We conduct extensive empirical validation of our approach and compare its performance to that of many existing methods across a wide range of data generation scenarios. The results show a clear advantage of the proposed approach, especially in complex scenarios or when the input is a mixture of discrete and continuous features (Section 4).

## 2 Preliminaries

**Problem Setup**. We consider the classical nonparametric regression setting with $d$-dimensional features $\mathbf{x} = (\mathbf{x}^1, \ldots, \mathbf{x}^d) \in \mathcal{X} = \mathbb{R}^d$ and a continuous response $y \in \mathbb{R}$. The features $\mathbf{x}$ are allowed to have a flexible nonlinear effect on $y$, such that:

$$y = f_0(\mathbf{x}) + e_i, \quad \text{where } e_i \overset{i.i.d.}{\sim} \mathcal{N}(0, \sigma^2), \tag{1}$$

with homoscedastic noise level $\sigma^2$. The data dimension $d$ is allowed to be large but assumed to be constant and does not grow with the sample size $n$. Here the data-generating function $f_0$ is a flexible nonlinear function that resides in an reproducing kernel Hilbert space (RKHS) $\mathcal{H}_0$ induced by a certain positive definite kernel function $k_0$, and the input space $\mathcal{X}_0$ of the true function spans only a small subset of the input features $(\mathbf{x}^1, \dots, \mathbf{x}^d)$, i.e., $\mathcal{X}_0 \subset \mathcal{X}$.

To this end, the goal of *global* variable importance estimation is to produce a variable importance score $\psi_j$ for each of the input features $(\mathbf{x}^1, \dots, \mathbf{x}^d)$ such that it can be used as a classification signal for whether $\mathbf{x}^j \in \mathcal{X}_0$. As a result, the variable selection decision can be made by threhsolding $\psi_j > s$ with a pre-defined threshold $s$. The quality of a variable selection signal $\psi_j$ can be evaluated comprehensively using a standard metric such as the *area under the receiver operating characteristic* (AUROC), which measures the Type-I and Type-II errors of variable selection decision $I(\psi_j > s)$ over a range of thresholds $s$.

## 2.1 Quantifying Model Uncertainty via Featurized GP

In the nonlinear regression scenario given by Equation (1), a classical approach to uncertainty-aware model learning is the Gaussian process (GP). Specifically, assuming that $f_0$ can be described by a flexible RKHS $\mathcal{H}_k$ governed by the kernel function $k$, the GP model imposes a Gaussian process prior $f \sim \mathcal{GP}(0, k)$, such that the function evaluated at any collection of examples follows a multivariate normal ($\mathcal{MVN}$) distribution

$$\mathbf{f} \equiv (f(\mathbf{x}_1), \dots, f(\mathbf{x}_n))^\top \sim \mathcal{MVN}(\mathbf{m}_{n \times 1}, \mathbf{K}_{n \times n}),$$

with mean $\mathbf{m}_i = m(\mathbf{x}_i)$ and covariance matrix $\mathbf{K}_{i,j} = k(\mathbf{x}_i, \mathbf{x}_j)$. The choice of the prior mean $m$ and kernel $k$ enables prior specification directly in the function space. For example, the Matérn kernel with parameter $\nu$ places a prior over $\lceil \nu \rceil - 1$ times differentiable functions, with length-scale $l^2$ and amplitude variance $\sigma^2$. As $\nu \to \infty$, this reduces to the common radial basis function (RBF) kernel $k(\mathbf{x}_i, \mathbf{x}_j) = \sigma^2 \exp(\|\mathbf{x}_i - \mathbf{x}_j\|_2^2 / l^2)$.

Under the above construction, the posterior predictive distribution of $f$ evaluated at new observations $\mathbf{x}_1^*, \dots, \mathbf{x}_{n^*}^*$ is also a multivariate normal,

$$\mathbf{f}^* | \{\mathbf{x}_i \, y_i\}_{i=1}^n \sim \mathcal{MVN}(\mathbb{E}[\mathbf{f}^*], \text{Cov}[\mathbf{f}^*]), \quad \text{where} \tag{2}$$

$$\mathbb{E}[\mathbf{f}^*] = \mathbf{m}^* + \mathbf{K}^*(\mathbf{K} + \sigma^2 \mathbf{I}_n)^{-1}(\mathbf{y} - \mathbf{m}); \quad \text{Cov}[\mathbf{f}^*] = \mathbf{K}^{**} - \mathbf{K}^*(\mathbf{K} + \sigma^2 \mathbf{I}_n)^{-1}\mathbf{K}^{*\top},$$

with $\mathbf{m}_i^* = m(\mathbf{x}_i^*)$, $\mathbf{K}_{ij}^* = k(\mathbf{x}_i^*, \mathbf{x}_j)$, and $\mathbf{K}_{ij}^{**} = k(\mathbf{x}_i^*, \mathbf{x}_j^*)$. Equation (2) is known as the kernel-based representation (or dual representation) of a GP Rasmussen and Williams [2005]. Although mathematically elegant, the posterior (2) is expensive to compute due to the need to invert the $n \times n$ matrix $(\mathbf{K} + \sigma^2 \mathbf{I})$.

**Feature-based Representation of A GP**. Alternatively, Mercer's theorem Cristianini and Shawe-Taylor [2000] states that as long as the kernel function $k(\cdot, \cdot)$ can be written as the inner product of a set of basis functions $\phi(\mathbf{x}) = \{\phi_k(\mathbf{x})\}_{k=1}^D$, such that $k(\mathbf{x}, \mathbf{x}') = \phi(\mathbf{x})^\top \phi(\mathbf{x}')$, then elements of the RKHS $f \in \mathcal{H}_k$ can be written in terms of a linear expansion of basis functions Rasmussen and Williams [2005]:

$$f(\mathbf{x}) = \sum_{k=1}^D \boldsymbol{\beta}_k \phi_k(\mathbf{x}) = \phi(\mathbf{x})^\top \boldsymbol{\beta}, \text{ where } \boldsymbol{\beta} \sim \mathcal{MVN}(\boldsymbol{\mu}, \mathbf{I}_D). \tag{3}$$

This is known as the feature-based representation (or primal representation) of a GP. Notice that (3) is not an approximation method but an *exact* reparametrization of the GP model whose kernel function is induced by feature representation $\phi(\mathbf{x})$. Also note that under this featurized representation (3), the predictive model $f$ is linear in terms of the model parameters $\boldsymbol{\beta} = \{\boldsymbol{\beta}_k\}_{k=1}^D$. However, this "linearity" in the model parameters does not restrict the expressiveness of $f$, since the GP model is essentially learning to use the weights $\{\boldsymbol{\beta}_k\}_{k=1}^D$ to flexibly combine the nonlinear basis functions $\{\phi_k\}_{k=1}^D$ to best fit the outcome. Furthermore, the basis functions $\{\phi_k(\mathbf{x})\}_{k=1}^D$ can be updated as part of the learning process, which we discuss in the sequel.

**Scalable Posterior Computation via Minibatch Updates**. The above feature-based representation is powerful in that it reduces the GP posterior inference into a Bayesian linear regression problem for $\boldsymbol{\beta}$. This brings two concrete benefits. First, the posterior of $\boldsymbol{\beta}$ in Equation (3) adopts a closed form:

$$\boldsymbol{\beta} \sim \mathcal{MVN}(\mathbb{E}[\boldsymbol{\beta}], \text{Cov}[\boldsymbol{\beta}]), \quad \text{where} \tag{4}$$

$$\mathbb{E}[\boldsymbol{\beta}] = \boldsymbol{\mu} + \Sigma_{\boldsymbol{\beta}} \Phi^\top (\mathbf{y} - \Phi \boldsymbol{\mu}) / \sigma^2; \quad \text{Cov}[\boldsymbol{\beta}] = \Sigma_{\boldsymbol{\beta}} = (\Phi^\top \Phi / \sigma^2 + \mathbf{I})^{-1},$$

where $\Phi = (\phi(\mathbf{x}_1)^\top, \ldots, \phi(\mathbf{x}_n)^\top)^\top \in \mathbb{R}^{n \times D}$ is the feature matrix evaluated on the training data Rasmussen and Williams [2005]. For large-scale applications, Equation (4) enables us to compute the exact posterior of $\boldsymbol{\beta}$ in a mini-batch fashion. For example, the posterior matrix $\text{Cov}[\boldsymbol{\beta}] = \Sigma_{\boldsymbol{\beta}}$ can be updated using the Woodbury identity:

$$\Sigma_{\boldsymbol{\beta},t+1} = \Sigma_{\boldsymbol{\beta},t} - \Sigma_{\boldsymbol{\beta},t}\Phi_m^\top(\sigma^2\mathbf{I} + \Phi_m\Sigma_{\boldsymbol{\beta},t}\Phi_m^\top)^{-1}\Phi_m\Sigma_{\boldsymbol{\beta},t}, \tag{5}$$

where $\Phi_m$ is the $D$-dimension batch-specific feature matrix evaluated on the mini-batch. Similarly, the posterior mean $\mathbb{E}[\boldsymbol{\beta}]$ can be computed by accumulating the $D \times 1$ vector $\Phi^\top(\mathbf{y} - \Phi\boldsymbol{\mu}) = \sum_m \Phi_m^\top(\mathbf{y}_m - \Phi_m\boldsymbol{\mu})$, and computing the posterior mean according to Equation (4) at the end.

The posterior distribution of $\boldsymbol{\beta}$ induces a GP posterior for the prediction function $\mathbf{f}^* = \Phi^*\boldsymbol{\beta}$, where $\Phi^*$ is the feature map evaluated on the test data, with mean $\mathbb{E}[\mathbf{f}^*] = \Phi^*\boldsymbol{\mu} + \Phi^*\Sigma_{\boldsymbol{\beta}}\Phi^\top(\mathbf{y} - \Phi\boldsymbol{\mu})/\sigma^2$ and covariance $\text{Cov}[\mathbf{f}^*] = \Phi^*\Sigma_{\boldsymbol{\beta}}\Phi^{*\top}$. This distribution is equivalent to the kernel-based representation (2) but reduces the computational complexity from cubic time $O(n^3)$ to linear time $O(n)$ and is minibatch compatible (i.e., Equation (5)). Algorithm 1 and 3 provides a summary of the learning algorithm. Finally, we note that the basis functions $\phi = \{\phi_k\}_{k=1}^D$ can also be updated as part of the learning procedure (e.g., via *maximum a posteriori* (MAP) inference), which we discuss in Appendix A.4.

**Incorporating Modern ML Model Classes**. The second key advantage of the feature-based representation (3) is its generality: a wide range of machine learning models can be written in the feature-based form $f(\mathbf{x}) = \phi(\mathbf{x})^\top\boldsymbol{\beta}$ Rahimi and Recht [2007], Davies and Ghahramani [2014], Lee et al. [2017], making the GP a unified framework for quantifying model uncertainty for a wide array of modern ML models. Appendix B summarizes important examples including GAMs, decision trees, random-feature models, deep neural networks and their ensembles. Appendix B.1 summarizes a list of general conditions the model should satisfy for it to be compatible with the proposed framework (i.e., weak differentiability, Lipschitz condition, and growth rate of model complexity). Furthermore, when a deterministically-trained $\hat{\boldsymbol{\beta}}$ is available (e.g., via a sophisticated adaptive shrinkage procedure that is not available in a Bayesian context), we can incorporate this as prior knowledge into GP modeling by setting $\boldsymbol{\mu} = \hat{\boldsymbol{\beta}}$ (Equation (3)).

## 2.2 Bayesian Nonparametric Guarantees for Probabilistic Learning

The quality of a Bayesian learning procedure is commonly measured by the learning rate of its posterior distribution $\Pi_n = \Pi(\cdot \mid \{\mathbf{x}_i, y_i\}_{i=1}^n)$. Intuitively, the rate of this convergence is measured by the size of the smallest shrinking balls around $f_0$ that contains most of the posterior probability. Specifically, we consider the size of the set $A_n = \{g \mid \|g - f_0\|_n^2 \leq M\epsilon_n\}$ such that $\Pi_n(A_n) \to 1$ [Ghosal and Vaart, 2007, Polson and Rockova, 2018]. The concentration rate $\epsilon_n$ here indicates how fast the small ball $A_n$ concentrates towards $f_0$ as the sample size increases. Below we state the formal definition of posterior convergence Ghosal and Vaart [2007].

**Definition 1** (Posterior Convergence). For $f_0 : \mathcal{X} \to \mathbb{R}$ where $\mathcal{X} = \mathbb{R}^d$, let $\mathcal{H}_0$ denote the true RKHS induced by a kernel function $k_0$, and let $\mathcal{H}_\phi$ denote the RKHS induced by the feature function $\phi : \mathcal{X} \to \mathbb{R}^D$. Let $f_0 \in \mathcal{H}_0$ be the true function, and let $\mathbb{E}_0$ denote the expectation with respect to the true data-generation distribution. Assuming $\mathcal{H}_\phi$ is dense in $\mathcal{H}_0$, then, the posterior distribution $\Pi_n(f)$ concentrates around $f_0$ at the rate $\epsilon_n$ if there exists an $\epsilon_n \to 0$ such that, for any $M_n \to \infty$,

$$\mathbb{E}_0\Pi_n(f : \|f - f_0\|_n^2 \geq M_n\epsilon_n) \to 0. \tag{6}$$

Notice that we allow the model space $\mathcal{H}_\phi$ and the true function space $\mathcal{H}_0$ to be different, but $\mathcal{H}_\phi$ must be *dense* in $\mathcal{H}_0$ for the convergence to happen. Fortunately, this condition is shown to hold for a wide variety of ML models, including random features, random forests, and neural networks [Biau, 2012, Hornik et al., 1989, Rahimi and Recht, 2008, Schmidt-Hieber, 2020, Ročková and van der Pas, 2020]. The notion of posterior convergence can also be used to discuss the learning quality of other probabilistic estimates (e.g., variable importance $\psi_j$). In that case, we can simply replace $(f, f_0)$ in (6) by their variable importance counterparts. This is the focus of Section 3.2.

# 3 Methods

## 3.1 Quantifying Variable Importance under Uncertainty

In this work, we consider quantifying the *global* importance of a variable based on the norm of the corresponding partial derivative. This is motivated by the observation that, if a function $f$ is

differentiable, the relative importance of a variable $\mathbf{x}^j$ at a point $\mathbf{x}$ can be captured by the magnitude of the partial derivative function, $|\frac{\partial}{\partial \mathbf{x}^j} f(\mathbf{x})|$ Rosasco et al. [2013]. This quantity requires the consideration of two issues. First, instead of quantifying the relevance of a variable on a single input point, we need to define a proper global notion of variable importance. Therefore, it is natural to integrate this partial derivative over the input space $\mathbf{x} \in \mathcal{X} : \Psi_j(f) = \|\frac{\partial}{\partial \mathbf{x}^j} f\|_{P_\mathcal{X}}^2 = \int_{\mathbf{x} \in \mathcal{X}} |\frac{\partial}{\partial \mathbf{x}^j} f(\mathbf{x})|^2 \, dP_\mathcal{X}(\mathbf{x})$. Second, since $P_\mathcal{X}(\mathbf{x})$ is not known from the training observations, $\Psi_j(f)$ can be approximated by its empirical counterpart,

$$\psi_j(f) = \|\frac{\partial}{\partial \mathbf{x}^j} f\|_n^2 = \frac{1}{n} \sum_{i=1}^n |\frac{\partial}{\partial \mathbf{x}^j} f(\mathbf{x}_i)|^2. \tag{7}$$

Notice that $\psi_j(f)$ is an estimator that is derived from the prediction function $f$ estimated using finite data. Consequently, to make a proper decision regarding the importance of an input variable $\mathbf{x}^j$, it is important to take into account uncertainty in $f$. To this end, by leveraging the featurized GP representation introduced in Section 3.1, we show that this can be done easily for a wide range of ML models $f(\mathbf{x}) = \phi(\mathbf{x})^\top \boldsymbol{\beta}$ by studying the posterior distribution of $\psi_j$.

**Posterior Distribution of Variable Importance**. After we obtain the posterior distribution of $\boldsymbol{\beta}$ (4), the posterior distribution of variable importance can be derived according to Equation (7):

$$\psi_j(f) = \frac{1}{n} |\frac{\partial}{\partial \mathbf{x}^j} f(\mathbf{X})|^\top |\frac{\partial}{\partial \mathbf{x}^j} f(\mathbf{X})| = \frac{1}{n} \boldsymbol{\beta}^\top \frac{\partial \Phi}{\partial \mathbf{x}^j} \frac{\partial \Phi^\top}{\partial \mathbf{x}^j} \boldsymbol{\beta}, \tag{8}$$

where $\frac{\partial \Phi}{\partial \mathbf{x}^j} \in \mathbb{R}^{D \times n}$ is the derivative of the feature map with respect to $\mathbf{x}^j$, across $n$ training samples. The posterior distribution of $\psi_j(f)$ adopts a closed form as a generalized chi-squared distribution (see Appendix A.2 for derivation). In practice, we can sample $\psi_j$ conveniently from its posterior distribution by computing $\frac{\partial}{\partial \mathbf{x}^j} f(\mathbf{X}) = \left(\frac{\partial \Phi}{\partial \mathbf{x}^j}\right)^\top \boldsymbol{\beta}^{(s)}$, where $\boldsymbol{\beta}^{(s)}$ are Monte Carlo samples from the closed-form posterior (4).

There are two ways in which uncertainty aids the variable importance estimation process. First, the posterior survival function $P(\psi_j(f) > s)$ of the variable importance utilizes the full posterior distribution of $\psi_j(f)$ to identify the probability that the variable $\mathbf{x}^j$ exceeds a given threshold $s$. By increasing $s \in (0, \infty)$, $P(\psi_j > s)$ provides an intuitive sense of how a model's belief about the importance of variable $\mathbf{x}^j$ changes as the criteria $s$ becomes more stringent, similar to the regularization path used by LASSO methods [Friedman et al., 2010] but with the incorporation of posterior uncertainty about the variable importance. See Appendix I for an application to a Bangladesh birth cohort study. Second, by integrating the survival function over the threshold, i.e., $\int_{s>0} P(\psi_j(f) > s) \, ds$, we obtain the posterior mean of $\psi_j(f)$, and this too incorporates uncertainty in $f$. To see this, notice that by using the "trace trick" we can write

$$\mathbb{E}[\psi_j(f)] = \mathbb{E}\left[\text{tr}\left(\boldsymbol{\beta}^\top \frac{\partial \Phi}{\partial \mathbf{x}^j} \frac{\partial \Phi^\top}{\partial \mathbf{x}^j} \boldsymbol{\beta}\right)\right] = \mathbb{E}[\boldsymbol{\beta}]^T \frac{\partial \Phi}{\partial \mathbf{x}^j} \frac{\partial \Phi^\top}{\partial \mathbf{x}^j} \mathbb{E}[\boldsymbol{\beta}] + \text{tr}\left(\frac{\partial \Phi}{\partial \mathbf{x}^j} \frac{\partial \Phi^\top}{\partial \mathbf{x}^j} \text{Cov}[\boldsymbol{\beta}]\right), \quad (9)$$

where all expectations are taken with respect to the posterior. Therefore, the posterior mean of $\psi_j(f)$ depends on the covariance structure of $\boldsymbol{\beta}$, and how it interacts with the eigenspace of the partial derivative functions (encoded by $\frac{\partial \Phi}{\partial \mathbf{x}^j} \frac{\partial \Phi^\top}{\partial \mathbf{x}^j}$). In Section 4 we provide an extensive investigation of AUROC scores using the posterior mean of $\psi_j(f)$ for quantifying variable importance.

In Appendix A.3, we summarize the algorithms for computing the posterior distributions of the featurized Gaussian process (Equation (4)) and for the posterior distributions of variable importance (Equation (8)), and discuss their space and time complexity.

## 3.2 Theoretical Guarantees

From a theoretical perspective, the variable importance measure $\psi_j$ introduced in (7) can be understood as a quadratic functional of the GP model $f$ Efromovich and Low [1996]. To this end, rigorous Bayesian nonparametric guarantees can be obtained for $\psi_j$'s ability in learning the true variable importance in finite samples (i.e., posterior convergence, Theorem 1) and its statistical optimality from a frequentist perspective, in providing a low-variance estimator that attains the Cramér-Rao bound (i.e., Bernstein von-Mises phenomenon, Theorem 2). Note that for a given general model $f(\mathbf{x}) = \phi(\mathbf{x})^\top \boldsymbol{\beta}$, it only need to satisfy three mild regularity conditions to be fully compatible with the proposed framework (i.e., weak differentiability, Lipschitz condition, and growth rate of model

complexity). We summarize these conditions in Appendix B.1 and explain them in detail in the sequel.

**Posterior Convergence**. We first show that, for an ML model $f$ that can learn the true function $f_0$ with rate $\epsilon_n$ (in the sense of Definition 1), the entire posterior distribution of the variable importance measure $\psi_j(f)$ converges consistently to a point mass at the true $\Psi_j(f_0)$ at a speed that is equal or faster than $\epsilon_n$.

**Theorem 1** (Posterior Convergence of Variable Importance $\psi_j$). *Suppose $y_i = f_0(\mathbf{x}_i) + e_i$, $e_i \overset{i.i.d.}{\sim} \mathcal{N}(0, \sigma^2)$, and denote as $\mathbb{E}_0$ the expectation with respect to the true data-generation distribution centered around $f_0$. For the RKHS $\mathcal{H}_\phi$ induced by the feature function $\phi : \mathcal{X} \to \mathbb{R}^D$ and $f \in \mathcal{H}_\phi$, if:*

*(1) The posterior distribution $\Pi_n(f)$ converges toward $f_0$ at a rate of $\epsilon_n$;*

*(2) The differentiation operator $D_j : f \to \frac{\partial}{\partial \mathbf{x}^j} f$ is bounded: $\|D_j\|_{op}^2 = \inf\{C \geq 0 : \|D_j f\|_2^2 \leq C\|f\|_2^2, \text{ for all } f \in \mathcal{H}_\phi\}$;*

*Then the posterior distribution for $\psi_j(f) = \|\frac{\partial}{\partial \mathbf{x}^j} f\|_n^2$ contracts toward $\Psi_j(f_0) = \|\frac{\partial}{\partial \mathbf{x}^j} f_0\|_{P_\mathcal{X}}^2$ at a rate not slower than $\epsilon_n$. That is, for any $M_n \to \infty$,*

$$\mathbb{E}_0 \Pi_n \left[ \sup_{j \in \{1,\dots,d\}} |\psi_j(f) - \Psi_j(f_0)| \geq M_n \epsilon_n \right] \to 0.$$

The proof is in Appendix C. Theorem 1 is a generalization of the classical result of quadratic functional convergence under linear models and sparse neural networks to a much wider range of ML models in the context of Bayesian variable importance estimation [Efromovich and Low, 1996, Liu, 2021, Wang and Rocková, 2020]. It confirms the important fact that, for an ML model $f$ that can accurately learn the true function $f_0$ under finite data, we can consistently recover the true variable importance at a fast rate by using the proposed variable importance estimate $\psi_j(f)$, despite the potential lack of identifiablity in the model parameters (e.g., weights in a neural network). Importantly, although our main setting assumes fixed data dimension $d$ (see Problem Setup), the posterior concentration result Theorem 1 does not rely on this assumption in its proof, and is in fact compatible with the high-dimensional setting where $d$ is allowed to grow with sample size at a rate of $o(n)$. See Appendix C (in particular, Remark 4) for further discussion.

From a practical point of view, Theorem 1 reveals that the finite-sample performance of variable importance $\psi_j(f)$ depends on two factors: (1) the finite-sample generalization performance of the prediction function $f$, and (2) the mathematical property of $f$ in terms of its Lipschitz condition. Therefore, to ensure effective variable importance estimation in practice, the practitioner should take care to select a model class $f$ that has a theoretical guarantee in capturing the target function $f_0$, empirically delivers strong generalization performance under finite data, and is well-conditioned in terms of the behavior of its partial derivatives. To this end, we note that, under the featurized Gaussian process $f = \phi(\mathbf{x})^\top \boldsymbol{\beta}$ discussed in this work, users are free to choose a performant model class (e.g., random forest, random-feature or DNN) whose feature representation spans an RKHS $\mathcal{H}_\phi$ that is *dense* in the infinite-dimensional function space (therefore $f$ enjoys a convergence guanrantee, see Remark 3 in Appendix C for further discussion) [Biau, 2012, Hornik et al., 1989, Rahimi and Recht, 2008, Schmidt-Hieber, 2020, Rocková and van der Pas, 2020], and is empirically more effective than the GP methods based on classical kernels such as RBF. We discuss the Lipschitz condition of these models in Appendix E.1. Indeed, as we will verify in experiments (Section 4), there does not exist an "optimal" model class that performs universally well across all data settings (i.e., no free lunch theorem [Wolpert and Macready, 1997]). This highlights the importance of having a general-purpose framework for variable importance estimation that can flexibly incorporate the most effective model for the task at hand. Finally, we notice that although Theorem 1 is stated as an asymptotic result, when a finite-sample error bound $\epsilon_n$ for the model class is available (i.e., Condition (1) in Theorem 1), it is trivial to obtain a finite-sample error bound for variable importance $\psi_j(f)$ by extending the proof of Theorem 1. Appendix C.1 provides an example of such a bound based on the Bernstein inequality.

**Statistical Efficiency** & **Uncertainty Quantification.** Next, we verify the uncertainty quantification ability of the variable importance measure $\psi_j(f)$ under a featurized GP by showing that it exhibits the *Bernstein-von Mises* (BvM) phenomenon. That is, its posterior measure $\Pi_n(\psi_j(f))$ converges towards a Gaussian distribution that is centered around the truth $\Psi_j(f_0)$, so that its $(1-\alpha)\%$ level

credible intervals achieve the nominal coverage probability for the true variable importance. More importantly, the BvM theorem verifies that the posterior distribution of $\psi_j(f)$ is *statistically optimal*, in the sense that its asymptotic variance attains the Cramér-Rao bound (CRB) that cannot be improved upon [Bickel and Kleijn, 2012].

**Theorem 2** (Bernstein-von Mises Theorem for Variable Importance $\psi_j$). *Suppose $y_i = f_0(\mathbf{x}_i) + e_i$, $e_i \overset{i.i.d.}{\sim} \mathcal{N}(0, \sigma^2)$, $i = 1, \ldots, n$. Denote $D_j : f \rightarrow \frac{\partial}{\partial \mathbf{x}^j} f$ the differentiation operator and $H_j = D_j^\top D_j$ the inner product of $D_j$, such that:*

$$\psi_j(f) = \|D_j(f)\|_n^2 = \frac{1}{n}\langle D_j f, D_j f \rangle = \frac{1}{n} f^\top H_j f. \tag{10}$$

*Assuming conditions (1)-(2) in Theorem 1 hold, and additionally:*

    *(3) $f_0$ is square-integrable over the support $\mathcal{X}$ and $\|f_0\|_2 = 1$;*

    *(4) $\mathtt{rank}(H_j) = o_p(\sqrt{n})$;*

*Then*

$$\sqrt{n}(\psi_j(f) - \psi_j(f_0)) \overset{d}{\rightarrow} \mathcal{N}(0, 4\sigma^2 \|H_j f_0\|_n^2).$$

The proof is in Appendix D. Theorem 2 provides a rigorous theoretical justification for $\psi_j(f)$'s ability to quantify its uncertainty about the variable importance. More importantly, it verifies that $\psi_j(f)$ has the good frequentist property that it quickly converges to a minimum-variance estimator at a fast speed, which is important for obtaining good variable importance estimation performance in practice. Compared to the previous BvM results that tend to focus on a specific Bayesian ML model, Theorem 2 is considerably more general (i.e., applicable to a much wider range of models) and comes with a simpler set of conditions [Rockova, 2020, Wang and Rocková, 2020, Liu, 2021]. Specifically, (3) is a standard assumption in nonparametric analysis. It ensures the true function $f_0$ does not diverge towards infinity and makes learning possible [Castillo and Rousseau, 2015]. The unit norm assumption $\|f_0\|_2 = 1$ is only needed to simplify the exposition of the proof, and the theorem can be trivially extended to $\|f_0\|_2 = C$ for any $C > 0$. The most interesting condition is (4). Let us denote $\mathcal{H}_j$ as the space of partial derivatives functions $\frac{\partial}{\partial \mathbf{x}^j} f$ of the model functions $f \in \mathcal{H}_\phi$. Then, intuitively, (4) says that to attain the BvM phenomenon, the effective dimension of the derivative function space $\mathcal{H}_j$ (as measured by $\mathtt{rank}(H_j) = \mathtt{rank}(D_j)$) cannot be too large. Since the effective dimension of the derivative space is bounded above by that of the original RKHS $f \in \mathcal{H}_\phi$, (4) essentially states that the effective dimension of the model space $\mathcal{H}_\phi$ cannot grow too fast with data size (i.e., $o_p(\sqrt{n})$). Fortunately, this condition is satisfied by a wide range of ML models including trees and deep networks [Rockova, 2020, Wang and Rocková, 2020]. See Appendix E.2 for further discussion.

## 4 Experiment Analysis

In this section, we investigate the finite-sample performance of the derivative norm metric $\psi_j$ for variable importance estimation (7) under a wide variety of ML methods. We illustrate the breath of our framework by applying it to tree ensembles (Appendix F.1), where a principled and gradient-based uncertainty-aware variable importance estimation approach has been previously unavailable. We also apply it to linear models and (approximate) kernel machines, which are standard approaches to variable selection in data science practice [Tibshirani, 1996, Bobb et al., 2015]. Over a wide range of complex and realistic data scenarios (e.g., discrete features, interactions, between-feature correlations) derived from socioeconomic and healthcare datasets, we investigate the method's statistical performance in accurately recovering the ground-truth features (in terms of the Type I and Type II errors), and compare it to other well-established approaches in each of the model classes (Table 1). Our main observations are:

**O1**: **Importance of generality**. There does not exist a model class that performs universally well across all data scenarios (i.e., no free lunch theorem [Wolpert and Macready, 1997], Figures 1, 6-15). This highlights the importance of an unified framework for variable importance that incorporates a wide range of models, so that practitioners have the freedom of choosing the most suitable model class for the task at hand.

**O2**: **Good prediction translates to effective variable importance estimation**. Comparing between different model classes, the ranking of models' predictive accuracy is generally consistent with the

ranking of their variable importance estimation performance under $\psi_j$ (i.e., better prediction translates to better variable importance estimation, as suggested in Theorem 1).

**O3**: **Statistical efficiency of** $\psi_j$. Comparing within each model class, the derivative norm metric $\psi_j$ generally outperforms other measures of variable importance. The advantage is especially pronounced in small samples and for correlated features. This empirically verifies that $\psi_j$ has good finite-sample statistical efficiency even under complex data scenarios (as suggested in Theorem 2).

| Model Class | (Ours) | Baselines |
|---|---|---|
| Tree Ensembles | RF-FDT | RF-Impurity, RF-Knockoff, BART |
| Kernel Methods & NNs | RFF, NN | BKMR, BAKR |
| Linear Models | GAM | BRR, BL |

**Table 1:** Summary of methods considered in the experiments.

**Models** & **Methods**. We consider three main classes of models (Table 1): (I) **Random Forests (RF)**. Given a trained forest, we quantify variable importance using $\psi_j$ by translating it to an ensemble of featurized decision trees (**FDT**) (Appendix F.1), and compare it to three baselines: *impurity* (**RF-impurity**) [Breiman et al., 1984], *RF-based kernel knockoff* (**RF-knockoff**) [Candes et al., 2017], and *Bayesian Additive Regression Trees* (**BART**). (II) **(Approximate) Kernel Methods** & **Neural Networks**. We apply $\psi_j$ to a random-feature model that approximates a GP with an RBF kernel [Rahimi and Recht 2007], and set the number of features to $\sqrt{n}\,log(n)$ to ensure proper approximation of the exact RBF-GP [Rudi and Rosasco 2018], which is termed *Random Fourier Feature model* (**RFF**). We also apply $\psi_j$ to *Neural Networks* (**NN**) based on wide ReLU neural network with 512 hidden units and LASSO regularization in the hidden layer weights [Lemhadri et al., 2021]. We compare them to *Bayesian Kernel Machine Regression* (**BKMR**) Bobb et al. [2015] based on a GP with an exact RBF kernel and a spike-and-slab prior, and *Bayesian Approximate Kernel Regression* (**BAKR**) based on random-feature model with a projection-based feature importance measure and an adaptive shrinkage prior [Crawford et al., 2018]. (III) **Linear Models**. We apply $\psi_j$ to a featurized GP representation of the *Generalized Additive Model* (**GAM**), with the prior center $\boldsymbol{\mu}$ set at the frequentist estimate of the original GAM model obtained from a sophisticated REML procedure [Wood, 2006]. We compare it to two baselines: *Bayesian Ridge Regression* (**BRR**) Hoerl and Kennard [1970] and *Bayesian LASSO* (**BL**) Park and Casella [2008]. Appendix G provides further detail. Previously, [Liu, 2021] studied the specialization of our framework to the deep neural networks (DNNs), so we do not repeat that work here as DNN is not yet a standard data science model for tabular data.

To quantify variable importance while accounting for posterior uncertainty of the variable importance $\psi_j(f)$, we examine its posterior survival function $\int_{s>0} P(\psi_j(f) > s)\, ds$ (i.e., the posterior likelihood of $\psi_j(f)$ being greater than the threshold $s$ integrated over the full range of thresholds $s$). For other methods, we use their default metrics to quantify variable importance (e.g., variable inclusion probabilities in **BART** and **BKMR**. See Appendix G).

**Datasets and Tasks.** We consider two synthetic benchmark datasets and three real-world socio-economic and healthcare datasets, encapsulating challenging phenomena such as between-feature correlations and interaction effects. For the synthetic benchmark datasets, we generate data under the Gaussian noise model $y \sim \mathcal{N}(f_0, 0.01)$ for four types of outcome-generation functions $f_0$ (`linear`, `rbf`, `matern32` and `complex`, see Appendix G.2 for a full description) with the number of causal variables set at $d^\star = 5$. We consider two types of feature distribution: (1) **synthetic-continuous**: all features follow $\mathbf{x}^j \sim Unif(-2, 2)$; (2) **synthetic-mixture**: two of the causal features and two of the non-causal features are distributed as $Bern(0.5)$ and the rest are distributed as $Unif(-2, 2)$. Features in both distributions are independent. We vary sample size $n \in \{100, 200, 500, 1000\}$ and data dimension $d \in \{25, 50, 100, 200\}$, leading to 128 total scenarios.

For real-world data, we consider (1) **adult**: 1994 U.S. census data of 48842 adults with eight categorical and six continuous features Kohavi; (2) **heart**: a coronary artery disease dataset of 303 patients from Cleveland clinic database with seven categorical and six continuous features Detrano et al. [1989]; and (3) **mi**: disease records of myocardial infarction (MI) of 1700 patients from Krasnoyarsk interdistrict clinical hospital during 1992-1995, with 113 categorical and 11 continuous features Golovenkin et al. [2020]. All datasets exhibit non-trival correlation structure among features (Appendix Figures 3-5). Since the ground-truth causal features on these datasets are not known, in order to rigorously evaluate variable importance estimation performance, we follow the standard practice in causal ML to simulate the outcome based on causal features selected from data [Yao et al., 2021]. We use the four outcome-generating functions as described previously and evaluate

over the same data size × dimension combinations, leading to 192 total scenarios[2]. We repeat the simulation 20 times for each scenario, and use AUROC to measure the variable importance estimation performance (in terms of Type I and Type II errors) of each method.

In Appendix I, we further evaluate the method on a well-studied environmental health dataset (Bangladesh birth cohort study [Kile et al., 2014]) with respect to the real outcome (infant development scores). We visualize the "Bayesian" regularization path as introduced earlier. The selected variables correspond well with the established toxicology pathways in the literature [Gleason et al., 2014].

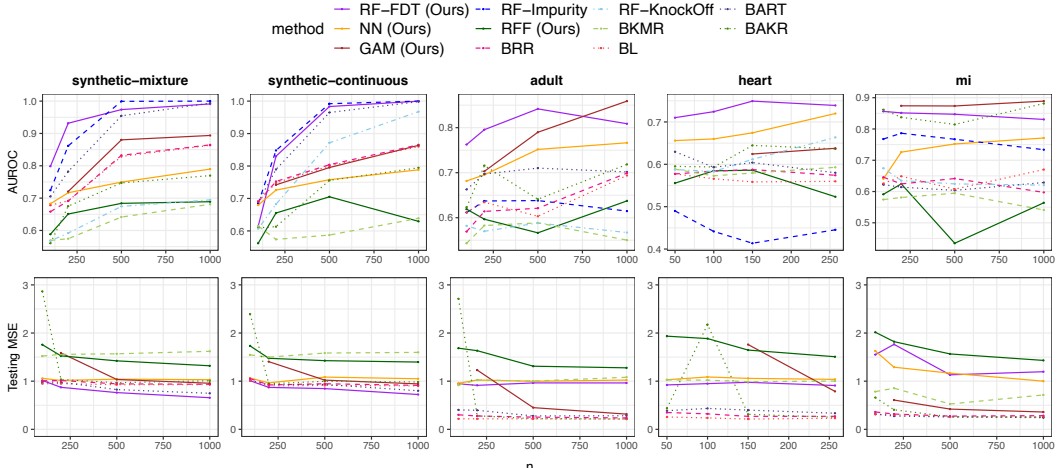

**Figure 1:** Method performance in variable importance estimation (measured by AUROC, row 1) and prediction (measured by test MSE, row 2) under `matern32` data-generation function and with input dimension 100 (five causal features). Therefore, the variable importance scores of the five causal features are expected to be higher than the other 95 variable importance estimations. The x-axis represents the training sample sizes $n \in \{100, 200, 500, 1000\}$. **GAM** does not produce valid results for case of $n \leq d$ so the results from this model in these cases are not shown. The ranking of **FDT** (solid purple) outperforms other methods in most of the data settings, and **GAM** outperforms in the setting of large data size and high percentage of categorical features (**adult** and **mi**). The rankings of performance are roughly consistent between prediction and variable importance estimation.

## 4.1 Results

Figure 1 shows the methods' performance in variable importance estimation (**Row 1**) and prediction (**Row 2**)[3] in an exemplary setting, where the true function $f_0$ is `matern32` with an input dimension $d = 100$. It represents the tabular data setting that we are the most interested in: nonlinear feature-response relationship with interaction effects and high input dimension. This is because $f_0$ is sampled from an RKHS induced by Mátern $\frac{3}{2}$ kernel, which contains a large space of continuous and at least once differentiable functions [Rasmussen and Williams, 2005]. We delay complete visualizations for all 320 scenarios to Appendix H. Recalling the three observations introduced earlier:

**O1 ("No free lunch")**: No method performs universally well. For example, **BAKR** performs robustly in correlated datasets (**heart** and **mi**), but poorly otherwise. Kernel approaches (**RFF** and **BKMR**) perform competitively in low dimension, but their performances deteriorate quickly as dimension $d$ increases (Figure 1 and Figure 6-7). This is likely due to the classical kernel method's well-known inability to learn an adaptive feature representation, which consequently leads to suffering from the curse of dimensionality and unstable and suboptimal variable importance performance in high dimensions [Bach, 2017]. **FDT** is generally the strongest method in small samples and high dimensions, but can be outperformed by **GAM** in large samples and data with a high percentage of categorical features (**adult** and **mi**). Notably, they often outperform **NN**, which is traditionally regarded as the go-to model for high-dimensional nonlinear settings. This highlights the importance of a unified framework that allows users to select the most appropriate model for variable importance estimation depending on the data setting.

---

[2]In the setting where required data dimension is higher than that of the real data, we generate additional synthetic features from $Unif(-2, 2)$. We use $n \in \{50, 100, 150, 257\}$ for **heart** due to data size restrictions.

[3]For the prediction plots, a method will not be visualized if they share the model fit with another method (**RF-impurity** and **RF-knockoff**), or if it does not produce valid results due to small sample size (**GAM**).

**O2 ("Good prediction implies effective variable importance estimation"):** Fixing the variable importance $\psi_j$ and comparing the variable importance estimation performance of each model class (i.e., **FDT**, **GAM**, **NN** and **RFF**, which are solid lines in Figure 1), we see that their rankings in prediction (row 2) are largely consistent with the corresponding rankings in variable importance estimation. It is worth noting that this pattern is occasionally violated (e.g., **GAM** in **adult**, $n = 500$ and **heart**, $n = 250$), but that does not contradict our conclusion (Theorem 1) since the convergence rate of the prediction function only forms an *upper bound* for the convergence rate of $\psi_j$. Finally, when models have comparable generalization performance, we observe that the Lipschitz condition plays a role in variance importance performance (which is consistent with our theoretical observations in Theorem 1). For example, in Figure 1, **NN** and **FDT** are largely comparable in predictive performance among the real datasets (**adult**, **heart** and **mi**). However, tree-based **FDT** are known to have well-conditioned Lipschitz behavior when compared to **NN** (Appendix E.1), which is consequently translated to improved finite-sample performance in variable importance estimation.

**O3 (Statistical efficiency of $\psi_j$):** When comparing among variable importance estimation methods from the same class (especially for tree models, i.e., **FDT** v.s. **RF-impurity / RF-knockoff / BART**), we see that **FDT** is competitive or strongly outperforms its baselines in variable importance estimation, despite being based on exactly the same fitted model (**RF-impurity / RF-knockoff**), or not accounting for the uncertainty in the tree growing process (**BART**). This pattern is consistent in most data settings, and the advantage is especially pronounced in high dimensions, small data sizes, and correlated datasets (Appendix H, Figure 6-10). This provides strong empirical evidence for the fact that $\psi_j$ is a statistically efficient estimator for variable importance with good finite-sample behavior (as suggested in Theorem 2), and can deliver strong performance for tabular data when combined with a performant ML model like random forests. Appendix H contains further discussion.

## 5 Discussion and Future Directions

The modern data analysis pipeline typically involves fitting multiple models, comparing their performance, and iterating as necessary. When variable selection is involved, the practitioner may ask *are the variable importance scores across models measuring the same behavior?* And, *what if the most suitable model does not have a satisfactory variable importance estimation procedure?* By framing model choice as the specification of a kernel — which includes kernels corresponding machine learning methods like neural networks and random forests in addition to the long list of traditional kernels — we propose a unified variable importance estimation procedure that is compatible across models and prove strong guarantees for this procedure.

**Limitations**. We do not consider uncertainty in the feature map itself. For example, the kernel induced by the featurized decision tree studied here does not consider uncertainty in the tree's partitioning process. Meanwhile, the fact that the full posterior inference is performed only with respect to $\beta$ indeed places a limitation on the model's ability in uncertainty quantification, as the uncertainty in the model hyperparameters is not accounted for. Yet, this does not seem to be a **significant limitation in the method's empirical performance** (e.g., **FDT** outperforms **BART** in our experiments), although this point still merits further investigation in the future. On the other hand, in the future, it would be worth expanding this framework to other model classes (e.g., MARS Friedman [1991]) and estimating the importance of interaction effects and higher-order terms (see Appendix K for details).

**Societal Impacts**. We expect the method proposed to provide a set of powerful tools for practitioners to understand the importance of input variables in their ML models with limited data, which is especially important for scientific investigations in the fields of epidemiology and computational biology. However, we recognize that this approach can potentially be utilized by bad actors to probe the input-variable uncertainty of an existing ML system, and use it to engineer more targeted white-box adversarial attacks. To this end, we recommend system developers to incorporate this approach into the formal verification procedure of an ML system, so as to monitor and understand the model uncertainty with respect to input variables, and devise proper improvement and prevention strategies (e.g., data augmentation or randomized smoothing targeted at specific variables) accordingly.

## Acknowledgement

This research was supported by NIH grants ES000002, ES028800, ES028811, ES028688, and ES030990. We would also like to thank Dr. Andrew Beam for the helpful discussions.

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
