# A   Additional Background and Technical Derivations

## A.1   Neural Network Representation of Decision Tree

For each node in a learned decision tree, we know the feature the node is splitting on and its corresponding threshold. Karthikeyan et al. [2021] provides a neural network representation of a decision tree:

$$f(\mathbf{x}|\mathbf{W}, \mathbf{b}, \boldsymbol{\beta}) = \sum_{l=0}^{D} \phi_l(\mathbf{x}|\mathbf{W}, \mathbf{b})\boldsymbol{\beta}_l, \text{ where}$$

$$\phi_l(\mathbf{x}|\mathbf{W}, \mathbf{b}) = \sigma_{\texttt{step}}\Big(\sum_{i=0}^{h-1} \sigma_{\texttt{step}}\big((\mathbf{x}^\top \mathbf{w}_{i,I(i,l)} + b_{i,I(i,l)})S(i,l)\big) - h\Big). \quad (11)$$

In the above equations, $\boldsymbol{\beta}_l \in \mathbb{R}$ is the prediction given by the $l^{th}$ leaf node, $h$ is the height of the tree and $D$ is the number of leaf nodes. $I(i,l)$ denotes the index of the $l^{th}$ leaf's predecessor in the $i^{th}$ level of the tree. $\mathbf{w}_{ij} \in \mathbb{R}^d$ indicates the feature the node is splitting on using one hot encoding, with only one element being 1 or $-1$ and the rest being 0. $b_{ij} \in \mathbb{R}$ is the corresponding threshold (or the threshold multiplied by $-1$). The $-1$ is to guarantee that $\mathbf{x}^\top \mathbf{w}_{i,j} + b_{i,j} > 0$ so that when multiplied by

$$S(i,l) = \begin{cases} -1 & \text{if } l^{th} \text{ leaf } \in \text{ left subtree of node } I(i,l), \\ +1 & \text{otherwise}, \end{cases}$$

the direction of $(\mathbf{x}^\top \mathbf{w}_{i,I(i,l)} + b_{i,I(i,l)})S(i,l)$ can be kept. $\sigma_{\texttt{step}}(\cdot)$ is the step function,

$$\sigma_{\texttt{step}}(a) = 1, \text{ if } a \geq 0, \text{ and } \sigma_{\texttt{step}}(a) = 0, \text{ if } a < 0.$$

Therefore, the model space can be regarded as a three-layer neural network with $\sigma_{\texttt{step}}$ as activation function, with $\mathbf{W}$ as hidden weights and $\mathbf{b}$ as hidden bias.

## A.2   Derivation of Posterior Distribution of Variable Importance

Recall from Equation 4 that the posterior distribution of $\boldsymbol{\beta}$ is $\mathcal{MVN}(\mathbb{E}[\boldsymbol{\beta}], \text{Cov}[\boldsymbol{\beta}])$, which can be computed in closed form. This induces a distribution over the variable importance $\psi_j(f)$:

$$\begin{aligned}
\psi_j(f) &= \frac{1}{n}|\frac{\partial}{\partial \mathbf{x}^j} f(\mathbf{X})|^\top |\frac{\partial}{\partial \mathbf{x}^j} f(\mathbf{X})| \\
&= \frac{1}{n}\boldsymbol{\beta}^\top \big(\frac{\partial}{\partial \mathbf{x}^j}\phi(\mathbf{X})\big)\big(\frac{\partial}{\partial \mathbf{x}^j}\phi(\mathbf{X})\big)^\top \boldsymbol{\beta} \\
&= \frac{1}{n}\boldsymbol{\beta}^\top \mathbf{Q}\boldsymbol{\Lambda}\mathbf{Q}^\top \boldsymbol{\beta} && \text{(Eigen-decomposition on } \big(\frac{\partial}{\partial \mathbf{x}^j}\phi(\mathbf{X})\big)\big(\frac{\partial}{\partial \mathbf{x}^j}\phi(\mathbf{X})\big)^\top) \\
&= \frac{1}{n}\sum_{i=1}^{D}\lambda_i(\mathbf{q}_i^\top \boldsymbol{\beta})^2 && (\lambda_i \text{ is eigenvalue, } \mathbf{q}_i \text{ is eigenvector}) \\
&= \frac{1}{n}\sum_{i=1}^{D}(\lambda_i V_i)\cdot Z_i, && (V_i = \mathbf{q}_i^\top \text{Cov}(\boldsymbol{\beta})\mathbf{q}_i)
\end{aligned}$$

where $Z_i := (\mathbf{q}_i^\top \boldsymbol{\beta})^2/V_i \sim \chi_1^2(\mu_i)$ are independent random variables that follows a noncentral $\chi^2$ distribution with 1 degree of freedom and parameter $\mu_i = (\mathbf{q}_i^\top \mathbb{E}[\boldsymbol{\beta}])^2$. The values $\{\lambda_i \cdot V_i\}_{i=1}^{D}$ are scalar constants weighting each noncentral $\chi^2$ random variable $Z_i$. As a result, the full distribution is a well-known distribution of a linear combination of non-central $\chi^2$ distributions [Harville, 1971]. This distribution has mean $\sum_{i=1}^{D}(\lambda_i V_i)\cdot(1 + \mu_i)$, variance $\frac{2}{n}\sum_{i=1}^{D}(\lambda_i V_i)^2\cdot(1 + 2\mu_i)$, and it can be sampled efficiently from by using the linear combination representation as introduced above.

## A.3 Algorithm Summary

Given a fixed[4] feature function $\phi : \mathcal{X} \to \mathbb{R}^D$, we present algorithm summaries for (1) Computing the posterior distribution of $\boldsymbol{\beta}$ in the feature-based representation of a Gaussian process, and (2) Computing the posterior distribution of the integrated partial derivative metric.

First consider (1), it involves computing two closed-form updates (for posterior mean and variance) over the training data in mini-batches for one epoch. The algorithm has a linear complexity with respect to data size.

---

**Algorithm 1** Posterior Computation, Feature-based Representation of Gaussian Process

---

1: **Input:** Training data mini-batches $\{(\mathbf{X}_m, \mathbf{y}_m)\}_{m=1}^M$. Fixed feature function $\phi : \mathcal{X} \to \mathbb{R}^D$.
2: **Output:** Posterior mean and variance $\mathbb{E}[\boldsymbol{\beta}]_{D\times 1}, \mathrm{Cov}[\boldsymbol{\beta}]_{D \times D}$.
3: **Initialize:** Feature-label product matrix $\mathbf{P} = \mathbf{0}_{D\times 1}$, covariance matrix $\boldsymbol{\Sigma} = \mathbf{0}_{D\times D}$
4: **for** $m = 1$ **to** $M$ **do**
5:     Compute minibatch feature representation $\boldsymbol{\Phi}_m = [\phi(\mathbf{x}_1), \dots, \phi(\mathbf{x}_{n_m})]_{n_m \times D}$
6:     Update $\mathbf{P} = \mathbf{P} + \boldsymbol{\Phi}_m^\top(\mathbf{y}_m - \boldsymbol{\Phi}_m \boldsymbol{\mu})/\sigma^2$
7:     Update $\boldsymbol{\Sigma} = \boldsymbol{\Sigma} - \boldsymbol{\Sigma}\boldsymbol{\Phi}_m^\top(\sigma^2 \mathbf{I} + \boldsymbol{\Phi}_m \boldsymbol{\Sigma} \boldsymbol{\Phi}_m^\top)^{-1}\boldsymbol{\Phi}_m \boldsymbol{\Sigma}$          $\triangleright$ Equation (5)
8: **end for**
9: Compute $\mathrm{Cov}[\boldsymbol{\beta}] = \boldsymbol{\Sigma}_{\boldsymbol{\beta}} = \boldsymbol{\Sigma}$          $\triangleright$ Equation (4)
10: Compute $\mathbb{E}[\boldsymbol{\beta}] = \boldsymbol{\mu} + \boldsymbol{\Sigma}_{\boldsymbol{\beta}}\mathbf{P}$          $\triangleright$ Equation (4)

---

As shown, during mini-batch computation, the algorithm computes the posterior mean and precision matrix by linearly accumulating the statistic $\boldsymbol{\Phi}_m^\top(\mathbf{y}_m - \boldsymbol{\Phi}_m \boldsymbol{\mu})$, and performs one computation in the end to obtain the $\mathbb{E}[\boldsymbol{\beta}]$. As a result, the space complexity of the algorithm is $O(D^2)$ (for the covariance matrix) and time complexity of the algorithm is $O(nD^3)$ for the matrix inversion. In large-scale applications, the model dimension $D$ is usually fixed and is significantly smaller than the data size $n$, leading to a linear-time algorithm. Notice that in actual implementation, this algorithm can be made much more efficient (i.e., $O(nD^2)$) by changing how covariance matrix is computed. We introduce this improved algorithm at the end of this section in Algorithm 3.

Now consider (2). Given the posterior of $\boldsymbol{\beta}$ from Algorithm 1, the posterior distribution of the integrated partial derivative metric $\psi_j(f) = \|\frac{\partial}{\partial \mathbf{x}^j} f\|_n^2 = \frac{1}{n}\boldsymbol{\beta}^\top \frac{\partial \boldsymbol{\Phi}}{\partial \mathbf{x}_i^j}\frac{\partial \boldsymbol{\Phi}^\top}{\partial \mathbf{x}_i^j}\boldsymbol{\beta}$ can be computed conveniently by sampling $\boldsymbol{\beta}$ from its posterior.

---

**Algorithm 2** Posterior Computation, Integrated Partial Derivative Metric

---

1: **Input:** Data $\mathbf{X}^*$ with size $n^*$. Posterior distribution $\mathcal{MVN}(\mathbb{E}[\boldsymbol{\beta}]_{D\times 1}, \mathrm{Cov}[\boldsymbol{\beta}]_{D\times D})$.
2: **Output:** Posterior samples of $\psi_j(f)$ of size $K$: $\{\psi_j(f)_k\}_{k=1}^K$
3: Sample $\{\boldsymbol{\beta}_k\}_{k=1}^K \sim \mathcal{MVN}(\mathbb{E}[\boldsymbol{\beta}], \mathrm{Cov}[\boldsymbol{\beta}])$
4: Compute partial derivative feature matrix $[\frac{\partial \boldsymbol{\Phi}}{\partial \mathbf{x}^j}]_{D\times N^*} = [\partial\phi(\mathbf{x}_1)^\top, \dots, \partial\phi(\mathbf{x}_{N^*})^\top]^\top$
5: Compute $\mathbf{G}_{j,D\times D} = \frac{\partial \boldsymbol{\Phi}}{\partial \mathbf{x}^j}\frac{\partial \boldsymbol{\Phi}^\top}{\partial \mathbf{x}^j}$
6: Compute $\psi_j(f)_k = \frac{1}{N^*}\boldsymbol{\beta}_k^\top \mathbf{G}_j \boldsymbol{\beta}_k$ for $k = 1, \dots, K$          $\triangleright$ Equation (8)

---

When the data size is large, the $\mathbf{G}_j$ matrices can usually be computed as part of Algorithm 1 by accumulating gradient partial derivative matrices $\mathbf{G}_j = \mathbf{G}_j + \frac{\partial \boldsymbol{\Phi}_m}{\partial \mathbf{x}^j}\frac{\partial \boldsymbol{\Phi}_m^\top}{\partial \mathbf{x}^j}$. The time complexity of the algorithm is $O(D^2 n^*)$ which is again a linear-time algorithm with respect to data size $n^*$. When the data size is extremely large, one can consider reduce computational burden by subsampling from $\mathbf{X}^*$, which is equivalent to performing a Monte Carlo approximation to the integration over the empirical measure (Equation (7)).

Finally, we present a more efficient implementation of Algorithm 1, which improved the run time from $O(nD^3)$ to $O(nD^2)$ by changing how covariance matrix is computed during minibatch accumulation:

---

[4]Namely, the feature function $\phi(\mathbf{x})$ is either fixed by construction like random feature models or kernel machine using classical kernels (RBF, Matérn, etc). Or $\phi(\mathbf{x})$ is already learned elsewhere (i.e., pre-trained on the same or a separate dataset) like random forests or neural networks.

---

**Algorithm 3** Posterior Computation, Feature-based Representation of Gaussian Process (Version 2)

---

1: **Input:** Training data mini-batches $\{(\mathbf{X}_m, \mathbf{y}_m)\}_{m=1}^M$. Fixed feature function $\phi : \mathcal{X} \to \mathbb{R}^D$.
2: **Output:** Posterior mean and variance $\mathbb{E}[\boldsymbol{\beta}]_{D \times 1}, \mathrm{Cov}[\boldsymbol{\beta}]_{D \times D}$.
3: **Initialize:** Feature-label product matrix $\mathbf{P} = \mathbf{0}_{D \times 1}$, precision matrix $\mathbf{S} = \mathbf{I}_{D \times D}$
4: **for** $m = 1$ **to** $M$ **do**
5:     Compute minibatch feature representation $\boldsymbol{\Phi}_m = [\phi(\mathbf{x}_1), \ldots, \phi(\mathbf{x}_{n_m})]_{n_m \times D}$
6:     Update $\mathbf{P} = \mathbf{P} + \Phi_m^\top (\mathbf{y}_m - \Phi_m \boldsymbol{\mu}) / \sigma^2$
7:     Update $\mathbf{S} = \mathbf{S} + \Phi_m^\top \Phi_m / \sigma^2$
8: **end for**
9: Compute $\mathrm{Cov}[\boldsymbol{\beta}] = \boldsymbol{\Sigma}_{\boldsymbol{\beta}} = \mathbf{S}^{-1}$                 ▷ Equation (4)
10: Compute $\mathbb{E}[\boldsymbol{\beta}] = \boldsymbol{\mu} + \boldsymbol{\Sigma}_{\boldsymbol{\beta}} \mathbf{P}$             ▷ Equation (4)

---

As shown, during mini-batch computation, the algorithm computes the posterior mean and precision matrix by linearly accumulating two statistics $\Phi_m^\top (\mathbf{y}_m - \Phi_m \boldsymbol{\mu})$ and $\Phi_m^\top \Phi_m / \sigma^2$, and performs one matrix inversion in the end to obtain the covariance matrix $\boldsymbol{\Sigma}_{\boldsymbol{\beta}}$ (hence even more efficient than the Woodbury update formula introduced in Algorithm 1, which requires an inversion for every single update step). As a result, the space complexity of the algorithm is $O(D^2)$ (for the covariance matrix) and time complexity of the algorithm is $O(nD^2 + D^3)$. Since in practice, the model dimension $D$ is usually fixed and much smaller than $n$, the time complexity is in fact $O(nD^2)$.

### A.4 Additional Algorithm Summary: Joint Learning of Basis Functions and Featurized GP

Notice that given a fixed set of feature functions $\phi = \{\phi_k\}_{k=1}^D$, the GP posterior can be learned scalably and in closed-form via Algorithm 1 (or Algorithm 3). Therefore, when the feature functions $\phi_{\boldsymbol{\theta}}$ are indexed by hyper-parameters $\boldsymbol{\theta}$ (e.g., the bandwidth parameter of an RBF kernel, or the hidden weights of a deep neural network kernel), we can combine Algorithm 1 (or Algorithm 3) with a pre-existing learning method for the hyperparamters to form a coherent procedure that jointly learns the feature functions and the GP posterior.

Concretely, for example, given a hyperparameter learning procedure `update_hyper` that relies on model prediction $(\mathbb{E}[\boldsymbol{\beta}], \mathrm{Cov}[\boldsymbol{\beta}])$, we can consider the below meta-algorithm for alternative inference:

---

**Algorithm 4** Joint learning of feature functions and the GP posterior. Alternative Inference.

---

1: **Input:** Training data mini-batches $\{(\mathbf{X}_m, \mathbf{y}_m)\}_{m=1}^M$. Fixed feature function $\phi_{\boldsymbol{\theta}} : \mathcal{X} \to \mathbb{R}^D$.
2: **Output:** Learned hyperparameter $\hat{\boldsymbol{\theta}}$. Posterior mean and variance $\mathbb{E}[\boldsymbol{\beta}|\hat{\boldsymbol{\theta}}]_{D \times 1}, \mathrm{Cov}[\boldsymbol{\beta}|\hat{\boldsymbol{\theta}}]_{D \times D}$.
3: **Initialize:** Hyperparameter $\boldsymbol{\theta}_0$.
4: **for** iterations $t = 1$ **to** $T$ **do**
5:     Update GP posterior based on $\boldsymbol{\theta}_{t-1}$: $\mathbb{E}[\boldsymbol{\beta}|\boldsymbol{\theta}_{t-1}], \mathrm{Cov}[\boldsymbol{\beta}|\boldsymbol{\theta}_{t-1}]$.    (using Algorithm 1 or 3)
6:     Update hyperparameter $\boldsymbol{\theta}_{t-1} \to \boldsymbol{\theta}_t$ based on GP estimate.    (using `update_hyper`)
7: **end for**
8: Set $\hat{\boldsymbol{\theta}} = \boldsymbol{\theta}_T$.
9: Compute $\mathbb{E}[\boldsymbol{\beta}|\hat{\boldsymbol{\theta}}], \mathrm{Cov}[\boldsymbol{\beta}|\hat{\boldsymbol{\theta}}]$ (using Algorithm 1 or 3).

---

This is essentially the idea behind many classical GP kernel learning algorithm. For example, for RBF kernel, $\boldsymbol{\theta}$ can be the kernel bandwidth parameter, and `update_hyper` is the gradient-based update procedure with respect to marginalized likelihood or leave-one-out cross-validation error (see [Rasmussen and Williams, 2005], Chapter 5.4, where both of these quantities are generally computed by integrating over the model's predictive posterior). Or, Algorithm 4 can be a MCMC procedure for the joint inference of $\boldsymbol{\theta}$ and GP posterior, where `update_hyper` is essentially a Metropolis-Hasting or Hamiltonian Monte Carlo step with respect to the model likelihood[5].

Alternatively, the hyperparameters can be learned using a procedure that does not rely on model posterior $(\mathbb{E}[\boldsymbol{\beta}], \mathrm{Cov}[\boldsymbol{\beta}])$. In this case, we can simply first learn the hyperparameter $\hat{\boldsymbol{\theta}}$ (e.g., for neural network kernel, we can learn the hidden weights of the neural networks using SGD with respect to a

---

[5]In this latter case, the algorithm can be modified to return the full samples of $\{\boldsymbol{\theta}_t\}_{t=1}^T$ and their corresponding conditional posterior samples $\{\mathbb{E}[\boldsymbol{\beta}|\boldsymbol{\theta}_t], \mathrm{Cov}[\boldsymbol{\beta}|\boldsymbol{\theta}_t]\}_{t=1}^T$.

target loss) and then perform posterior inference using Algorithm 1 or 3. Denoting such procedure as `pretrain_hyper`, this leads to the below "pre-training"-style meta-algorithm for joint inference:

---
**Algorithm 5** Joint learning of feature functions and the GP posterior. Pretraining.

---
1: **Input:** Training data mini-batches $\{(\mathbf{X}_m, \mathbf{y}_m)\}_{m=1}^{M}$. Fixed feature function $\phi_{\boldsymbol{\theta}} : \mathcal{X} \to \mathbb{R}^D$.
2: **Output:** Learned hyperparameter $\hat{\boldsymbol{\theta}}$. Posterior mean and variance $\mathbb{E}[\boldsymbol{\beta}|\hat{\boldsymbol{\theta}}]_{D \times 1}, \mathrm{Cov}[\boldsymbol{\beta}|\hat{\boldsymbol{\theta}}]_{D \times D}$.
3: Compute hyperparameter $\hat{\boldsymbol{\theta}}$ using `pretrain_hyper`.
4: Compute $\mathbb{E}[\boldsymbol{\beta}|\hat{\boldsymbol{\theta}}], \mathrm{Cov}[\boldsymbol{\beta}|\hat{\boldsymbol{\theta}}]$ (using Algorithm 1 or 3).

---

This is essentially the core idea behind some of the classical or state-of-the-art neural Gaussian process algorithms (e.g., [Hinton and Salakhutdinov, 2007] or [Liu et al., 2020]), where a Gaussian process model is fit on top of the basis functions (i.e., the hidden features) of a neural network. There, the hidden weights of the neural network can be first learned via regular SGD-based inference with respect to the MAP objective. Then, in the final epoch, the GP posterior is computed conditional on the learned parameters using Algorithm 1 (see, e.g., Algorithm 1 of [Liu et al., 2020]). The similiar idea can be applied to the tree-based models. For example, the partition rules in a decision-tree kernel can be first learned via a conventional tree-learning procedure [Geurts et al., 2006]. Then, a Gaussian process posterior can be fitted to the decision tree by leveraging its featurized representation (Equation (11)).

# B   Featurized Representation of ML Models

The second key advantage of the feature-based representation (3) is its generality: a wide range of machine learning models can be written in term of the feature-based form $f(\mathbf{x}) = \phi(\mathbf{x})^\top \boldsymbol{\beta}$ Rahimi and Recht [2007], Davies and Ghahramani [2014], Lee et al. [2017], making the Gaussian process a unified framework for quantifying model uncertainty with a wide array of modern machine learning models. This section enumerates a few important examples:

**Generalized Additive Models (GAM)**. For a regression task with $d$ input features, a generalized additive model (GAM) has the form $f(\mathbf{x}) = \beta_0 + \sum_{j=1}^d \beta_j h_j(\mathbf{x}^j)$, where $h'_j s$ are flexible functions (e.g., splines) with bounded norm Hastie et al. [2009]. GAM induces a $d$-dimensional feature representation ([Hastie et al., 2009], Chapter 9):

$$\phi(\mathbf{x})_{d \times 1} = [1, h_1(\mathbf{x}^1), \dots, h_d(\mathbf{x}^d)],$$

where $h'_j s$ are usually spline functions that are differentiable. In the special case where all $h'_j s$ are identity functions, GAM reduces to a linear model, and the corresponding $f = \phi(\mathbf{x})^\top \boldsymbol{\beta}$ becomes a GP with linear kernel.

**Decision Trees**. By partitioning the whole feature space into $D$ cells $\mathcal{X} = \cup_{j=1}^D \mathcal{X}_j$, a decision tree model essentially induces a one-hot feature map, e.g.,

$$\phi(\mathbf{x})_{D \times 1} = [0, \dots, 1, \dots, 0],$$

where each element is a indicator function $\mathbb{1}(\mathbf{x} \in \mathcal{X}_j)$ for whether the data point $\mathbf{x}$ falls into the $j^{th}$ cell (Figure 2). This connection is crucial for extending Gaussian process treatment to tree models. Appendix F.1 introduce this formulation in more detail. Following the same construction, the features learned by the majority of partition-based learning methods (e.g., CART, PRIM, etc.) can be used to construct Gaussian process kernels.

**Random Feature Models**. The random-feature model takes the form:

$$\phi(\mathbf{x})_{D \times 1} = \sqrt{2}\sigma(\mathbf{W}^\top \mathbf{x} + \mathbf{b}),$$

where $\mathbf{W}_{d \times D}$ and $\mathbf{b}_{D \times 1}$ are frozen weights initialized from i.i.d. samples from certain fixed distributions, and $\sigma$ is an activation function. For example, in the case of classical random Fourier features whose inner product approximates the RBF kernel, we have $\sigma(\cdot) = \cos(\cdot), \mathbf{W} \stackrel{iid}{\sim} N(0,1), \mathbf{b} \stackrel{iid}{\sim} Unif(0, 2\pi)$ Liu et al. [2021]. Although first introduced as a scalable approximation to GP models equipped with certain kernels (e.g., radial basis function (RBF)), the modern literature treats it as a standalone class of models with its own unique set of theoretic guarantees [Mei and Montanari, 2019, Jacot et al., 2020, Rahimi and Recht, 2009].

**(Deep) Neural Networks**. For a trained $L$-layer neural network of the from $f(\mathbf{x}) = \boldsymbol{\beta}^\top g_L \cdot g_{L-1} \cdots g_0(\mathbf{x})$ with $g_l(\mathbf{x}) = \sigma_l(\mathbf{W}_l^\top \mathbf{x} + \mathbf{b}_l)$, the last-layer representation function

$$\phi(\mathbf{x}) = g_L \cdot g_{L-1} \cdots g_1(\mathbf{x})$$

can be understood as the feature map. Then, the feature map can be used to construct the Gaussian process kernel $k(\mathbf{x}, \mathbf{x}') = \phi(\mathbf{x})^\top \phi(\mathbf{x}')$. This approach was studied extensively in prior literature, due to a neural network's appealing ability in learning an effective representation for the task at hand [Hinton and Salakhutdinov, 2007, Calandra et al., 2016]. Works like [Wilson et al., 2016a,b, Liu et al., 2020] further extended this in the context of modern deep learning.

**Ensembles.** An ensemble model of linear models, trees, or neural networks can be written as a mixture of Gaussian processes. Specifically, an ensemble model can be written as $f(\mathbf{x}) = \sum_{m=1}^M \alpha_m h_m(\mathbf{x})$, where $h'_m s$ are weak learners such as linear models, trees, or neural networks, and $\alpha_m$ are model weights that are either learned or set to uniform $\frac{1}{M}$. This formulation covers well-known examples such as AdaBoost, boosted trees, and random forests Hastie et al. [2009]. As introduced above, since many classical weak learners $h_m = \phi_m(\mathbf{x})^\top \boldsymbol{\beta}_m$ induces a Gaussian process with kernel $k_m$ via their feature representation $k_m(\mathbf{x}, \mathbf{x}') = \phi_m(\mathbf{x}')^\top \phi_m(\mathbf{x})$, the full ensemble model induces a mixture of Gaussian processes with fixed mixing weights dictated by the ensemble weights $\{\alpha\}_{m=1}^M$. That is, the ensemble induces a Bayesian model $f'(\mathbf{x}) = \sum_{m=1}^M \alpha_m h'(\mathbf{x})$ where $\alpha_m$'s are fixed constants and $h'_m(\mathbf{x})$'s are Gaussian process models with prior $\mathcal{GP}(0, k_m)$. In the actual implementation, we

fit each of the individual GP model $h'_m(\mathbf{x})$ following exactly how it is done in the original ensemble model. For example, for random forest models, we fit each $h'_m(\mathbf{x})$ models independently with respect to the original label $y$. While not a focus of this work, for gradient boosting models, we fit $h'_m$'s recursively with respect to the residual $y - \sum_{l<m} \alpha_l h'_l(\mathbf{x})$ [Sigrist, 2020].

## B.1 Summary of model assumptions

This section introduces the theoretical conditions on the featurized model $f(\mathbf{x}) = \phi(\mathbf{x})^\top \boldsymbol{\beta}$ that is required by the proposed framework (i.e., for satisfying Theorems 1 and 2). The intention of this section is to provide a centralized place for readers to verify whether a general model would fit into the framework, and provide pointers to more detailed discussions (e.g., the interpretation of these conditions for important model classes).

First recall that our measure of variable importance $\psi_j(f) = ||D_j f||_n^2$ relies on the differentiation operator $D_j$. For continuous features $x^j \in \mathbb{R}$, the differentiation operator is defined as the conventional partial derivative: $D_j : f \to \frac{\partial}{\partial \mathbf{x}^j} f$, while for discrete features $x^j \in \{0, 1\}$, we overload the derivative operator as the discrete difference $\frac{\partial}{\partial \mathbf{x}^j} f(\mathbf{x}) = f(\mathbf{x}^j = 1, \mathbf{x}^{-j}) - f(\mathbf{x}^j = 0, \mathbf{x}^{-j})$. In such way, the differentiation operator is well-defined for both types of features.

Under the above-defined notion of differentiation operator $D_j$, the proposed framework imposes below three assumptions on the fitted model $f(\mathbf{x}) = \phi(\mathbf{x})^\top \hat{\boldsymbol{\beta}}$ with $\hat{\boldsymbol{\beta}} \in \mathbb{R}^D$:

**1. Weak differentiability.** For the set of causal features $j^* \in \mathcal{A}^*$, the partial derivative $D_{j^*} f(\mathbf{x})$'s exist almost everywhere for $\mathbf{x} \in \mathcal{X}$, so that $||D_{j^*} f(\mathbf{x})||_2^2 > \delta$ for a small positive $\delta > 0$ with non-zero probability $\forall j^* \in \mathcal{A}^*$.

This is a basic condition to ensures the model $f$'s partial derivative is well-defined and can capture the true variable importance $\Phi_{j^*}(f_0) = ||\frac{\partial}{\partial \mathbf{x}^{j^*}} f_0||_2^2$. This condition is weaker than full differentiability, and allows model to have non-differentiable or discontinuous points at discrete locations (e.g., a ReLU network whose gradient is not well-defined at 0). However, we do note that the fully non-differentiable models (e.g., decision trees) can still be incorporated into our framework by applying differentiable approximations (see Appendix F). Remark 2 in Appendix C) provides additional relevant discussion of this condition.

**2. Lipschitz condition.** The model $f$ should be Lipschitz with a bounded Lipschitz constant $C < \infty$, so that $\frac{|f(\mathbf{x}_1) - f(\mathbf{x}_2)|}{||\mathbf{x}_1 - \mathbf{x}_2||_2} \leq C$ for all pairs $(\mathbf{x}_1, \mathbf{x}_2) \in \mathcal{X} \times \mathcal{X}$. Here $||\mathbf{x}_1 - \mathbf{x}_2||_2$ is the $L_2$ metric in $\mathcal{X}$.

The Lipschitz condition is a mild requirement indicating that the model gradient is well-behaved (i.e., bounded away from infinity). It is essential for the posterior concentration of the variable importance metric, and is used to satisfy condition (2) of Theorem 1 (Section 3.2). Appendix E.1 discusses the Lipschitz condition of important model classes, including Generalized additive models, deep neural networks, and decision trees under differentiable approximation.

**3. Growth condition on model rank ($o_p(\sqrt{n})$).** Given the feature map $\phi : \mathcal{X} \in \mathbb{R}^D$ and the data-generating distribution $P(\mathbf{x})$, the rank of the model space $\mathcal{H}$ can be empirically measured as $\lim_{n\to\infty} rank(\Phi)$, where $\Phi_{n\times D} = [\phi^\top(\mathbf{x}_1), \ldots, \phi^\top(\mathbf{x}_n)]^\top$ is the feature matrix evaluated at $\mathbf{x}_i \sim P(\mathbf{x})$.

To this end, we require the growth condition of the model rank to not increase faster than $o_p(\sqrt{n})$, so that the model complexity is well-controlled. This condition is essential for the BvM phenomenon, and is used to satisfy condition (4) of Theorem 2 (see Section 3.2 for related discussion). This condition is easily satisfied by ML models in practice. For example, since the model rank is upper bounded by the dimension of the feature map $D$, models with fixed feature dimension trivially satisfies this condition. This growth condition can also be satisfied by adaptive-rank models such random forest or sparse neural networks, and are in fact much less stringent than what is required by the BvM theorems in the existing literature. See Appendix E.2 for detailed discussion.

## C   Proof for Posterior Convergence

**Proof for Theorem 1**

Recall the list of technical conditions:

i) **(Convergence of Prediction Function $f$)** The posterior distribution $\Pi_n(f)$ converges toward $f_0$ at a rate of $\epsilon_n$. (Note that in nonparametric learning setting, this rate is not faster than $O_p(n^{-\frac{1}{2}})$ which is the optimal parametric rate);

ii) **(Well-conditioned Derivative Functions)** $D_j : f \to \frac{\partial}{\partial \mathbf{x}^j} f$ the differentiation operator is bounded: $\|D_j\|_{op}^2 = \inf\{C \geq 0 : \|D_j f\|_2^2 \leq C\|f\|_2^2, \text{ for all } f \in \mathcal{H}_\phi\}$;

*Proof.* Denote $A_n = \{f : \|f - f_0\|_n^2 > M_n \epsilon_n\}$ and $B_n = \{f : |\psi_j(f) - \Psi_j(f_0)| > M_n \epsilon_n\}$, then showing the statement in Theorem 1 is equivalent to showing $\Pi_n(B_n) \to 0$.

Specifically, we assume below two facts hold:

**Fact 1.** $|\psi_j(f) - \psi_j(f_0)| \leq \|D_j f - D_j f_0\|_n^2$

**Fact 2.** $\sup_{j \in \{1,\ldots,d\}} |\psi_j(f_0) - \Psi_j(f_0)| \lesssim \|f - f_0\|_n^2$

Because if the above facts hold, we then have

$$
\begin{aligned}
\sup_{j \in \{1,\ldots,d\}} |\psi_j(f) - \Psi_j(f_0)| &\leq \sup_{j \in \{1,\ldots,d\}} |\psi_j(f) - \psi_j(f_0)| + \sup_{j \in \{1,\ldots,d\}} |\psi_j(f_0) - \Psi_j(f_0)| \\
&\leq \sup_{j \in \{1,\ldots,d\}} \|D_j f - D_j f_0\|_n^2 + \sup_{j \in \{1,\ldots,d\}} |\psi_j(f_0) - \Psi_j(f_0)| \\
&\leq \sup_{j \in \{1,\ldots,d\}} \|D_j f - D_j f_0\|_2^2 + O_p(n^{-\frac{1}{2}}) + \sup_{j \in \{1,\ldots,d\}} |\psi_j(f_0) - \Psi_j(f_0)| \\
&\leq C\|f - f_0\|_2^2 + O_p(n^{-\frac{1}{2}}) + \sup_{j \in \{1,\ldots,d\}} |\psi_j(f_0) - \Psi_j(f_0)| \\
&\qquad\qquad (D_j \text{ is bounded}) \\
&\leq C\|f - f_0\|_n^2 + O_p(n^{-\frac{1}{2}}) + \sup_{j \in \{1,\ldots,d\}} |\psi_j(f_0) - \Psi_j(f_0)| \\
&\lesssim \|f - f_0\|_n^2.
\end{aligned}
$$

It then follows that:

$$
\mathbb{E}_0 \Pi_n \Big( \sup_{j \in \{1,\ldots,d\}} |\psi_j(f) - \Psi_j(f_0)| \geq M_n \epsilon_n \Big) \lesssim \mathbb{E}_0 \Pi_n \Big( \|f - f_0\|_n^2 \geq M_n' \epsilon_n \Big) \to 0.
$$

We now show Facts 1 and 2 are true.

- **Fact 1** follows simply from the triangular inequality:

$$
\begin{aligned}
|\psi_j(f) - \psi_j(f_0)| &= \Big| \|D_j f\|_n^2 - \|D_j f_0\|_n^2 \Big| \\
&= \max\Big\{ \|D_j f\|_n^2 - \|D_j f_0\|_n^2, \ \|D_j f_0\|_n^2 - \|D_j f\|_n^2 \Big\} \leq \|D_j f - D_j f_0\|_n^2.
\end{aligned}
$$

- **Fact 2** follows from standard Bernstein-type concentration inequality (see, e.g., Lemma 18 of Rosasco et al. [2013]). Specifically, for $|D_j f_0(\mathbf{x})|^2$ a random variable with respect to probability measure $P(\mathbf{x})$ that is bounded by $L$. Given $n$ iid samples $\{|D_j f_0(\mathbf{x}_i)|^2\}_{i=1}^n$, recall that $\psi_j(f_0) = \frac{1}{n} \sum_{i=1}^n |D_j f_0(\mathbf{x}_i)|^2$ and $\Psi(f_0) = \mathbb{E}(|D_j f_0|^2)$, then with probability $1 - \eta$:

$$
|\psi_j(f_0) - \Psi(f_0)| \leq n^{-\frac{1}{2}} * \big( 2\sqrt{2} * L * \log(2/\eta) \big),
$$

that is, $|\psi_j(f_0) - \Psi(f_0)| \to 0$ at the rate of $O(n^{-\frac{1}{2}})$. Notice that $O(n^{-\frac{1}{2}})$ is the optimal parametric rate that cannot be surpassed by the convergence speed of the ReLU networks

(recall the typical convergence rate is $\epsilon_n \asymp n^{-\frac{\beta}{2\beta+\delta}} * \log(n)^\gamma$ for some $\delta > 0$ and $\gamma > 1$). Therefore we have:

$$\sup_{j \in \{1,\ldots,d\}} |\psi_j(f_0) - \Psi_j(f_0)| \lesssim \|f - f_0\|_n^2.$$

$\square$

*Remark* 1 (**Convergence of $L_2$ norm**). The sample $L_2$ norm and the expected $L_2$ norm are closed to each other at the rate of $O(n^{-\frac{1}{2}})$. This will happen when $\mathbf{x}$'s are sampled randomly from a probability measure $P(\mathbf{x})$.

*Remark* 2 (**A condition on weak differentiabilty**). Although not listed explicitly in the main theorem, we also impose a weak technical condition (i.e., Non-trivial Gradient Function) on model function $f$ and true function $f_0$ to avoid certain pathological situations:

   iii) (**Non-trivial Derivative Functions**) Denote $j^* \in \{1, \ldots, d^*\}$ the index of the causal variables, and recall $P_{\mathcal{X}}(\mathbf{x})$ the distribution of the input features $\mathbf{x}$. Then there exists $\delta > 0$ such that for all $j^* \in \{1, \ldots, d^*\}$, $\|D_{j^*} f_0(\mathbf{x})\|_2^2 > \delta$ and $\|D_{j^*} f(\mathbf{x})\|_2^2 > \delta$ with non-zero probability.

Note that this condition is weak in that it only requires the partial derivative under model function $f$ and $f_0$ are not zero almost everywhere. For differentiable functions under continuous features, this should be satisfied by definition. This basic technical condition is intended to remove two pathological situations. The first is non-differentiable models (e.g., tree models), whose gradient is zero almost everywhere in the feature space. The second case are the discrete features, where the traditional sense of partial derivative is not well defined. In Appendix F, we discuss how to incorporate non-differentiable models and discrete features into our framework. Briefly, a non-differentiable model (e.g., partition-based models) can be made differentiable by employing a differentable approximation. For discrete features, we can compute the discrete version of the differentiable operator, e.g., $D_j f(\mathbf{x}) = f(\mathbf{x}^j = 1, \mathbf{x}^{-j}) - f(\mathbf{x}^j = 0, \mathbf{x}^{-j})$ for binary feature where $\mathbf{x}_{d \times 1} = [\mathbf{x}^j, [\mathbf{x}^{-j}]_{(d-1) \times 1}^\top]^\top$ (known as *contrast* in statistics). Notice that this discrete differentiation operator $D_j f(\mathbf{x})$ is a linear function of the original prediction function $f$. As a result, the posterior convergence of $\psi_j$ with respect to this operator is again guaranteed by the convergence of the prediction $f$.

*Remark* 3 (**Convergence guanrantee of $f$**). Note that our result focuses on posterior concentration of variable importance $\psi_j$, not of prediction function $f$. In fact, the convergence of $\psi_j$ depends on the convergence of the prediction function $f$, as introduced in the assumptions of 1. In practice, it is up to the practitioners to select a proper prediction model $f$ that has a convergence guarantee for the task at hand. Specifically, we showed that for any model, if its prediction function has a posterior concentration guarantee, its variable importance has a convergence guarantee as well. To this end, we notice that majority of popular machine learning methods (e.g., random features, neural networks, tree ensembles) has a posterior concentration guarantee for target functions in certain general function space (e.g., the space of $\alpha$-Hölder space), given the recent advances in the approximation and convergence guarantees of parametric (finite-dimensional) ML models in both frequentist and Bayesian settings Ročková and van der Pas [2020], Wang and Rocková [2020], Liu [2021], Schmidt-Hieber [2020].

Furthermore, we note that although the ML models covered in our work are not traditional universal kernels Micchelli et al. [2006], most of them (e.g., random features, neural networks, tree ensembles) do come with a universal approximation guarantee for an appropriately defined function class Rahimi and Recht [2008], Biau [2012], Schmidt-Hieber [2020]. As a result, the kernel functions defined by these models provide basis functions that span function spaces that are often dense in an infinite-dimensional RKHS, implying that the resulting model can approximate $f_0$ to arbitrary precision Rahimi and Recht [2008]. Please see Rahimi and Recht [2008], Hornik et al. [1989], Biau [2012] for specific results for random features, neural networks and random forests.

*Remark* 4 (**Compatibility with high-dimensional settings**). Although in the main text, we assumed the data dimension $d$ is fixed with respect to $n$ (see Section 2, Problem Setup). Theorem 1 in fact does not rely on a fixed $d$, and the proof can go through whenever $d = o(n)$. That is to say, the posterior concentration of variable importance measure $\psi_j(f)$ can occur even in the high-dimensional settings of $d$, which is allowed to grow with sample size $n$. In comparison, the *Bernstein-von Mises*

(BvM) result in Theorem 2 implies a stronger sense of convergence (i.e., convergence in distribution) and requires more stringent conditions (i.e., fixed data dimension $d$). This is in fact consistent with modern BvM analysis of Bayesian ML models, where a fixed data dimension is commonly assumed [Wang and Rocková, 2020, Rockova, 2020, Yang et al., 2015, Burnaev et al., 2013, Castillo and Rousseau, 2015].

## C.1 A finite-sample error bound

Although Theorem 1 is stated as an asymptotic result, we note that if the error bound for the prediction function $\epsilon_n$ (Condition 1) is a finite-sample error bound, then a finite-sample error bound for the variable importance $\psi_j(f)$ can be trivially derived by extending the proof of Theorem 1. For example, we can have the below finite-sample concentration result:

**Proposition 3** (Finite-sample Error Bound for $\psi_j(f)$). *Assume*

    *i) (**Finite-sample Error Bound for Prediction Function** $f$) The posterior distribution of $f$ on average converges toward $f_0$ at a finite-sample rate of $\epsilon_n$, such that $\mathbb{E}_0 E_n \|f - f_0\|_n^2 \leq \epsilon_n$ and $\epsilon_n$ is finite-sample error bound that is an explicit function of $n$, and $E_n$ is the expectation with respect to the posterior distribution $\Pi_n$.*

    *ii) (**Well-conditioned Derivative Functions**) $D_j : f \to \frac{\partial}{\partial \mathbf{x}^j} f$ the differentiation operator is bounded: $\|D_j\|_{op}^2 = \inf\{C \geq 0 : \|D_j f\|_2^2 \leq C\|f\|_2^2, \text{ for all } f \in \mathcal{H}_\phi\};$*

*Also assumes the data $\mathbf{x} \sim P(\mathbf{x})$ is generated with respect to probability measure $P(\mathbf{x})$ that is bounded by $L$. Then, with probability $1 - \eta$, the posterior distribution for $\psi_j(f) = \|\frac{\partial}{\partial \mathbf{x}^j} f\|_n^2$ contracts toward $\Psi_j(f_0) = \|\frac{\partial}{\partial \mathbf{x}^j} f_0\|_{P_\mathcal{X}}^2$ at a finite-sample rate $\epsilon_n'$ such that:*

$$\mathbb{E}_0 E_n \left( \sup_{j \in \{1,\dots,d\}} |\psi_j(f) - \Psi_j(f_0)| \right) \leq \epsilon_n'$$

*where*

$$\epsilon_n' = C * \epsilon_n + n^{-\frac{1}{2}} * \left( 2\sqrt{2} * L * \log(2/\eta) \right).$$

*Proof.* Recall in **Fact 2** of the proof of Theorem 1. By Bernstein inequality, we have, with probability $1 - \eta$:

$$|\psi_j(f_0) - \Psi(f_0)| \leq n^{-\frac{1}{2}} * \left( 2\sqrt{2} * L * \log(2/\eta) \right).$$

Then, following the same line of argument as the proof of Theorem 1, we have:

$$
\begin{aligned}
\sup_{j \in \{1,\dots,d\}} |\psi_j(f) - \Psi_j(f_0)| &\leq \sup_{j \in \{1,\dots,d\}} |\psi_j(f) - \psi_j(f_0)| + \sup_{j \in \{1,\dots,d\}} |\psi_j(f_0) - \Psi_j(f_0)| \\
&\leq \sup_{j \in \{1,\dots,d\}} \|D_j f - D_j f_0\|_n^2 + \sup_{j \in \{1,\dots,d\}} |\psi_j(f_0) - \Psi_j(f_0)| \\
&\leq C\|f - f_0\|_n^2 + \sup_{j \in \{1,\dots,d\}} |\psi_j(f_0) - \Psi_j(f_0)|. \\
&\leq C\|f - f_0\|_n^2 + n^{-\frac{1}{2}} * \left( 2\sqrt{2} * L * \log(2/\eta) \right).
\end{aligned}
$$

Now, take expectation $\mathbb{E}_0 E_n$ for both sides, we arrive at:

$$
\begin{aligned}
\mathbb{E}_0 E_n \left( \sup_{j \in \{1,\dots,d\}} |\psi_j(f) - \Psi_j(f_0)| \right) &\leq C\mathbb{E}_0 E_n \|f - f_0\|_n^2 + n^{-\frac{1}{2}} * \left( 2\sqrt{2} * L * \log(2/\eta) \right) \\
&= C\epsilon_n + n^{-\frac{1}{2}} * \left( 2\sqrt{2} * L * \log(2/\eta) \right).
\end{aligned}
$$

$\square$

# D Proofs for Asymptotic Normality

**Lemma 4.** *Functional Delta Method (univariate) Suppose $\mathcal{P}_n$ is the empirical distribution of a random sample $X_1, \ldots, X_n$ from a distribution $P$, and $\phi$ is a function that maps the distribution of interest into some space. Define the Gateaux derivative*

$$\phi'_P(\delta_x - P) = \frac{\mathrm{d}}{\mathrm{d}t} \mid_{t=0} \phi((1-t)P + t\delta_x) = IF_{\phi,P}(x),$$

*which is also the Influence Function, and $\gamma^2 = \int IF_{\phi,P}(x)^2 dP$. If integration and differentiation can be exchanged, then*

$$\int \phi'_P(\delta_x - P)dP = 0.$$

*Further, if $\sqrt{n}R_n \xrightarrow{P} 0$, where*

$$R_n = \phi(\mathcal{P}_n) - \phi(P) - \frac{1}{n}\sum_i \phi'_P(\delta_{x_i} - P),$$

*then from the Central Limit Theory that*

$$\sqrt{n}(\phi(\mathcal{P}_n) - \phi(P)) \xrightarrow{d} \mathcal{N}(0, \gamma^2).$$

**Lemma 5.** *Functional Delta Method (multivariate) Suppose $\mathcal{P}_n$ is the empirical distribution of a random sample $X_1, \ldots, X_n$ from a distribution $P$, and $\boldsymbol{\phi} : \mathbb{R}^d \to \mathbb{R}^k$. Define the Gateaux derivative*

$$\boldsymbol{\phi}'_P(\delta_x - P) = \frac{\mathrm{d}}{\mathrm{d}t} \mid_{t=0} \boldsymbol{\phi}((1-t)P + t\delta_x) = IF_{\boldsymbol{\phi},P}(x),$$

*which is also the Influence Function, and $[\mathbf{V}_0]_{i,j} = \int \langle [IF_{\boldsymbol{\phi},P}(x)]_i, [IF_{\boldsymbol{\phi},P}(x)]_j \rangle dP$. If integration and differentiation can be exchanged, then*

$$\int \boldsymbol{\phi}'_P(\delta_x - P)dP = 0.$$

*Further, if $\sqrt{n}\mathbf{R}_n \xrightarrow{P} 0$, where*

$$\mathbf{R}_n = \boldsymbol{\phi}(\mathcal{P}_n) - \boldsymbol{\phi}(P) - \frac{1}{n}\sum_i \boldsymbol{\phi}'_P(\delta_{x_i} - P),$$

*then from the Central Limit Theory that*

$$\sqrt{n}(\boldsymbol{\phi}(\mathcal{P}_n) - \boldsymbol{\phi}(P)) \xrightarrow{d} \mathcal{MVN}(0, \mathbf{V}_0).$$

**Proof for Theorem 2**

To make our assumptions explicit, we list out a collection of easily-satisfied technical conditions.

(1) $f$ is a consistent estimator of $f_0$;

(2) $D_j$ is bounded: $\|D_j\|_{op}^2 = \inf\{C \geq 0 : \|D_j f\|_2^2 \leq C\|f\|_2^2, \text{ for all } f \in \mathcal{H}_\phi\}$;

(3) $f_0$ is square-integrable over the support of $X$ and $\|f_0\|_2 = 1$;

(4) $\mathtt{rank}(H_j) = o_p(\sqrt{n})$;

*Proof.* Since $H_j = D_j^\top D_j$, Condition (2) is equivalent to the largest eigenvalue of $H_j$ being bounded, i.e., $\lambda_{max}(H_j) = O_p(1)$. From the definition in Equation (10), we have

$$\psi'_j(f) = \frac{\partial}{\partial f}\psi_j(f) = \frac{2}{n}H_j f.$$

Define a mean functional $m : F \to E(F)$, where $F$ is the distribution. Then in our case, $f_0 = E(F) = m(F)$. According to Lemma 4, we have

$$\psi_j(f_0) = \psi_j(E(F)) = \psi_j(m(F)) = \phi(F),$$

i.e., $\phi(\cdot) = \psi_j(m(\cdot))$. Therefore,

$$
\begin{aligned}
\phi'_F(\delta_y - F) &= \psi'_j(m(\delta_y - F)) \\
&= \frac{\mathrm{d}}{\mathrm{d}t}\big|_{t=0}\, \psi_j(m((1-t)F + t\delta_y)) \\
&= \frac{\mathrm{d}}{\mathrm{d}t}\big|_{t=0}\, \psi_j((1-t)f_0 + ty) \\
&= \frac{\mathrm{d}}{\mathrm{d}t}\big|_{t=0}\, \frac{1}{n}[(1-t)f_0 + ty]^\top H_j[(1-t)f_0 + ty] \\
&= \frac{2}{n}(y - f_0)^\top H_j f_0 \\
&= IF_{\phi,F}(y).
\end{aligned}
$$

On the other hand,

$$
\begin{aligned}
\gamma^2 &= \int IF_{\phi,F}(y)^2 dF \\
&= 4\int \frac{1}{n} \cdot f_0^\top H_j(y - f_0)(y - f_0)^\top H_j f_0 \cdot \frac{1}{n} dF \\
&= 4\sigma^2 \|H_j f_0\|_n^2.
\end{aligned}
$$

Moreover, we have

$$\int \phi'_F(\delta_y - F)dF = \frac{2}{n}\int (y - f_0)^\top H_j f_0 dF = 0,$$

and

$$
\begin{aligned}
\sqrt{n}R_n &= \sqrt{n}[\phi(\mathcal{F}_n) - \phi(F) - \frac{1}{n}\sum_i \phi'_F(\delta_{y_i} - F)] \\
&= \sqrt{n}[\psi_j(f) - \psi_j(f_0) - \frac{1}{n} \cdot \frac{2}{n}\sum_i (y_i - f_{0,i})^\top[H_j f_0]_i] \\
&= \sqrt{n}[\frac{1}{n} \cdot (f^\top H_j f - f_0^\top H_j f_0) - \frac{1}{n} \cdot \frac{2}{n}(y - f_0)^\top H_j f_0] \\
&= \frac{1}{\sqrt{n}}[f^\top H_j f - f^\top H_j f_0 + f^\top H_j f_0 - f_0^\top H_j f_0 - \frac{2}{n}(y - f_0)^\top H_j f_0] \\
&= \frac{1}{\sqrt{n}}[(f - f_0)^\top H_j(f + f_0) - \frac{2}{n}(y - f_0)^\top H_j f_0] \\
&= \frac{1}{\sqrt{n}}[(f - f_0)^\top H_j(f - f_0) + 2(f - f_0)^\top H_j f_0 - \frac{2}{n}(y - f_0)^\top H_j f_0] \\
&= \frac{1}{\sqrt{n}}[\hat{\epsilon}_n^\top H_j \hat{\epsilon}_n + 2\hat{\epsilon}_n^\top H_j f_0 - \frac{2}{n}(y - f_0)^\top H_j f_0] \quad\quad (12) \\
&= \frac{1}{\sqrt{n}}o_p(\sqrt{n}) \quad\quad (13) \\
&= o_p(1) \xrightarrow{P} 0,
\end{aligned}
$$

where $\hat{\epsilon}_n = f - f_0$. We can prove the result from Equation (12) to Equation (13) as following: Denote $k = \mathtt{rank}(H_j)$, then the eigendecomposition of $H_j$ is $H_j = U_j \Lambda U_j^\top$, with $U_j = [\mathbf{u}_1, \ldots, \mathbf{u}_k]$ a $n \times k$ orthogonal matrix and $\Lambda$ a $k \times k$ diagonal matrix with elements $\{\lambda_i\}_{i=1}^k$ being the eigenvalues of $H_j$, then define

$$\mathbf{v} = U_j^\top \hat{\epsilon}_n = \begin{bmatrix} \mathbf{u}_1^\top \hat{\epsilon}_n \\ \vdots \\ \mathbf{u}_k^\top \hat{\epsilon}_n \end{bmatrix}.$$

Therefore,

$$
\begin{aligned}
\hat{\epsilon}_n^\top H_j \hat{\epsilon}_n = \mathbf{v}^\top \Lambda \mathbf{v} &= \sum_{i=1}^{k} \lambda_i v_i^2 \\
&\leq \lambda_{max}(H_j) \sum_{i=1}^{k} v_i^2 \\
&= \lambda_{max}(H_j) \sum_{i=1}^{k} \mathbf{u}_i^\top \hat{\Sigma}_n \mathbf{u}_i \\
&\leq \lambda_{max}(H_j) \sum_{i=1}^{k} \lambda_{max}(\hat{\Sigma}_n) \\
&= k \cdot \lambda_{max}(H_j) \cdot \lambda_{max}(\hat{\Sigma}_n) \\
&= o_p(\sqrt{n}) \cdot O_p(1) \cdot O_p(1) \\
&= o_p(\sqrt{n}),
\end{aligned}
$$

where $E(\hat{\epsilon}_n) = \mathbf{0}$, $\mathrm{cov}(\hat{\epsilon}_n) = \hat{\Sigma}_n$, and $\lambda_{max}(\hat{\Sigma}_n)$ is the largest eigenvalue of $\hat{\Sigma}_n$.

On the other hand, $2\hat{\epsilon}_n^\top H_j f_0 = o_p(\sqrt{n})$ because $f$ is a consistent estimator of $f_0$. Moreover, since $y_i - f_{0,i} = O_p(1)$, we know $\frac{2}{n}(y - f_0)^\top H_j f_0 = o_p(\sqrt{n})$. So,

$$
\hat{\epsilon}_n^\top H_j \hat{\epsilon}_n + 2\hat{\epsilon}_n^\top H_j f_0 - \frac{2}{n}(y - f_0)^\top H_j f_0 = o_p(\sqrt{n}) \tag{14}
$$

Therefore, by Lemma 4, we have

$$
\sqrt{n}(\psi_j(f) - \psi_j(f_0)) \xrightarrow{d} \mathcal{N}(0, 4\sigma^2 \|H_j f_0\|_n^2).
$$

$\square$

**Theorem 6** (Asymptotic Distribution of Variable Importance (multivariate))**.** *Suppose* $y_i = f_0(\mathbf{x}_i) + e_i$, $e_i \overset{i.i.d.}{\sim} \mathcal{N}(0, \sigma^2)$, $i = 1, \ldots, n$. *Denote* $\boldsymbol{\psi} = [\psi_1, \ldots, \psi_d]$ *for* $\psi_j$ *as defined in Equation* (10). *If the following conditions are satisfied:*

   *i)* `rank`$(H_j) = o_p(\sqrt{n}), j = 1, \ldots, d$;

   *ii)* $f_0$ *is square-integrable over the support of* $X$ *and* $\|f_0\|_2 = 1$;

   *iii)* $f$ *is a consistent estimator of* $f_0$;

   *iv)* $D_j$ *is bounded:* $\|D_j\|_{op}^2 = \inf\{C \geq 0 : \|D_j f\|_2^2 \leq C\|f\|_2^2$, *for all* $f \in \mathcal{H}_\phi\}$.

*Then* $\boldsymbol{\psi}(f)$ *asymptotically converges toward a multivariate normal distribution surrounding* $\boldsymbol{\psi}(f_0)$, *i.e.,*

$$
\sqrt{n}(\boldsymbol{\psi}(f) - \boldsymbol{\psi}(f_0)) \xrightarrow{d} \mathcal{MVN}(\mathbf{0}, \mathbf{V}_0),
$$

*where* $\mathbf{V}_0$ *is a* $d \times d$ *matrix such that* $[\mathbf{V}_0]_{j1,j2} = 4\sigma^2 \langle H_{j1} f_0, H_{j2} f_0 \rangle_n$.

*Proof.* Define a mean function $m : F \to E(F)$, where $F$ is the distribution. Then in our case, $f_0 = E(F) = m(F)$. According to Lemma 5, we have

$$
[\boldsymbol{\psi}(f_0)]_j = \psi_j(E(F)) = \psi_j(m(F)) = [\boldsymbol{\phi}(F)]_j,
$$

i.e., $\phi(\cdot) = \psi(m(\cdot))$ and $[\phi(\cdot)]_j = \psi_j(m(\cdot))$, where $\phi : \mathcal{R} \to \mathcal{R}^P$. Therefore,

$$
\begin{aligned}
[\phi'_F(\delta_y - F)]_j &= \psi'_j(m(\delta_y - F)) \\
&= \frac{\mathrm{d}}{\mathrm{d}t}|_{t=0}\, \psi_j(m((1-t)F + t\delta_y)) \\
&= \frac{\mathrm{d}}{\mathrm{d}t}|_{t=0}\, \psi_j((1-t)f_0 + ty) \\
&= \frac{\mathrm{d}}{\mathrm{d}t}|_{t=0}\, \frac{1}{n}[(1-t)f_0 + ty]^\top H_j[(1-t)f_0 + ty] \\
&= \frac{2}{n}(y - f_0)^\top H_j f_0 \\
&= [IF_{\phi,F}(y)]_j.
\end{aligned}
$$

On the other hand,

$$
\begin{aligned}
[\mathbf{V}_0]_{j1,j2} &= \int \langle [IF_{\phi,F}(y)]_{j1}, [IF_{\phi,F}(y)]_{j2}\rangle dF \\
&= 4\int \frac{1}{n}\cdot f_0^\top H_{j1}(y-f_0)(y-f_0)^\top H_{j2}f_0 \cdot \frac{1}{n}dF \\
&= 4\sigma^2\langle H_{j1}f_0, H_{j2}f_0\rangle_n.
\end{aligned}
$$

Moreover, we have

$$
[\int \phi'_F(\delta_y - F)dF]_j = \frac{2}{n}\int (y-f_0)^\top H_j f_0 dF = 0,
$$

and

$$
\begin{aligned}
[\sqrt{n}\mathbf{R}_n]_j &= \sqrt{n}[[\phi(\mathcal{F}_n)]_j - [\phi(F)]_j - \frac{1}{n}\sum_i[\phi'_F(\delta_{y_i} - F)]_j] \\
&= \frac{1}{\sqrt{n}}[f^\top H_j f - f_0^\top H_j f_0 - \frac{2}{n}(y-f_0)^\top H_j f_0] \\
&= \frac{1}{\sqrt{n}}[(f-f_0)^\top H_j(f-f_0) + 2(f-f_0)^\top H_j f_0 - \frac{2}{n}(y-f_0)^\top H_j f_0] \\
&= \frac{1}{\sqrt{n}}[\hat{\epsilon}_n^\top H_j \hat{\epsilon}_n + 2\hat{\epsilon}_n^\top H_j f_0 - \frac{2}{n}(y-f_0)^\top H_j f_0] \qquad (15) \\
&= \frac{1}{\sqrt{n}}o_p(\sqrt{n}) \qquad\qquad\qquad\qquad\qquad\qquad\qquad\quad (16) \\
&= o_p(1) \xrightarrow{P} 0,
\end{aligned}
$$

where $\hat{\epsilon}_n = f - f_0$ and the reason from Equation (15) to Equation (16) is because of Equation (14). Therefore, by Lemma 5, we have

$$
\sqrt{n}(\psi(f) - \psi(f_0)) \xrightarrow{d} \mathcal{MVN}(\mathbf{0}, \mathbf{V}_0),
$$

where $\mathbf{V}_0$ is a $d \times d$ matrix such that $[\mathbf{V}_0]_{j1,j2} = 4\sigma^2\langle H_{j1}f_0, H_{j2}f_0\rangle_n$. $\qquad\square$

# E   Additional Theoretical Discussions

## E.1   Lipschitz condition of ML models

The condition of the differentiation operator $D_j : f \to \frac{\partial}{\partial \mathbf{x}^j} f$ being bounded is guaranteed if $f$ is differentiable and Lipschitz, so that $\frac{|f(\mathbf{x}_1) - f(\mathbf{x}_2)|}{||\mathbf{x}_1 - \mathbf{x}_2||_2} \leq C$ where $||\mathbf{x}_1 - \mathbf{x}_2||_2$ is the $L_2$ metric in $\mathcal{X}$. Fortunately, a wide range of machine learning models (under proper regularity condition) satisfy the Lipschitz condition. Below we consider a few important examples:

**Generalized Additive Models (GAM).** The generalized additive models is often written as the sum of smooth functions,

$$f(\mathbf{x}) = \beta_0 + \sum_{j=1}^{d} \boldsymbol{\beta}_j h_j(\mathbf{x}^j).$$

As a result, $f$ is Lipschitz if every individual smooth function $h_j$ is Lipschitz. To this end, we notice that in the GAM algorithm, the $h_j$'s are commonly estimated under a smoothness constraint in terms of its second derivatives [Wood, 2006] $\psi_{2,j} = \int_{\mathcal{X}} |\frac{\partial^2}{\partial (\mathbf{x}^j)^2} f(\mathbf{x}^j)|^2 d\mathbf{x}$, which essentially imposes an upper bound on the first-order partial derivatives $\frac{\partial}{\partial \mathbf{x}^j} f(\mathbf{x}^j)$ (assuming bounded support). As a result, the Lipschitz of GAM function is guanranteed by the virtue of its smoothing constraints.

**Decision Trees**. Interestingly, we can understand the Lipschitz condition of a tree-type model by investigating its model structure from a neural network lens. Specifically, for a depth-$L$ tree model with $D$ leaf nodes, Karthikeyan et al. [2021] shows that a it can be written in the form of a neural network layer:

$$f(\mathbf{x}) = \sum_{k=1}^{D} q_k(\mathbf{x}) \boldsymbol{\beta}_k, \quad \text{where} \quad q_k(\mathbf{x}) = \sigma_{\text{step}} \Big( \sum_{l=1}^{L} \sigma_{\text{step}} \big( (\mathbf{x}^\top \mathbf{w}_{k,I(l,k)} + b_{k,I(l,k)}) S(l,k) \big) - h \Big).$$

Here $q_k(\mathbf{x})$ is a re-parametrization for the indicator function of whether $\mathbf{x}$ belongs to the $k^{th}$ leaf node, i.e., $\prod_{l=1}^{L} \sigma_{\text{step}} \Big[ (\mathbf{x}^\top \mathbf{w}_{k,I(l,k)} + b_{k,I(l,k)}) S(l,k) \Big]$. (See Appendix A.1 or Section 3 of Karthikeyan et al. [2021] for full detail.) Briefly, $\sigma_{\text{step}}(x) = I(x > 0)$ is the step function, $I(l,k)$ indicates the index for the ancestor node for the $k^{th}$ leaf at depth $l$, and $S(l,k) \in \{-1, 1\}$ is a sign function for whether $k^{th}$ leaf is the right subtree of node $I(l,k)$. As a result, $q_k(\mathbf{x})$ measures whether $\mathbf{x}$ satisfies every ancestry decision rules $I\big[ S(l,k)(\mathbf{x}^\top \mathbf{w}_{k,I(l,k)} - b_{k,I(l,k)}) > 0 \big]$ at every level $l \in \{1, \ldots, L\}$, where $\mathbf{w}_{k,I(l,k)}$ is a $d \times 1$ one-hot vector indicating the index of feature being selected by that node.

As a result, the tree model can be viewed as a wide 1-hidden layer neural network model with bounded activation function $\sigma_{\text{step}}$ and hidden weights bounded within $[-1, 1]$, which leads to a Lipschitz function. Furthermore, the function $f(\mathbf{x})$ remains Lipschitz if we replace the non-differentiable $\sigma_{\text{step}}$ with a differentiable activation function that is Lipschitz (e.g., Appendix F).

**Random Feature Models**. The random feature methods are also structured the same way as $f(\mathbf{x}) = \sigma(\mathbf{W}^\top \mathbf{x} + \mathbf{b})$, where $\mathbf{W}$ are frozen weights that are independently sampled from distribution with finite second moments (e.g., Gaussian distribution), and $\sigma$ is a trignomitric function ($sin$ and $cos$), or common activation functions that are used in the neural networks [Choromanski et al., 2018, Liu et al., 2021]. As a result, $f(\mathbf{x})$ is also Lipschitz with high probability. In practice, the Lipschitz condition can be guaranteed in absolute terms by truncating the individual terms in $\mathbf{W}$ to be within a range $[-C, C]$ (e.g., $C = 4$. for $W \overset{iid}{\sim} N(0, 1)$), which often leads to almost identical performance.

**(Deep) Neural Networks**. Both deep neural networks and random-feature models can be written as a composition of functions:

$$f(\mathbf{x}) = \boldsymbol{\beta}^\top g_L \cdot g_{L-1} \cdots g_1(\mathbf{x}), \quad \text{where} \quad g_l(\mathbf{x}) = \sigma(\mathbf{W}_l^\top \mathbf{x} + \mathbf{b}_l).$$

As a result, due to chain rule, $f$ is Lipschitz if each of its individual layer $g_l$ is Lipschitz [Virmaux and Scaman, 2018]. Similarly, since the layer function $g_l$ is a composition of the linear function $\mathbf{W}_l^\top \mathbf{x} + \mathbf{b}_l$ and a non-linear activation $\sigma$, $g_l$ is guanranteed to be Lipschitz if both the linear function is bounded with high probability, and the activation function $\sigma$ is also Lipschitz. In the context of neural network learning, this is often satisfied by the common practice of imposing $L_1$ or $L_2$ regularization to neural network weights, and by using standard choices of activation functions such as ReLU, leaky ReLU, tanh, etc [Virmaux and Scaman, 2018, Liu, 2019].

### E.2  Discussion on *Bernstein-von Mises* (BvM) phenomenon

**Dimensionality of the Derivative Function Space.**  Denote $\mathcal{H}$ the space of model functions spanned by the basis functions $\{b_k(\mathbf{x})\}_{k=1}^D$, such that $f(\mathbf{x}) = \sum_{k=1}^D \alpha_k b_k(\mathbf{x})$. Then, the space of partial derivative function is $\mathcal{H}_j = \{\frac{\partial}{\partial \mathbf{x}^j} f | f \in \mathcal{H}\}$. Furthermore, for every element in $\mathcal{H}_j$, we have:

$$\frac{\partial}{\partial \mathbf{x}^j} f = \sum_{k=1}^D \alpha_k \cdot [\frac{\partial}{\partial \mathbf{x}^j} b_k(\mathbf{x})].$$

That is, the derivative function space $\mathcal{H}_j$ can be spanned by $\{\frac{\partial}{\partial \mathbf{x}^j} b_k(\mathbf{x})\}_{k=1}^D$, the partial derivatives of the basis functions for the original model space $\mathcal{H}$. Furthermore, since differentiation is a linear operator, the set of linearly independent functions in $\{\frac{\partial}{\partial \mathbf{x}^j} b_k(\mathbf{x})\}_{k=1}^D$ should be equivalent to that in $\{b_k(\mathbf{x})\}_{k=1}^D$. As a result, the effective dimensionality of the derivative function space $\mathcal{H}_j$ can be controlled by the effective dimensionality of the model space $\mathcal{H}$. As an aside, for a model space $\mathcal{H}_\phi$ induced by the feature representation $\phi : \mathcal{X} \to \mathbb{R}^D$, its effective dimensionality can be measured by the rank of the feature matrix $\mathtt{rank}(\Phi)$ for $\Phi = [\phi(\mathbf{x}_1)^\top, \ldots, \phi(\mathbf{x}_n)^\top]^\top$. Alternatively, in the nonparametric literature, the effective dimensionality can also be measured by model-specific notions of "parameter count", such as the number of leaf partitions of a tree model, or the number of non-zero hidden weights of a deep neural network [Schmidt-Hieber, 2020].

**Effective Dimensionality of Statistical ML Models.**  The BvM result (Theorem 2) contains a key condition (4) $H_j = o_p(\sqrt{n})$. As stated in the main text, this condition can be satisfied if the effective dimensionality of model space $\mathcal{H}_\phi$ does not grow faster than $o_p(\sqrt{n})$ with respect to the data.

Combined with the posterior convergence condition (i.e., (1)-(2) from Theorem 1), (4) provides a more precise characterization of the convergence behavior of the model $f \in \mathcal{H}_\phi$ for the BvM phenomenon to occur. Loosely, (1)-(2) states that the model $f$ should balance its bias-variance tradeoff well enough so that the overall error rate is controlled at the rate $\epsilon_n$. Then, (4) goes one step further and states that within this bias-variance tradeoff, the variance term must be well managed, which is guaranteed by bounding the model complexity at the rate of $o_p(\sqrt{n})$.

As a matter of fact, for a wide class of ML models, a $o_p(\sqrt{n})$ bound on model complexity is not a stringent requirement, as it only prescribes a growth rate of model complexity with respect to data size. For example, the effective data size can be $C * \sqrt{n}$ for an bounded but very large $C$). Interestingly, condition (4) is in fact equivalent or looser than some of the previous BvM results obtained for specific ML models. For example, the decision tree models (e.g., BART) obtains a optimal rate when its number of partitions grow at a rate of $O((n/log n)^{d/2\gamma+d})$ for learning the space of $\gamma$-Hölder continuous functions with $\gamma > d/2$ [Rockova, 2020], which leads to a more stringent $o(\sqrt{n/log n}) < o(\sqrt{n})$ bound on complexity. A similar result also holds for deep learning models, where the number of non-zero model weights is controlled at $O(n^{d/(2\gamma+d)})$ for $\gamma > \frac{d}{2}$ ([Wang and Rocková, 2020], Theorem 3.2), which also leads to a rate of $o(\sqrt{n})$.

# F Incorporating Non-differentiability

## F.1 Incorporating Non-differentiable Model: featurized decision trees (FDT)

Several techniques have been proposed to learn a (soft) tree-structured model using gradient-optimization methods. However, either their accuracies do not match the state-of-the-art tree learning methods Yang et al. [2018] or result in models that do not obey the tree structure Irsoy et al. [2012], Frosst and Hinton [2017], Biau et al. [2019], Tanno et al. [2019]. We propose to translate a learned tree into its exact feature representation, and leverage this representation to unlock a rigorous uncertainty-aware variable importance estimation method that was previously not available for this class of models.

**Feature-based Representation of a Decision Tree**   For a certain decision tree $m$ in a learned random forest, consider the following feature map $\phi : \mathbb{R}^d \to \mathbb{R}^D$:

1. The decision tree partitions the whole feature space into $D$ cells $\mathcal{X} = \cup_{k=1}^D \mathcal{X}_k$. Label the cells of the generated partition by $1, 2, \ldots, D$ in arbitrary order.

2. To encode a data point $\mathbf{x} \in \mathbb{R}^d$, look up the label $y$ of the cell that $\mathbf{x}$ falls into and set $\phi(\mathbf{x})$ to be the (column) indicator vector of whether $\mathbf{x} \in \mathcal{X}_k$, i.e., $\phi(\mathbf{x}) = \{\mathbb{1}(\mathbf{x} \in \mathcal{X}_k)\}_{k=1}^D$.

The dimensionality $D$ of $\phi$ equals the number of leaf nodes, and each feature mapping $\phi(\mathbf{x})$ takes the one-hot form. This feature map $\phi$ induces a kernel

$$k_{dt}(\mathbf{x}, \mathbf{x}') := \phi(\mathbf{x})^\top \phi(\mathbf{x}') = \begin{cases} 1 & \text{if } \mathbf{x}, \mathbf{x}' \text{ in the same partition cell} \\ 0 & \text{otherwise} \end{cases}$$

As a result, the feature mapping $\phi(\mathbf{x})$ defines a featurized decision tree.

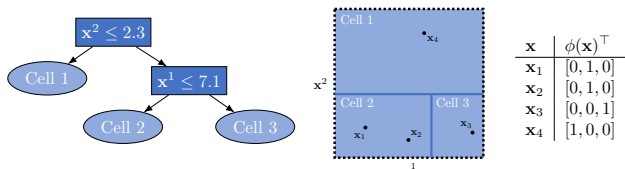

**Figure 2:** Feature expansion of a decision tree evaluated on 4 data points in $\mathbb{R}^2$. The middle panel shows the partition of $\mathbb{R}^2$ defined by the decision tree on the left. On the right is the associated feature map.

As introduced in Section 3.1, the solution for $\boldsymbol{\beta}$ is $(\Phi^\top \Phi + \sigma^2 \mathbf{I}_D)^{-1} \Phi^\top \mathbf{y}$. Note that under the decision tree kernel, $\Phi^\top \Phi = diag(n_1, \ldots, n_D)$ is a diagonal matrix of the number of training samples in each leaf cell. Therefore, the time complexity to invert the matrix $(\Phi^\top \Phi + \sigma^2 \mathbf{I}_D)$ is $O(D)$.

**Differentiable Approximation**   The random features generated by Figure 2 can be written as

$$\begin{aligned} \phi(\mathbf{x}) &= (\mathbb{1}(\mathbf{x}^2 \leq 2.3), \mathbb{1}(\mathbf{x}^2 > 2.3, \mathbf{x}^1 \leq 7.1), \mathbb{1}(\mathbf{x}^2 > 2.3, \mathbf{x}^1 > 7.1)) \\ &= (\mathbb{1}(\mathbf{x}^2 \leq 2.3), \mathbb{1}(\mathbf{x}^2 > 2.3) \cdot \mathbb{1}(\mathbf{x}^1 \leq 7.1), \mathbb{1}(\mathbf{x}^2 > 2.3) \cdot \mathbb{1}(\mathbf{x}^1 > 7.1)). \end{aligned}$$

To calculate variable importance, the indicator function needs to be approximated by a smooth function, so that we can take the derivative with respect to each feature. In this work, we consider approximating the indicator function using the sigmoid function Irsoy et al. [2012]:

$$\mathbb{1}(x > a) \approx i_c(x > a) = \frac{1}{1 + \exp(-c \cdot (x - a))},$$

and analogously, $\mathbb{1}(x \leq a) = 1 - i_c(x > a)$. Here $c$ is a hyperparameter that controls the smoothness of the approximation. A larger $c$ leads to a better approximation to the random forest algorithm, but may result in a non-smooth prediction function which may be undesirable for approximating an continuous regression function $f_0$.

## F.2 Incorporating Discrete Features

Compared to the empirical derivative norm, a more principled way to measure the variable importance of a discrete feature is *contrast*, which is the square of the difference in predictions when fixing the feature to a certain value versus fixing it to the other value, while keeping the other features the same. Specifically, we can consider defining a discrete version of the derivative:

$$D_j f = f(\mathbf{x}^j = 1, \mathbf{x}^{-j}) - f(\mathbf{x}^j = 0, \mathbf{x}^{-j}), \tag{17}$$

where $\mathbf{x}^{-j}$ denotes all features with $\mathbf{x}^j$ removed.

Then, in the case where the feature takes two values, we can set one of them as the reference group with value 0 and the other group with value 1,

$$
\begin{aligned}
\Psi_j(f) = \|D_j f\|_2^2 &= \int_{\mathbf{x} \in \mathcal{X}} |D_j f|^2 dP(\mathbf{x}) \\
&= \int_{\mathbf{x} \in \mathcal{X}} |f(\mathbf{x}^j = 1, \mathbf{x}^{-j}) - f(\mathbf{x}^j = 0, \mathbf{x}^{-j})|^2 dP(\mathbf{x}),
\end{aligned}
$$

Since $P(\mathbf{x})$ is not known from the training observations, $\Psi_j(f)$ can be approximated by its empirical counterpart:

$$
\begin{aligned}
\psi_j(f) = \|D_j f\|_n^2 &= \frac{1}{n} \sum_{i=1}^n |D_j f|^2. \\
&= \frac{1}{n} \sum_{i=1}^n |f(\mathbf{x}_i^j = 1, \mathbf{x}_i^{-j}) - f(\mathbf{x}_i^j = 0, \mathbf{x}_i^{-j})|^2.
\end{aligned}
$$

In the case where the feature takes multiple groups, we can calculate the pairwise contrasts and take the $L_2$ norm. Empirically, using contrast for discrete feature improves the performance of variable importance estimation. As contrast is a linear function of the original prediction function $f$, the posterior convergence of $\psi_j$ with respect to this operator is guaranteed by the convergence of the prediction function $f$. Similarly, the BvM phenomenon is guaranteed when $D_j f$ is bounded and $H_j = D_j^\top D_j$ has rank $o_p(\sqrt{n})$ (i.e., the similar set of conditions in Theorem 2 but with the original $D_j$ replaced by its discrete counterpart Equation (17)).

# G   Further Experiment Detail

## G.1   Methods

We consider three main classes of models (Table 1).

I  **Random Forests (RF)**

- **FDT**: Given a trained forest, we quantify variable importance using $\psi_j$ by translating it to an ensemble of **FDT** (Appendix F.1). We use a variant of random forest here, extra trees Geurts et al. [2006] since it performs better. We use $50$ trees to build the forest and maximum number of leaf nodes for each tree is $\sqrt{n}\,log(n)$. Throughout our experiment, we fix $c = 1$ for continuous features calculated using integrated partial derivatives and fix $c = 0.1$ for discrete features calculated using contrasts. We use `scikit-learn` package in Python to train the random forest.
- **RF-impurity** [Breiman et al., 1984]: It measures variable importance with their impurity based on the average reduction of the loss function were the variable to be removed. We also use extra trees here. We use $50$ trees to build the forest and maximum number of leaf nodes for each tree is $\sqrt{n}\,log(n)$. We use `scikit-learn` package in Python to train the random forest.
- **RF-knockoff** [Candes et al., 2017]: It uses random forest statistics to assess variable importance in our case. We use `knockoff` package in R to calculate the statistic.
- Bayesian additive regression trees (**BART**) Chipman et al. [2010]: It produces a measure of variable importance by tracking variable inclusion proportions, enabling variable selection with a user-defined threshold. We use `bartMachine` package in R to train the model.

II  **(Approximate) Kernel Methods** & **Neural Networks**

- Random Fourier Feature model (**RFF**): We apply $\psi_j$ to a random-feature model that approximates a Gaussian process with an RBF kernel Rahimi and Recht [2007], and set the number of features to $\sqrt{n}\,log(n)$ to ensure proper approximation of the exact RBF-GP Rudi and Rosasco [2018]. We choose the lengthscale parameter of RBF-GP from a list of lengthscale candidates $\{5, 10, 16, 23\}$ based on the prediction performance on testing data.
- Bayesian kernel machine regression (**BKMR**) Bobb et al. [2015]: It is based on a GP with exact RBF kernel and spike-and-slab prior, using posterior inclusion probabilities to perform variable selection. We use `bkmr` package in R to train the model and the number of iterations of the MCMC sampler is set to be $4000$.
- Bayesian Approximate Kernel Regression (**BAKR**) [Crawford et al., 2018]: It is based on random-feature model with a projection-based feature importance measure and an adaptive shrinkage prior, using squared estimates of the parameter coefficients to perform variable selection. We use `BAKR` repository from the author's GitHub to train the model and the number of iterations of the MCMC sampler is set to be $2000$.
- Sparse Neural Networks (**NN**): We apply $\psi_j$ to a 1-layer neural network with 512 hidden units and $L_1$ regularization (i.e., LASSO net) on the hidden layer, implemented in the `tensorflow.keras` framework. We train the model with Adam optimizer and early stopping with respect to validation RMSE. We sweep the regularization strength of $L_1$ penalty in exponential grids with exponents $\{-3, -2, -1, 0, 1., 2, 3\}$. We also experimented with deeper layers (up to 3 layers) and observed similar performance.

III  **Linear Models**

- **GAM**: We apply $\psi_j$ to a featurized GP representation of the **GAM**, with the prior center $\mu$ set at the frequentist estimate of the original GAM model obtained from a sophisticated REML procedure [Wood, 2006]. We use `mgcv` package in R to train the model.
- Bayesian Ridge Regression (**BRR**) Hoerl and Kennard [1970]: It applies a fixed prior for each feature, using squared estimates of the parameter coefficients to perform variable selection. We use `BGLR` package in R to train the model and the number of iterations of the MCMC sampler is set to be $2000$.

- Bayesian Lasso (**BL**) Park and Casella [2008]: It developed a Bayesian way to access the Lasso estimate which allows tractable full conditional distributions, using squared estimates of the parameter coefficients to perform variable selection. We use `BGLR` package in R to train the model and the number of iterations of the MCMC sampler is set to be 2000.

The results in this paper were obtained using R 4.1.0 or Python 3.7. All experiments were run on a Linux-based high performance computing cluster using SLURM-managed CPU resources.

## G.2 Data

**Outcome-generating function**   As discussed earlier, we generate data under the homoscedastic Gaussian noise model $y \sim \mathcal{N}(f_0(\mathbf{x}), 0.01)$ for different sparse functions $f_0$ and features $\mathbf{x}$. Given $n \in \{100, 200, 500, 1000\}$ observations in $d \in \{25, 50, 100, 200\}$ dimensions, the goal is to model $f_0$ while identifying the $d^* = 5$ features on which $f_0$ depends. To this end we report mean squared error (MSE) to quantify prediction performance and AUROC scores to quantify variable importance estimation performance.

We consider four settings of the data-generation function $f_0$:

1) `linear`: a simple linear function $f_0(\mathbf{x}) = \mathbf{x}^1 - \mathbf{x}^2 + \mathbf{x}^3 + 0.5\mathbf{x}^4 + 2\mathbf{x}^5$;

2) `rbf`: a Gaussian RBF kernel with length-scale 1. This kernel represents the space of functions that are smooth (i.e., infinitely differentiable) and have reasonable complexity (i.e., does not have fast-varying fluctuations that are difficult to model);

3) `matern32`: a matérn $\frac{3}{2}$ kernel with length-scale 1. Compared to RBF, it has the same degree of complexity but is less smooth, in the sense that it represents the space of once-differentiable functions, but is not necessarily infinitely differentiable;

4) `complex`: a complicated and non-smooth multivariate function that is outside the RKHS $\mathcal{H}$:
$f_0(\mathbf{x}) = \frac{\sin(\max(\mathbf{x}^1, \mathbf{x}^2)) + \arctan(\mathbf{x}^2)}{1 + \mathbf{x}^1 + \mathbf{x}^5} + \sin(0.5\mathbf{x}^3)(1 + \exp(\mathbf{x}^4 - 0.5\mathbf{x}^3)) + \mathbf{x}^{3^2} + 2\sin(\mathbf{x}^4) + 4\mathbf{x}^5$, which is non-continuous in terms of $\mathbf{x}^1, \mathbf{x}^2$ but infinitely differentiable in terms of $\mathbf{x}^3, \mathbf{x}^4, \mathbf{x}^5$.

**Synthetic Benchmarks**   We create synthetic benchmark datasets of varying number of observations $n$ and number of features $d$. The **synthetic-continuous** dataset uses only continuous features, and the **synthetic-mixture** dataset uses a mixture of continuous and discrete features. The synthetic features are drawn either from $Bern(0.5)$ (if discrete) or $Unif(-2, 2)$ (if continuous). Additionally, each feature is either causal (i.e., used by $f_0$) or non-causal. For each simulation setting, there are always $d^* = 5$ causal features. Specifically, in the **synthetic-continuous** dataset, all features are continuous, while in the **synthetic-mixture** dataset, there are 2 discrete and 3 continuous causal features, while there are 2 discrete non-causal features (all the rest of non-causal features are continous).

For each sample size - data dimension scenario, we use the same set of generated features across the repeated simulation runs.

**Socio-economic and Healthcare Data**

- **adult**: 1994 U.S. census data of 48842 adults with 8 categorical and 6 continuous features Kohavi. The data is publicly available[6] and does not contain personally identifiable information or offensive content. We concatenated the training data (`adult.data`) and testing data (`adult.test`), and remove all observations with missing features. Additionally, we removed the redundant feature "education", and performed suitable re-categorization for discrete features: For "`race`", we encoded "White" as 0 and the rest as 1; for "`sex`", we encoded "Female" as 1 and "Male" as 0; for "`relationship`", we encoded "Husband" as 0, "Not-in-family" as 1 and the rest as 2; for "`workclass`", we encoded "Private" as 0, "Self-emp-not-inc" as 1 and the rest as 2; for "marital_status", we encoded "Married-civ-spouse" as 0, "Never-married" as 1 and the rest as 2; for "`occupation`", we encoded "Prof-specialty" as 0, "Craft-repair" as 1 and the rest as 2;

---

[6]`https://archive.ics.uci.edu/ml/machine-learning-databases/adult/`

for "native_country", we encoded "United-States" as 0, "Mexico" as 1 and the rest as 2. The final features in the dataset are: ("race", "sex", "education_num", "hours_per_week", "age", "relationship", "workclass", "fnlwgt", "capital_gain", "capital_loss", "marital_status", "occupation", "native_country"). If the data dimension is higher than 13, additional features will be generated from $Unif(-2, 2)$.

- **heart**: a coronary artery disease dataset of 303 patients from Cleveland clinic database with 7 categorical and 6 continuous features Detrano et al. [1989]. The data is publicly available[7] and does not contain personally identifiable information or offensive content. All observations with missing features are removed before analysis.

  The list of features used in the final datasets are ("sex", "exang", "thal", "oldpeak", "age", "ca", "cp", "chol", "trestbps", "thalach", "fbs", "restecg", "slope"). If the data dimension is higher than 13, additional features will be generated from $Unif(-2, 2)$.

- **mi**: disease records of myocardial infarction (MI) of 1700 patients from Krasnoyarsk interdistrict clinical hospital during 1992-1995, with 113 categorical and 11 continuous features Golovenkin et al. [2020]. The data is publicly available[8] and does not contain personally identifiable information or offensive content. We imputed missing values using the IterativeImputer method from scikit-learn package and with a BayesianRidge regressor. Specifically, it imputes each feature with missing values as a function of other features in a round-robin fashion: At each step, a feature column is designated as output $y$ and the other feature columns are treated as inputs $X$. A regressor is fit on $(X, y)$ for known $y$. Then, the regressor is used to predict the missing values of $y$. This is done for each feature in an iterative fashion, and then is repeated for 10 imputation rounds. The results of the final imputation round are returned.

  The listed of features used in the analysis are as below: ("sex", "ritm_ecg_p_01", "age", "s_ad_orit", "d_ad_orit", "ant_im", "ibs_post", "k_blood", "na_blood", "l_blood", "inf_anam", "stenok_an", "fk_stenok", "ibs_nasl", "gb", "sim_gipert", "dlit_ag", "zsn_a", "nr11", "nr01", "nr02", "nr03", "nr04", "nr07", "nr08", "np01", "np04", "np05", "np07", "np08", "np09", "np10", "endocr_01", "endocr_02", "endocr_03", "zab_leg_01", "zab_leg_02", "zab_leg_03", "zab_leg_04", "zab_leg_06", "s_ad_kbrig", "d_ad_kbrig", "o_l_post", "k_sh_post", "mp_tp_post", "svt_post", "gt_post", "fib_g_post", "lat_im", "inf_im", "post_im", "im_pg_p", "ritm_ecg_p_02", "ritm_ecg_p_04", "ritm_ecg_p_06", "ritm_ecg_p_07", "ritm_ecg_p_08", "n_r_ecg_p_01", "n_r_ecg_p_02", "n_r_ecg_p_03", "n_r_ecg_p_04", "n_r_ecg_p_05", "n_r_ecg_p_06", "n_r_ecg_p_08", "n_r_ecg_p_09", "n_r_ecg_p_10", "n_p_ecg_p_01", "n_p_ecg_p_03", "n_p_ecg_p_04", "n_p_ecg_p_05", "n_p_ecg_p_06", "n_p_ecg_p_07", "n_p_ecg_p_08", "n_p_ecg_p_09", "n_p_ecg_p_10", "n_p_ecg_p_11", "n_p_ecg_p_12", "fibr_ter_01", "fibr_ter_02", "fibr_ter_03", "fibr_ter_05", "fibr_ter_06", "fibr_ter_07", "fibr_ter_08", "gipo_k", "giper_na", "alt_blood", "ast_blood", "kfk_blood", "roe", "time_b_s", "r_ab_1_n", "r_ab_2_n", "r_ab_3_n", "na_kb", "not_na_kb", "lid_kb", "nitr_s", "na_r_1_n", "na_r_2_n", "na_r_3_n", "not_na_1_n", "not_na_2_n", "not_na_3_n", "lid_s_n", "b_block_s_n", "ant_ca_s_n", "gepar_s_n", "asp_s_n", "tikl_s_n", "trent_s_n").

We standardize (by subtracting from mean and dividing by standard deviation) all features except for 2 discrete causal features and 2 discrete non-causal features.

---

[7]https://archive.ics.uci.edu/ml/machine-learning-databases/heart-disease/processed.cleveland.data

[8]https://archive.ics.uci.edu/ml/machine-learning-databases/00579/

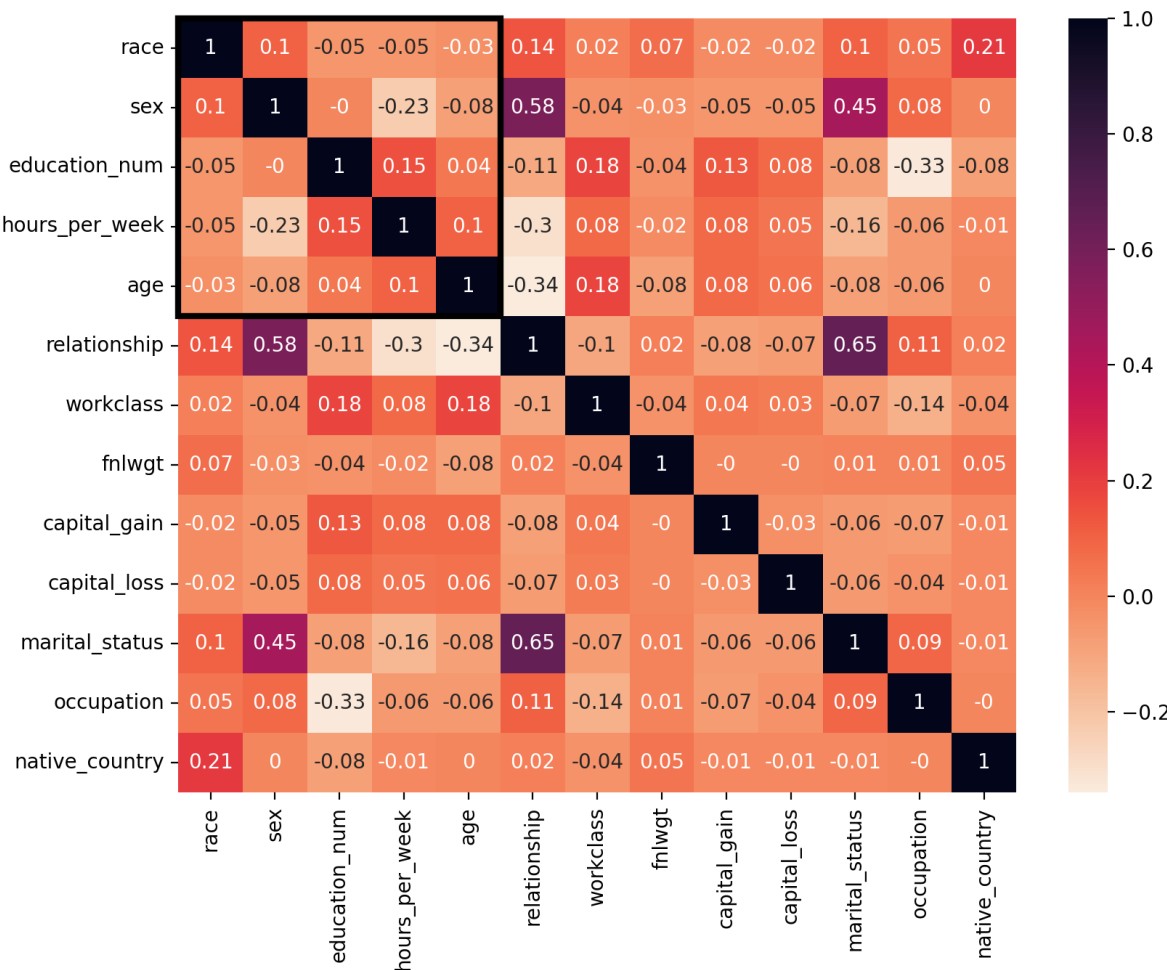

**Figure 3:** Correlation matrix for **adult** dataset, where the upper left black box indicates the five causal features.

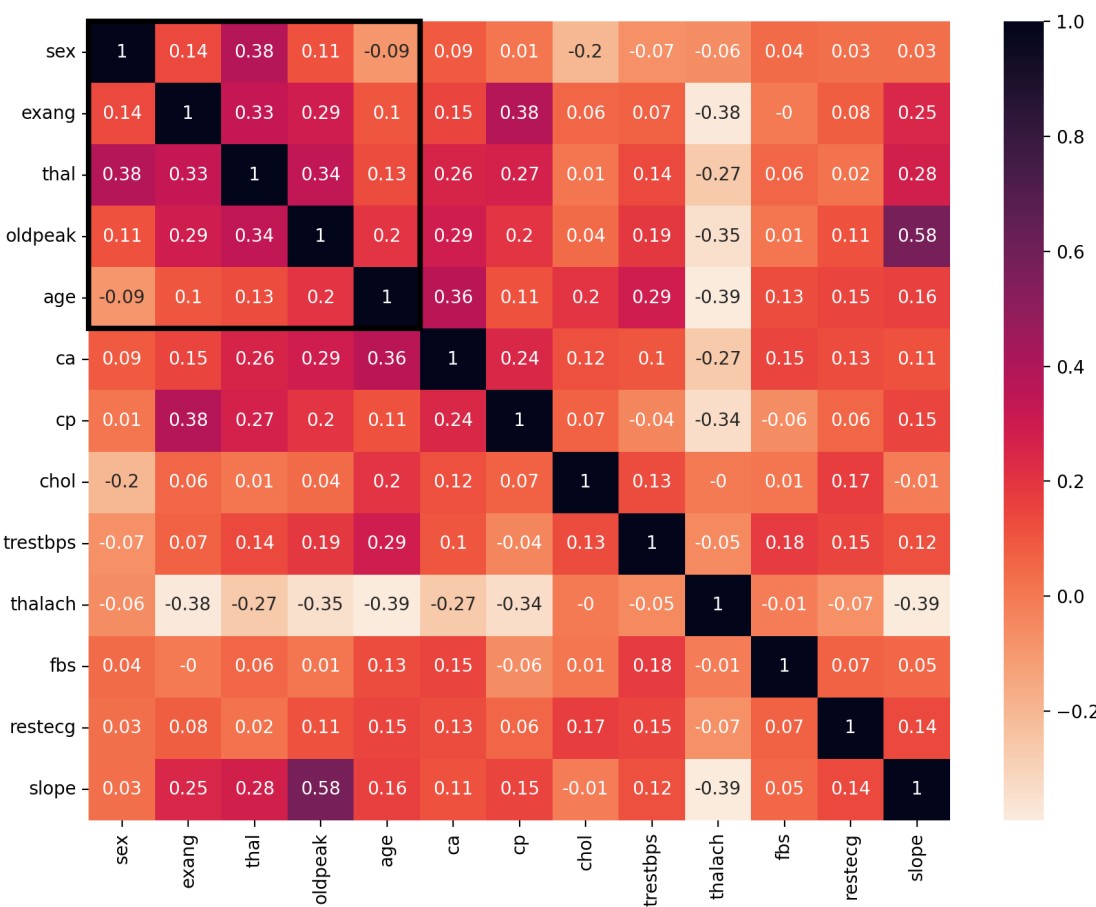

**Figure 4:** Correlation matrix for **heart** dataset, where the upper left black box indicates the five causal features.

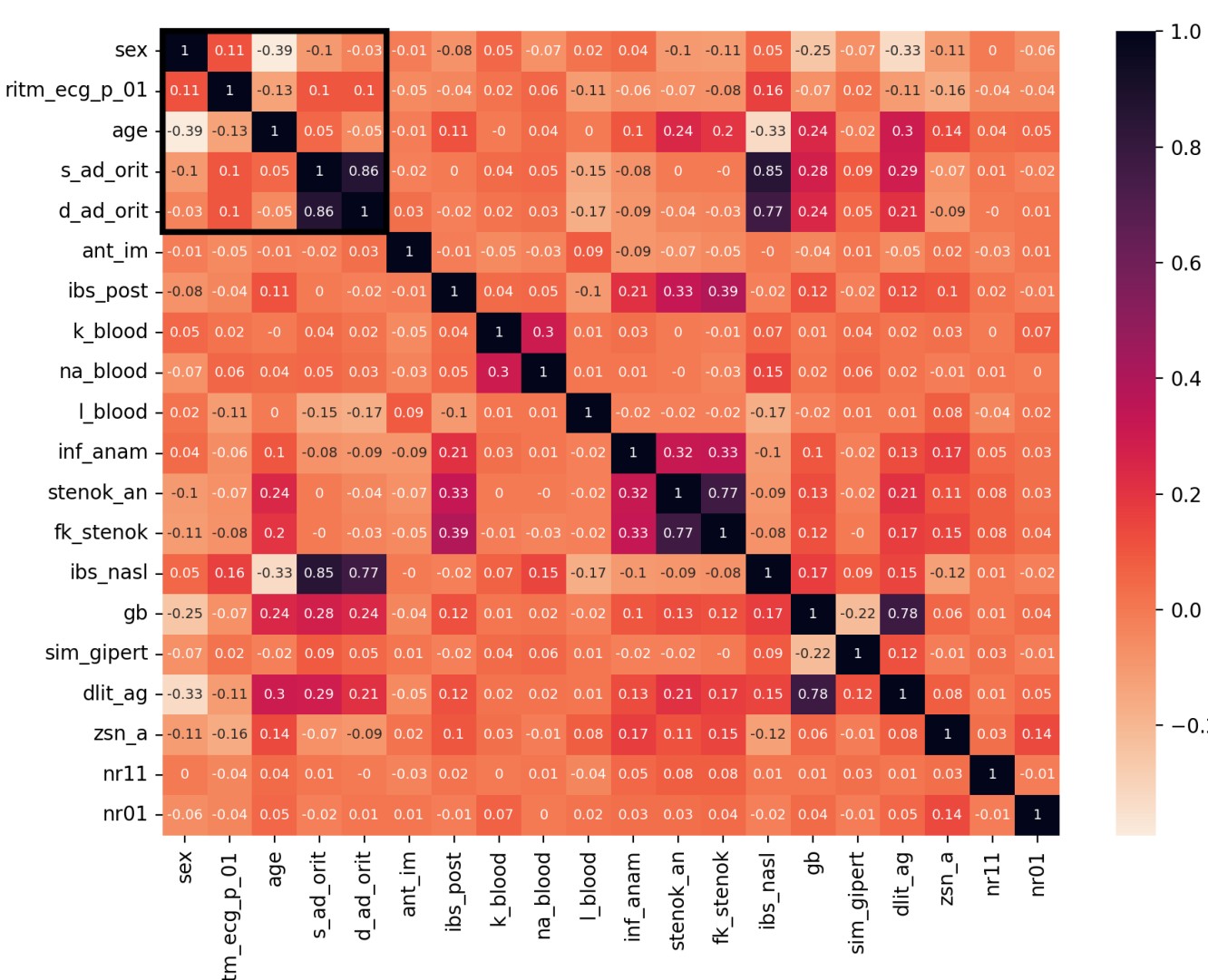

**Figure 5:** Correlation matrix for the first 20 features in **mi** dataset, where the upper left black box indicates the five causal features.

## G.3 Error Bars

Tables 2-11 shows the AUROC scores or Testing MSE's for the result presented in the main text. For the Testing MSE tables, a method will not be shown if they share the model fit with another method (**RF-impurity** and **RF-knockoff**), or if the method does not produce valid result due to small sample size (**GAM**).

**Table 2:** AUROC scores and their standard deviations for **synthetic-mixture** dataset.

| n | RF-FDT (Ours) | NN (Ours) | GAM (Ours) | RF-Impurity | RFF (Ours) | BRR | RF-KnockOff | BKMR | BL | BART | BAKR |
|---|---|---|---|---|---|---|---|---|---|---|---|
| 100 | 0.8 (0.09) | 0.68 (0.03) | NaN (NA) | 0.72 (0.13) | 0.59 (0.11) | 0.66 (0.12) | 0.57 (0.06) | 0.57 (0.09) | 0.68 (0.09) | 0.71 (0.16) | 0.56 (0.09) |
| 200 | 0.93 (0.1) | 0.72 (0.03) | 0.72 (0.15) | 0.86 (0.2) | 0.65 (0.18) | 0.69 (0.16) | 0.59 (0.06) | 0.57 (0.08) | 0.69 (0.17) | 0.78 (0.14) | 0.68 (0.12) |
| 500 | 0.97 (0.05) | 0.75 (0.03) | 0.88 (0.07) | 1 (0) | 0.68 (0.2) | 0.83 (0.09) | 0.67 (0.1) | 0.64 (0.11) | 0.83 (0.09) | 0.95 (0.08) | 0.75 (0.14) |
| 1000 | 0.99 (0.03) | 0.79 (0.03) | 0.89 (0.1) | 1 (0) | 0.69 (0.25) | 0.86 (0.1) | 0.69 (0.1) | 0.68 (0.16) | 0.86 (0.1) | 0.99 (0.02) | 0.77 (0.1) |

**Table 3:** AUROC scores and their standard deviations for **synthetic-continuous** dataset.

| n | RF-FDT (Ours) | NN (Ours) | GAM (Ours) | RF-Impurity | RFF (Ours) | BRR | RF-KnockOff | BKMR | BL | BART | BAKR |
|---|---|---|---|---|---|---|---|---|---|---|---|
| 100 | 0.61 (0.13) | 0.68 (0.05) | NaN (NA) | 0.68 (0.13) | 0.56 (0.12) | 0.69 (0.17) | 0.61 (0.12) | 0.62 (0.1) | 0.69 (0.13) | 0.69 (0.14) | 0.61 (0.1) |
| 200 | 0.83 (0.13) | 0.73 (0.04) | 0.74 (0.12) | 0.85 (0.13) | 0.66 (0.11) | 0.75 (0.15) | 0.68 (0.13) | 0.57 (0.08) | 0.75 (0.13) | 0.79 (0.12) | 0.61 (0.14) |
| 500 | 0.98 (0.03) | 0.76 (0.03) | 0.8 (0.14) | 0.99 (0.02) | 0.71 (0.16) | 0.8 (0.12) | 0.87 (0.1) | 0.59 (0.1) | 0.8 (0.12) | 0.96 (0.03) | 0.76 (0.12) |
| 1000 | 1 (0) | 0.79 (0.03) | 0.86 (0.11) | 1 (0) | 0.63 (0.24) | 0.87 (0.14) | 0.97 (0.08) | 0.64 (0.15) | 0.86 (0.14) | 1 (0) | 0.79 (0.14) |

**Table 4:** AUROC scores and their standard deviations for **adult** dataset.

| n | RF-FDT (Ours) | NN (Ours) | GAM (Ours) | RF-Impurity | RFF (Ours) | BRR | RF-KnockOff | BKMR | BL | BART | BAKR |
|---|---|---|---|---|---|---|---|---|---|---|---|
| 100 | 0.76 (0.09) | 0.68 (0.04) | NaN (NA) | 0.61 (0.15) | 0.62 (0.13) | 0.57 (0.09) | 0.58 (0.13) | 0.54 (0.1) | 0.61 (0.09) | 0.66 (0.11) | 0.62 (0.13) |
| 200 | 0.8 (0.09) | 0.7 (0.04) | 0.7 (0.14) | 0.64 (0.11) | 0.6 (0.14) | 0.61 (0.11) | 0.57 (0.1) | 0.58 (0.12) | 0.63 (0.09) | 0.7 (0.12) | 0.72 (0.12) |
| 500 | 0.84 (0.07) | 0.75 (0.02) | 0.79 (0.13) | 0.64 (0.09) | 0.57 (0.18) | 0.62 (0.09) | 0.59 (0.12) | 0.59 (0.08) | 0.6 (0.1) | 0.71 (0.08) | 0.64 (0.11) |
| 1000 | 0.81 (0.1) | 0.77 (0.02) | 0.86 (0.1) | 0.61 (0.08) | 0.64 (0.18) | 0.7 (0.12) | 0.57 (0.09) | 0.55 (0.08) | 0.69 (0.1) | 0.7 (0.11) | 0.72 (0.14) |

**Table 5:** AUROC scores and their standard deviations for **heart** dataset.

| n | RF-FDT (Ours) | NN (Ours) | GAM (Ours) | RF-Impurity | RFF (Ours) | BRR | RF-KnockOff | BKMR | BL | BART | BAKR |
|---|---|---|---|---|---|---|---|---|---|---|---|
| 50 | 0.71 (0.06) | 0.66 (0.04) | NaN (NA) | 0.49 (0.15) | 0.56 (0.14) | 0.58 (0.08) | 0.59 (0.06) | 0.59 (0.09) | 0.58 (0.07) | 0.63 (0.09) | 0.6 (0.09) |
| 100 | 0.72 (0.06) | 0.66 (0.04) | NaN (NA) | 0.44 (0.11) | 0.58 (0.12) | 0.58 (0.09) | 0.58 (0.07) | 0.57 (0.08) | 0.57 (0.08) | 0.59 (0.11) | 0.59 (0.12) |
| 150 | 0.75 (0.08) | 0.67 (0.05) | 0.62 (0.12) | 0.41 (0.12) | 0.59 (0.13) | 0.59 (0.06) | 0.61 (0.11) | 0.58 (0.07) | 0.56 (0.08) | 0.6 (0.07) | 0.64 (0.12) |
| 257 | 0.74 (0.09) | 0.72 (0.03) | 0.64 (0.12) | 0.45 (0.12) | 0.52 (0.17) | 0.57 (0.07) | 0.66 (0.13) | 0.59 (0.09) | 0.56 (0.06) | 0.58 (0.09) | 0.64 (0.11) |

**Table 6:** AUROC scores and their standard deviations for **mi** dataset.

| n | RF-FDT (Ours) | NN (Ours) | GAM (Ours) | RF-Impurity | RFF (Ours) | BRR | RF-KnockOff | BKMR | BL | BART | BAKR |
|---|---|---|---|---|---|---|---|---|---|---|---|
| 100 | 0.86 (0.05) | 0.64 (0.04) | NaN (NA) | 0.77 (0.08) | 0.59 (0.13) | 0.65 (0.1) | 0.67 (0.12) | 0.57 (0.09) | 0.63 (0.11) | 0.62 (0.14) | 0.86 (0.05) |
| 200 | 0.85 (0.04) | 0.73 (0.05) | 0.87 (0.05) | 0.79 (0.07) | 0.62 (0.08) | 0.62 (0.09) | 0.63 (0.12) | 0.58 (0.1) | 0.65 (0.1) | 0.61 (0.11) | 0.84 (0.07) |
| 500 | 0.85 (0.05) | 0.75 (0.03) | 0.87 (0.07) | 0.77 (0.06) | 0.43 (0.15) | 0.64 (0.09) | 0.62 (0.08) | 0.59 (0.1) | 0.61 (0.09) | 0.6 (0.08) | 0.81 (0.1) |
| 1000 | 0.83 (0.04) | 0.77 (0.02) | 0.89 (0.06) | 0.73 (0.07) | 0.56 (0.17) | 0.6 (0.13) | 0.62 (0.11) | 0.54 (0.08) | 0.67 (0.1) | 0.63 (0.11) | 0.88 (0.09) |

**Table 7:** Testing MSE's and their standard deviations for **synthetic-mixture** dataset. A method will not be shown if they share the model fit with another method (**RF-impurity** and **RF-knockoff**), or if the method does not produce valid result due to small sample size (**GAM**)

| n | RF-FDT (Ours) | NN (Ours) | GAM (Ours) | RFF (Ours) | BRR | BKMR | BL | BART | BAKR |
|---|---|---|---|---|---|---|---|---|---|
| 100 | 1.02 (0.25) | 1.06 (0.22) | NaN (NA) | 1.76 (0.28) | 1.01 (0.23) | 1.52 (0.14) | 0.95 (0.21) | 0.98 (0.23) | 2.87 (1.06) |
| 200 | 0.87 (0.16) | 1.02 (0.12) | 1.59 (0.29) | 1.53 (0.22) | 1.01 (0.13) | 1.56 (0.09) | 0.95 (0.12) | 0.98 (0.16) | 1.04 (0.14) |
| 500 | 0.76 (0.13) | 1.04 (0.12) | 1.04 (0.2) | 1.42 (0.15) | 0.94 (0.13) | 1.57 (0.08) | 0.93 (0.13) | 0.83 (0.14) | 0.97 (0.11) |
| 1000 | 0.66 (0.12) | 1.03 (0.13) | 0.96 (0.18) | 1.32 (0.15) | 0.94 (0.16) | 1.62 (0.07) | 0.93 (0.17) | 0.75 (0.12) | 1.01 (0.13) |

**Table 8:** Testing MSE's and their standard deviations for **synthetic-continuous** dataset. A method will not be shown if they share the model fit with another method (**RF-impurity** and **RF-knockoff**), or if the method does not produce valid result due to small sample size (**GAM**)

| n | RF-FDT (Ours) | NN (Ours) | GAM (Ours) | RFF (Ours) | BRR | BKMR | BL | BART | BAKR |
|---|---|---|---|---|---|---|---|---|---|
| 100 | 1.01 (0.2) | 1.06 (0.16) | NaN (NA) | 1.73 (0.26) | 1.05 (0.15) | 1.55 (0.11) | 1.01 (0.16) | 1.02 (0.15) | 2.39 (0.59) |
| 200 | 0.87 (0.15) | 0.97 (0.19) | 1.41 (0.32) | 1.48 (0.23) | 0.92 (0.18) | 1.5 (0.15) | 0.9 (0.16) | 0.93 (0.19) | 0.95 (0.19) |
| 500 | 0.85 (0.19) | 1.08 (0.12) | 1.02 (0.25) | 1.43 (0.13) | 0.95 (0.19) | 1.58 (0.09) | 0.91 (0.19) | 0.93 (0.2) | 1 (0.15) |
| 1000 | 0.72 (0.15) | 1.05 (0.18) | 0.94 (0.2) | 1.4 (0.19) | 0.91 (0.18) | 1.6 (0.11) | 0.9 (0.18) | 0.8 (0.19) | 0.98 (0.18) |

**Table 9:** Testing MSE's and their standard deviations for **adult** dataset. A method will not be shown if they share the model fit with another method (**RF-impurity** and **RF-knockoff**), or if the method does not produce valid result due to small sample size (**GAM**)

| n | RF-FDT (Ours) | NN (Ours) | GAM (Ours) | RFF (Ours) | BRR | BKMR | BL | BART | BAKR |
|---|---|---|---|---|---|---|---|---|---|
| 100 | 0.95 (0.4) | 0.96 (0.13) | NaN (NA) | 1.69 (0.44) | 0.3 (0.11) | 0.92 (0.15) | 0.22 (0.07) | 0.4 (0.11) | 2.71 (0.92) |
| 200 | 0.91 (0.24) | 1.02 (0.2) | 1.23 (0.32) | 1.63 (0.31) | 0.28 (0.07) | 1.03 (0.07) | 0.22 (0.05) | 0.4 (0.12) | 0.28 (0.07) |
| 500 | 0.96 (0.18) | 1 (0.11) | 0.45 (0.08) | 1.31 (0.14) | 0.25 (0.06) | 1 (0.08) | 0.22 (0.07) | 0.28 (0.08) | 0.22 (0.07) |
| 1000 | 0.96 (0.18) | 1.02 (0.14) | 0.32 (0.06) | 1.28 (0.17) | 0.24 (0.05) | 1.08 (0.04) | 0.21 (0.04) | 0.28 (0.12) | 0.21 (0.04) |

**Table 10:** Testing MSE's and their standard deviations for **heart** dataset. A method will not be shown if they share the model fit with another method (**RF-impurity** and **RF-knockoff**), or if the method does not produce valid result due to small sample size (**GAM**)

| n | RF-FDT (Ours) | NN (Ours) | GAM (Ours) | RFF (Ours) | BRR | BKMR | BL | BART | BAKR |
|---|---|---|---|---|---|---|---|---|---|
| 50 | 0.92 (0.21) | 1.03 (0.16) | NaN (NA) | 1.93 (0.5) | 0.35 (0.12) | 1.03 (0.13) | 0.26 (0.1) | 0.39 (0.12) | 0.44 (0.18) |
| 100 | 0.95 (0.27) | 1.09 (0.23) | NaN (NA) | 1.88 (0.29) | 0.32 (0.08) | 1.02 (0.1) | 0.24 (0.06) | 0.43 (0.13) | 2.18 (0.68) |
| 150 | 0.98 (0.22) | 1.06 (0.19) | 1.76 (0.37) | 1.65 (0.32) | 0.27 (0.08) | 0.99 (0.12) | 0.21 (0.08) | 0.4 (0.14) | 0.31 (0.1) |
| 257 | 0.91 (0.26) | 1.04 (0.15) | 0.79 (0.2) | 1.51 (0.2) | 0.27 (0.06) | 1 (0.1) | 0.23 (0.06) | 0.33 (0.12) | 0.25 (0.06) |

**Table 11:** Testing MSE's and their standard deviations for **mi** dataset. A method will not be shown if they share the model fit with another method (**RF-impurity** and **RF-knockoff**), or if the method does not produce valid result due to small sample size (**GAM**)

| n | RF-FDT (Ours) | NN (Ours) | GAM (Ours) | RFF (Ours) | BRR | BKMR | BL | BART | BAKR |
|---|---|---|---|---|---|---|---|---|---|
| 100 | 1.55 (0.94) | 1.63 (1.1) | NaN (NA) | 2.02 (0.44) | 0.36 (0.15) | 0.78 (0.22) | 0.31 (0.1) | 0.32 (0.1) | 0.66 (0.32) |
| 200 | 1.76 (2.86) | 1.29 (0.63) | 0.61 (0.27) | 1.82 (0.48) | 0.32 (0.11) | 0.85 (0.22) | 0.3 (0.11) | 0.28 (0.12) | 0.41 (0.15) |
| 500 | 1.13 (0.35) | 1.17 (0.44) | 0.42 (0.22) | 1.57 (0.27) | 0.27 (0.07) | 0.53 (0.25) | 0.26 (0.06) | 0.26 (0.09) | 0.26 (0.09) |
| 1000 | 1.2 (1.05) | 1 (0.16) | 0.36 (0.08) | 1.43 (0.3) | 0.28 (0.07) | 0.71 (0.34) | 0.27 (0.07) | 0.24 (0.07) | 0.25 (0.07) |

### G.4 Code

For the code, data, instructions, the total amount of compute and the type of resources used to reproduce the experimental results, please visit `https://github.com/wdeng5120/featurized-decision-tree`.

# H Experiment Results and Additional Figures

Figures 6-10 and 11-15 show the AUROC scores and MSE results, respectively, across all of the datasets. Here we also summarize additional observations that are not included in the main text. The figure captions contain further descriptions of the results.

**Synthetic Benchmarks**. In the synthetic datasets, where all features are independent, FDT, RF, BART, GAM, BRR and BL perform better and more stable than they do in the real datasets where there's feature correlation. The better performance of FDT compared to **RF-impurity** and **RF-knockoff** illustrates the advantage of the proposed integrated partial derivative metric for variable importance estimation. For the **synthetic-continuous** and **synthetic-mixture** cases, FDT has higher AUROC scores across most scenarios, especially when data are generated having high complexity with quickly-varying local fluctuations (rbf, matern32). Moreover, all 11 methods perform only moderately well in complex data settings. The two tree-based methods, RF and BART also have high AUROC scores across scenarios, since the tree-based methods naturally rank by how well the features improve the purity of the node. Note that under low dimension case ($d = 25$), BKMR is comparable to FDT when $f_0 \in \mathcal{H}$ (linear, rbf, matern32). However, when it comes to medium- or relatively high-dimension settings ($d = 50, 100$), BKMR produces low AUROC scores due to suffering from the issue of curse of dimensionality van der Vaart and Zanten [2011]. RFF, also a kernel-based method, has similar trend as BKMR. Finally, BAKR performs consistently poorly and has lowest AUROC scores in relatively low-dimension setting ($d = 25, 50$). Linear models (GAM, BRR and BL) achieve comparable or superior performance under the linear data setting. However, for more complicated data generation functions, BRR and BL consistently perform poorly with low AUROC scores.

**Socio-economic and Healthcare Datasets** In the **adult**, **heart** and **mi** cases, where the features are correlated, the performances of all 11 methods are worse than in the **synthetic-mixture** and **synthetic-continuous** cases (where the features are independent). Their performance tends to saturate earlier and are less stable with respect to the sample size. In relatively low-dimension settings ($d = 25, 50$), the standard methods such as BART has higher AUROC scores than FDT. However, when the dimension is higher ($d = 100, 200$), FDT consistently performs better.

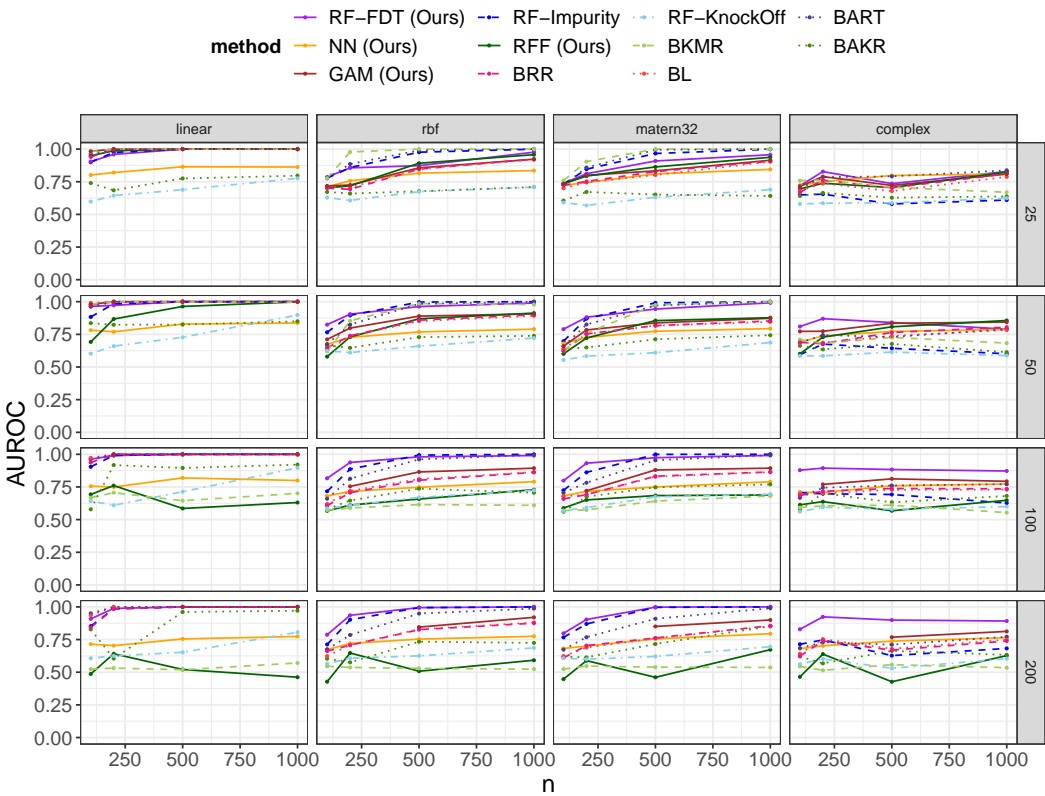

**Figure 6:** AUROC scores for **synthetic-mixture** data. FDT generally outperforms other methods in most of the data settings in relatively higher dimension ($d = 50, 100, 200$). Knockoff with random forest statistics produce lower AUROC scores than in **synthetic-continuous**, even in linear data settings. Additive models BRR, BL and GAM have mediocre scores under the nonlinear settings. Some model (e.g., GAM) reports missing result in $n > p$ setting due to the restrictions of their implementations.

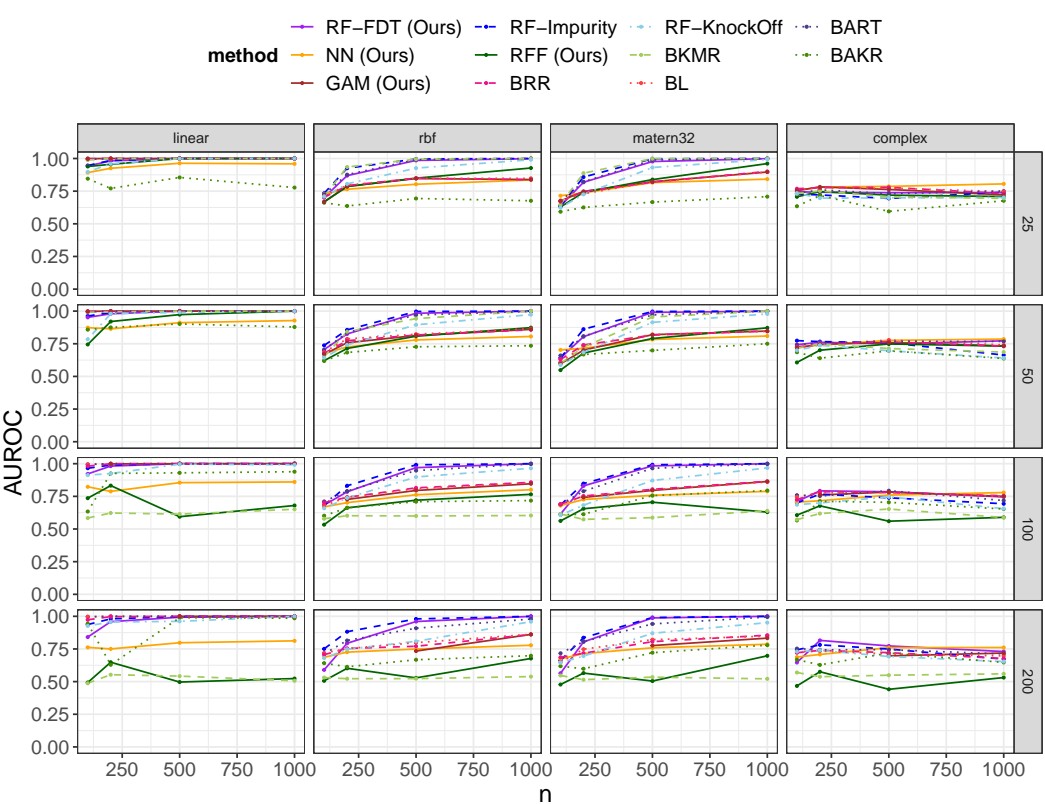

**Figure 7:** AUROC scores for **synthetic-continuous** data. FDT generally outperforms other methods in most of the data settings, with BKMR as the comparable one when $d = 25$. However, BKMR performs poorly in higher dimension. Tree-based methods RF, BART and Knockoff with random forest statistics have high AUROC scores. Additive models BRR, BL and GAM have mediocre scores under the nonlinear settings. Some model (e.g., GAM) reports missing result in $n > p$ setting due to the restrictions in their implementations.

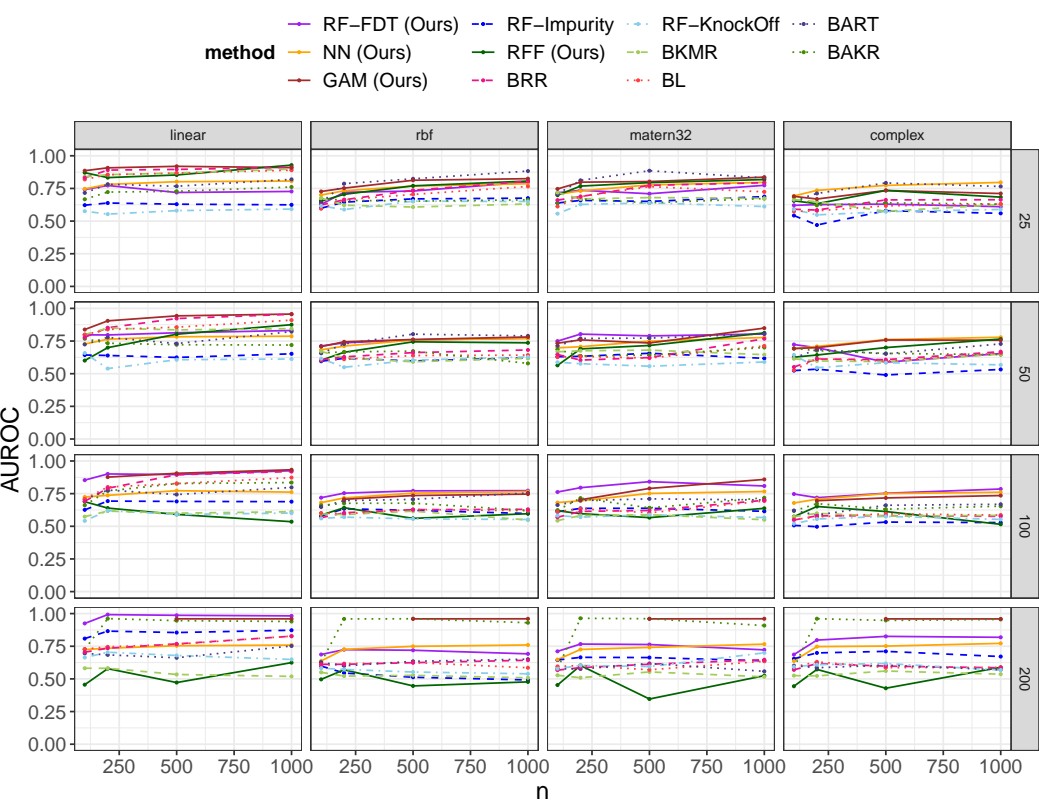

**Figure 8:** AUROC scores for **adult** data. In relatively low-dimension settings ($d = 25, 50$), the standard methods such as BART has higher AUROC scores than FDT. However, when the dimension is higher ($d = 100, 200$), FDT performs better consistently. Some model (e.g., GAM) reports missing result in $n > p$ setting due to the restrictions in their implementations.

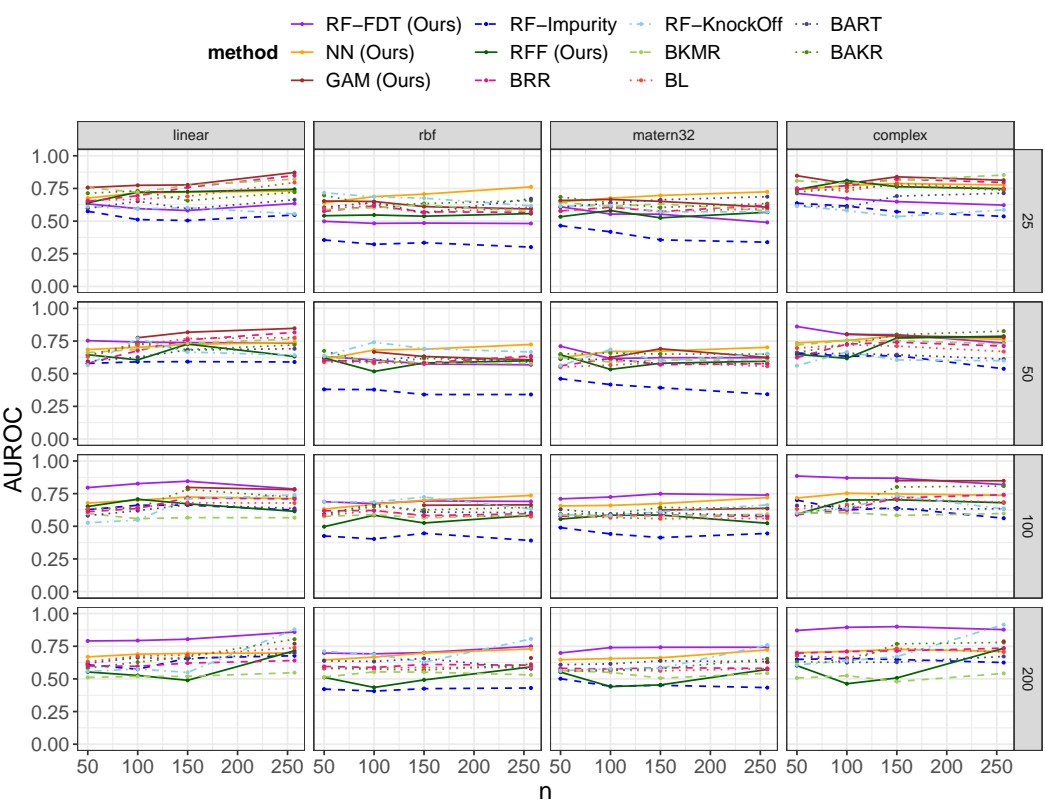

**Figure 9:** AUROC scores for **heart** data. In relatively low-dimension settings ($d = 25, 50$), the standard methods such as BART has higher AUROC scores than FDT. However, when the dimension is higher ($d = 100, 200$), FDT performs better consistently. Some model (e.g., GAM) reports missing result in $n > p$ setting due to the restrictions in their implementations.

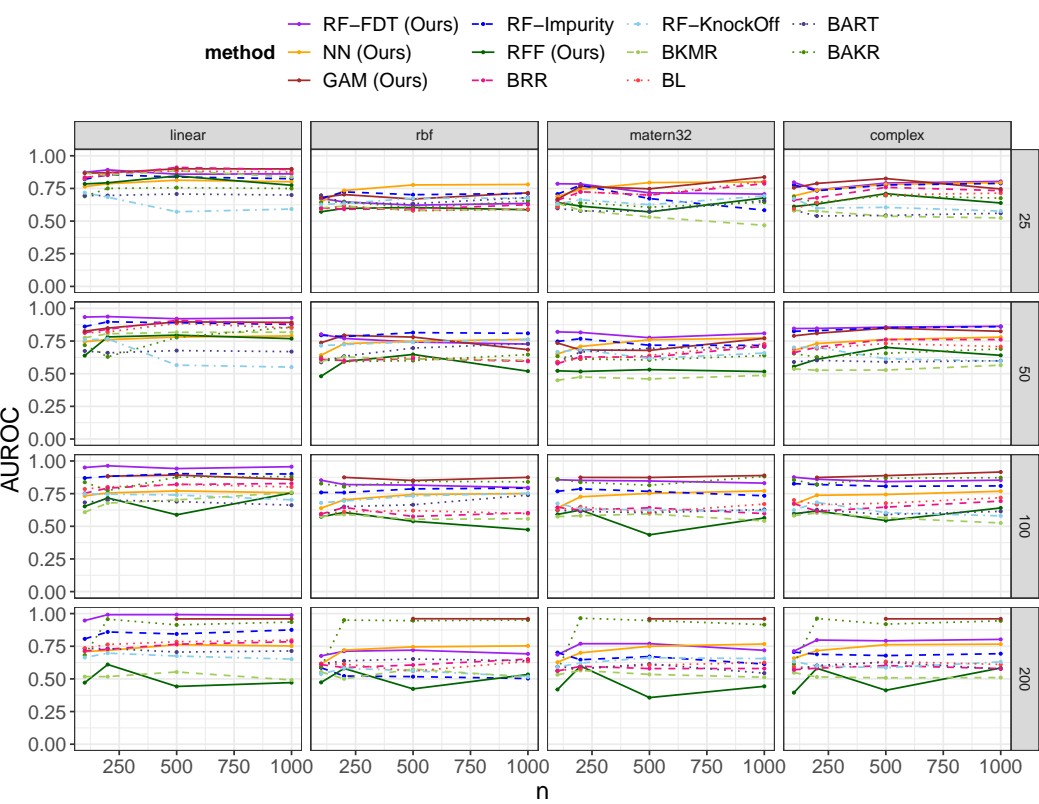

**Figure 10:** AUROC scores for **mi** data. In low-dimension setting ($d = 25$), the standard methods such as BART has higher AUROC scores than FDT. However, when the dimension is higher ($d = 50, 100$), FDT and GAM perform better consistently. Some model (e.g., GAM) reports missing result in $n > p$ setting due to the restrictions in their implementations.

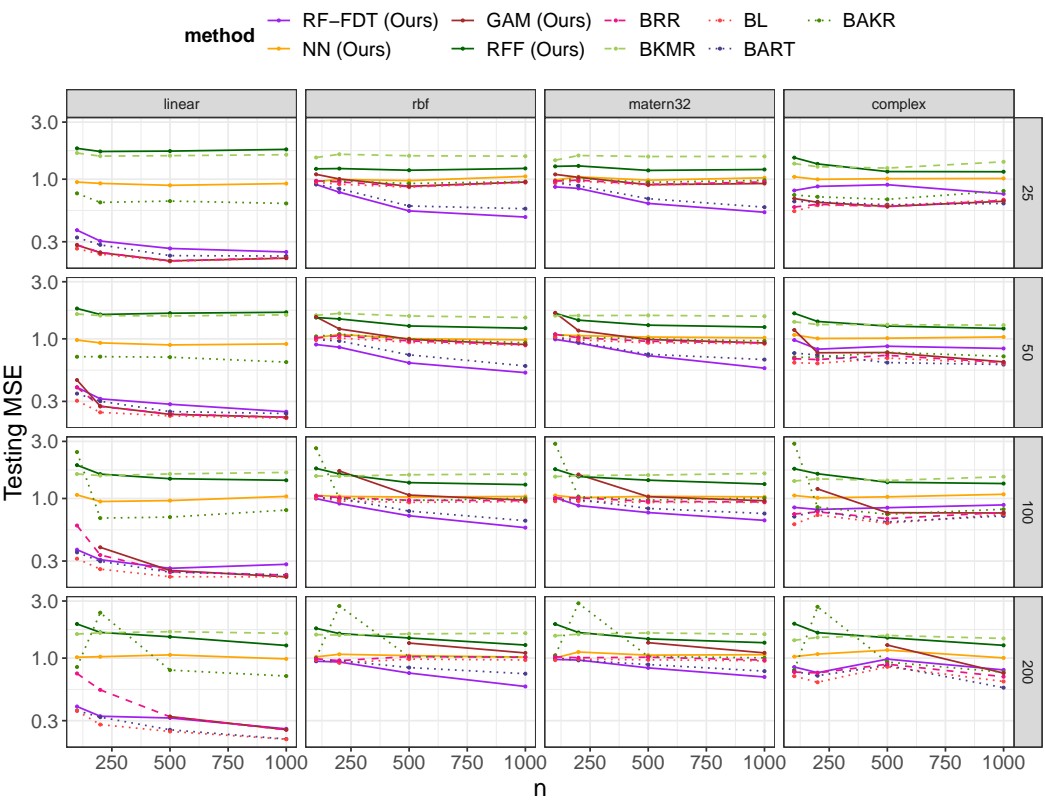

**Figure 11:** Testing MSE for **synthetic-mixture** data. FDT generally performs better or competitively with baselines, except in the `linear` case where **BL** unsurprisingly does best. **BKMR** consistently performs worse than other methods, except in the low data size, high dimension setting when **BAKR** performs worst. Some model (e.g., GAM) reports missing result in $n > p$ setting due to the restrictions in their implementations. Notice that this dataset contains a setting $n = p$, which can lead to the double descent phenomenon for some random-feature-based models [d'Ascoli et al., 2020].

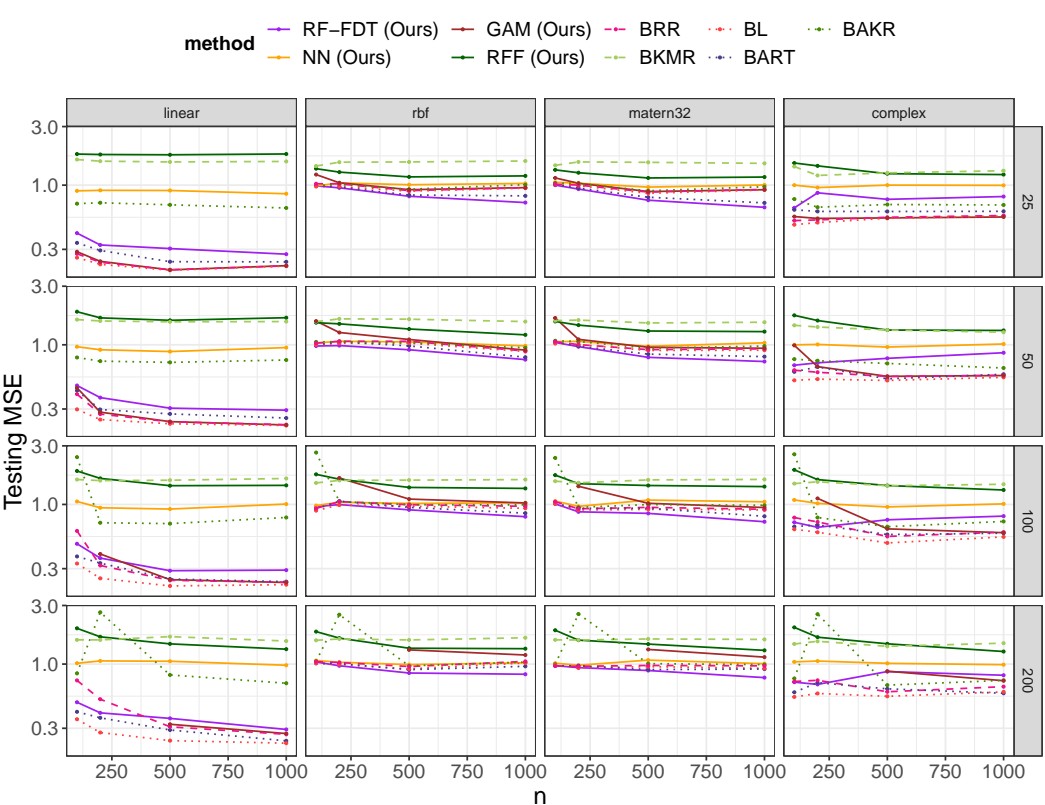

**Figure 12:** Testing MSE for **synthetic-continuous** data. A method will not be shown if they share the model fit with another method (**RF-impurity** and **RF-knockoff**), or if the method does not produce valid result due to small sample size (**GAM**).

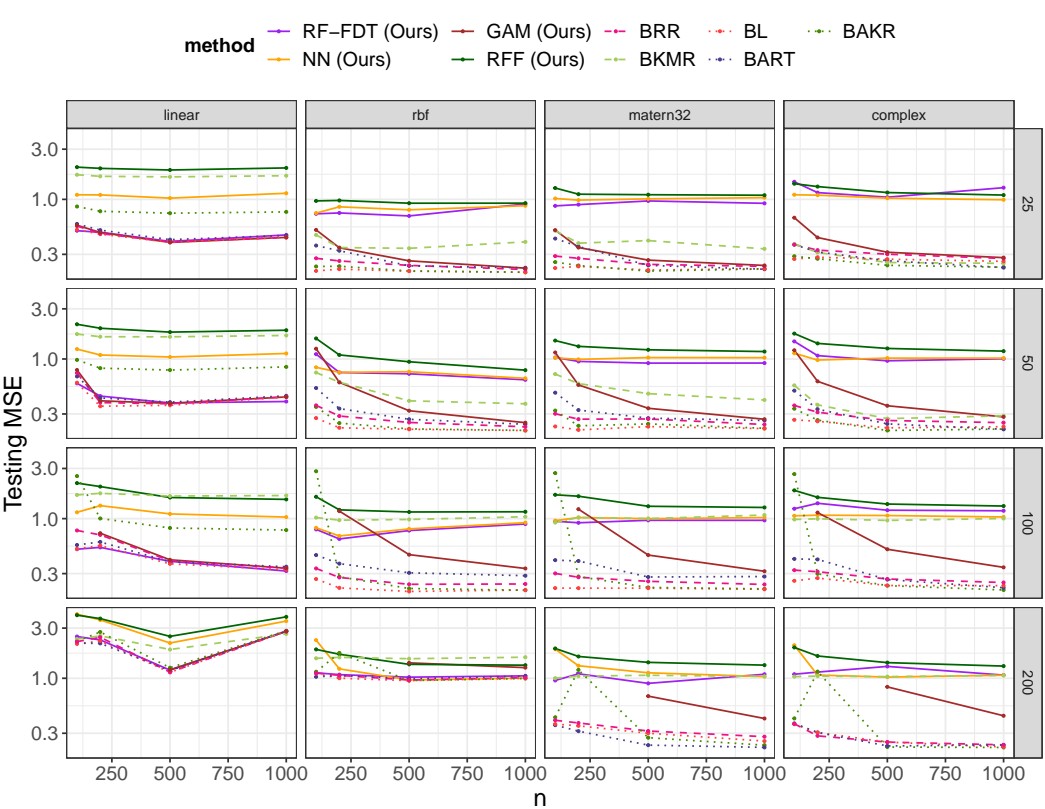

**Figure 13:** Testing MSE for **adult** data. A method will not be shown if they share the model fit with another method (**RF-impurity** and **RF-knockoff**), or if the method does not produce valid result due to small sample size (**GAM**).

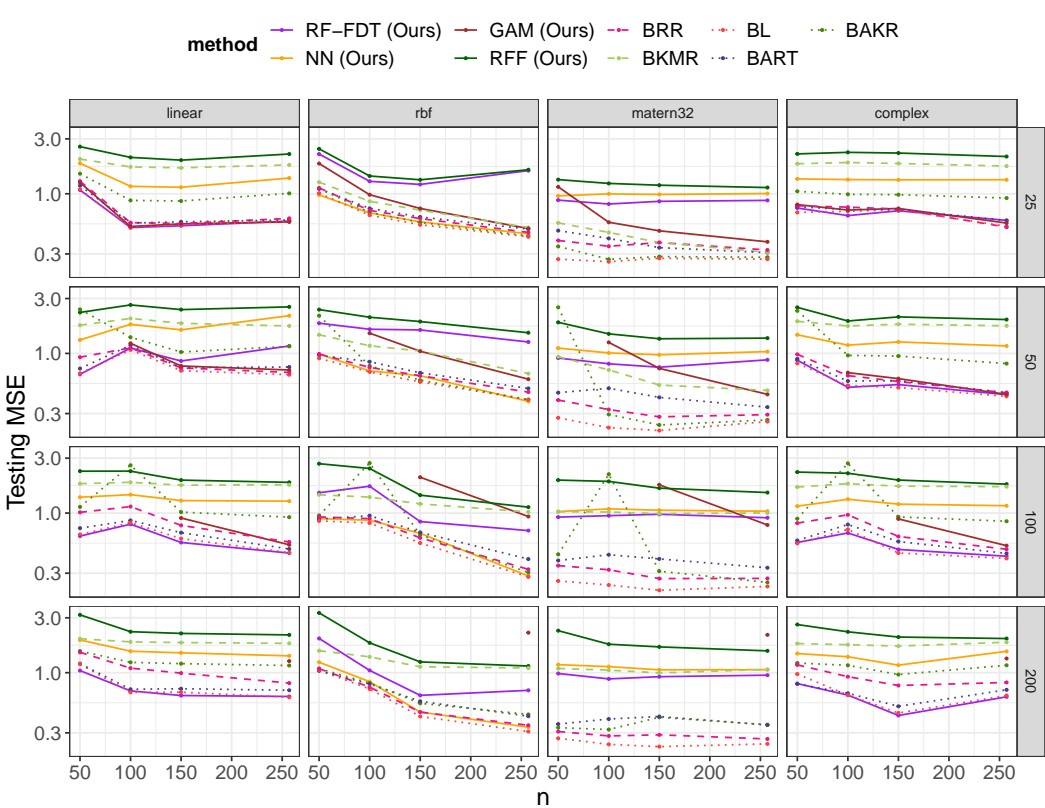

**Figure 14:** Testing MSE for **heart** data. A method will not be shown if they share the model fit with another method (**RF-impurity** and **RF-knockoff**), or if the method does not produce valid result due to small sample size (**GAM**). Notice that this dataset contains a setting $n = p$, which can lead to the double descent phenomenon for some random-feature-based models [d'Ascoli et al., 2020].

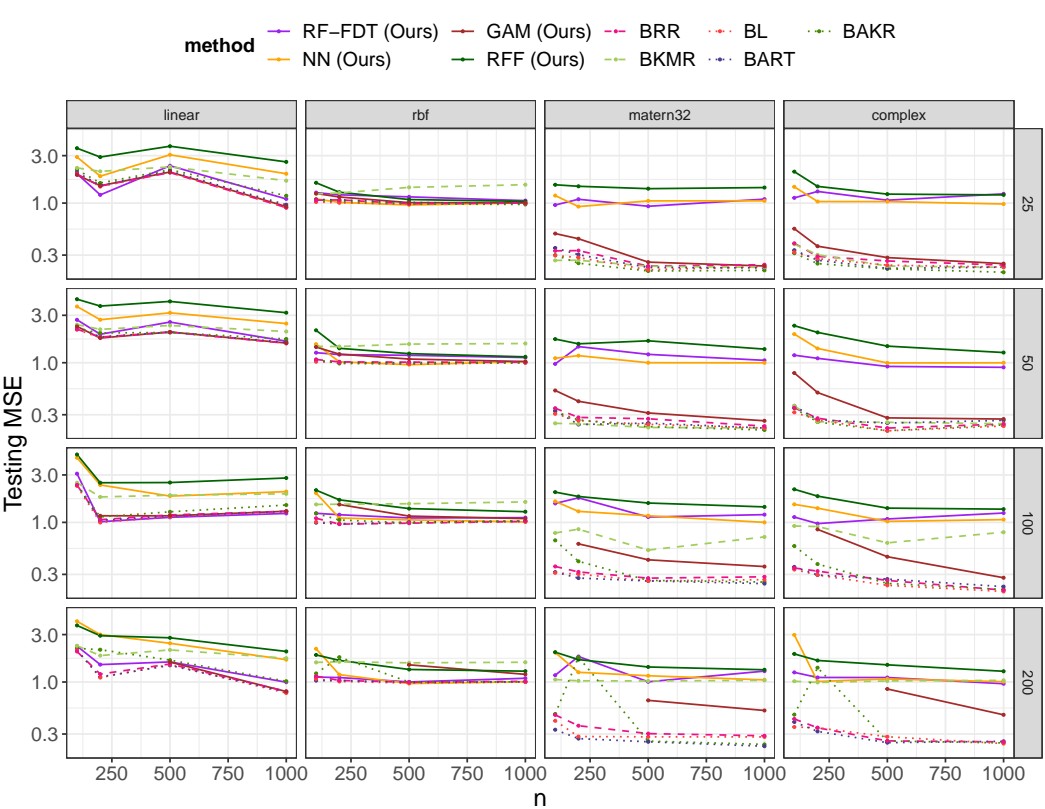

**Figure 15:** Testing MSE for **mi** data. A method will not be shown if they share the model fit with another method (**RF-impurity** and **RF-knockoff**), or if the method does not produce valid result due to small sample size (**GAM**).

# I  Additional Experiments: Regularization Path for Bangladesh birth cohort study

We propose a way to visualize the selection path that incorporates the uncertainty of variable importance scores. Specifically, we consider the posterior survival function $S(s) = P(\psi_j > s), j = 1, \ldots, d$ for increasing $s$ starting from 0. Larger value of $S(s)$ indicates larger probability of that certain feature being relevant. This is analogous to the regularization path under the LASSO method. However, our approach incorporates posterior uncertainty, and does not require repeated model fitting at different levels of regularization strength Mairal and Yu [2012].

We apply this to Bangladesh birth cohort study [Kile et al., 2014] (a well-established dataset in the environmental health literature), where we fit models to learn the association between infant's neural development scores and key environmental factors such as hospital location (`clinic`), sex (`sex`), levels of macro nutrient intake (`prot`, `fat`, `carb`, `fib`, `ash`) and levels of measured concentration of environmental toxins in body fluids (`as_ln`, `mn_ln`, `pb_ln`), while controlling for other socio-economic and biological factors (family income, parent education levels, etc). In general, the level of macro-nutrient intake (in particular fiber and protein) indicates a child's general nutrition status (i.e., whether he/she is eating well), and is known to be positively associated with neural development. On the other hand, the existing studies in the Bangladesh population have established a neurotoxic effect between arsenic exposure (i.e., `as_ln`, through drink water) on the early-stage cognitive development [Hamadani et al., 2011], as well as weak but significant effect of the joint mixture of other environmental toxins (manganese (`mn_ln`) and lead (`pb_ln`))) [Gleason et al., 2014, Valeri et al., 2017]. Furthermore, due the fact that the model has already controlled for biological and socio-economic confounding factors, non-nutrient-related factors such as hospital location and sex should not have a significant effect on the children's neural development status.

The result of variable importance estimation is shown in Figure 16, where we plot the posterior survival function $P(\psi_j > s)$ for $s \in (0, 1)$, and compare it to the survival function under *Bayesian Approximate Kernel Regression* (**BAKR**), *Bayesian Ridge Regression* (**BRR**), *Bayesian LASSO* (**BL**), and also the frequentist LASSO regularization path under the GAM model. We normalized all variable importance scores within the range $(0, 1)$. As a result, the variable selection performance is indicated by the relative magnitudes of the area under the curve for each variable (and not by the absolute magnitude due to the normalization).

As shown in Figure 16, the top variables selected by our method (FDT) correspond well with existing conclusions in the literature: it correctly picked up the larger impact of macro-nutrients (in particular, fibre, fat and protein) and smaller but still significant effects of environmental toxins (arsenic, manganese and lead), also notice that it ranked known non-causal factors such as hospital location and sex to be the lowest. In comparison, the linear methods (**GAM**, **BRR** and **BL**) all incorrectly reported high effect from hospital location on children's neural developement outcome (likely due to their restrict model form), while the nonlinear model (BAKR, based on RBF kernel) did not properly pick up the effect of environmental toxins.

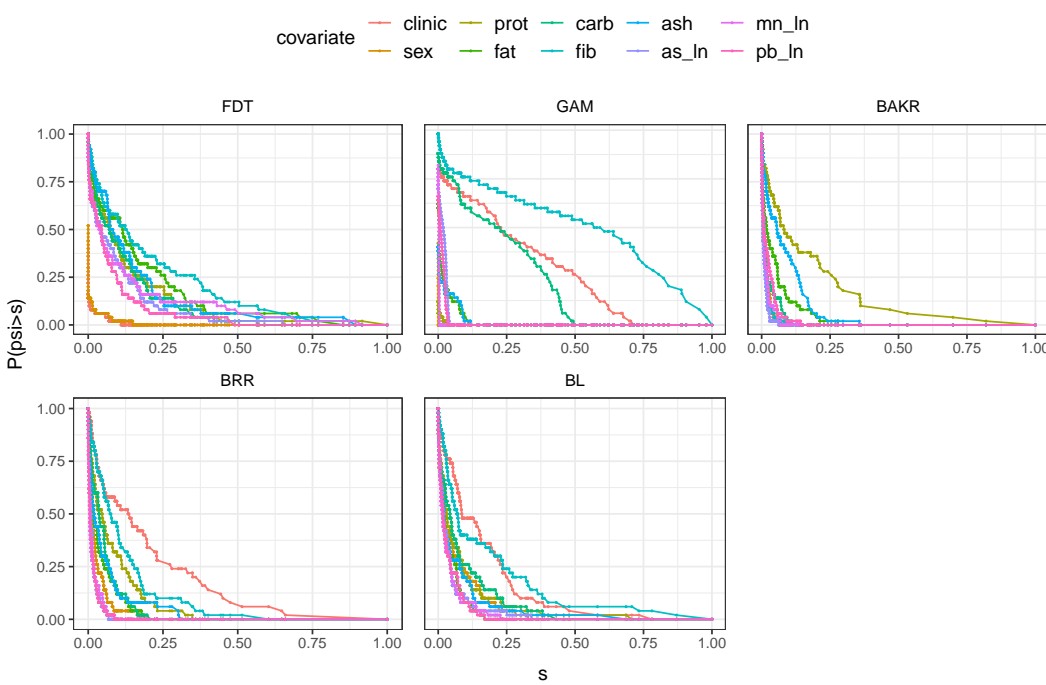

**Figure 16:** Regularization path for Bangladesh birth cohort study. The top variables selected by our method (FDT) correspond well with established toxicology pathways in the literature.

## J  Detailed Comparison with Related Work

Liu (2021) [Liu, 2021] is a closely-related work that derives posterior concentration and BvM result for the partial-derivative estimator of variable importance under Bayesian neural networks. Comparing to the current work, [Liu, 2021] focuses narrowly on the Deep Bayesian neural network model, while our work establishes a general set of conditions that is applicable to a much wider classes of machine learning models (GAMs, decision trees, random-feature models, DNNs, ensemble). More specifically, [Liu, 2021]'s result relies on specific assumptions on hidden weights and network width of a ReLU network (Section 2 and Assumption 1), and cannot be generalized straightforwardly to the wider class of models. In comparison, this work established much more generalized conditions in terms of the Lipschitz condition of predictive function, and the effective dimensionality of the model space (see Appendix B.1 for a summary), and we consider in detail the implication of these assumptions for different model classes (Appendix E). On the empirical side, [Liu, 2021] only studied model performance on simulated data sampled i.i.d. from fixed simple distributions, while our work investigated performance of the metric on a wide range of model classes and on non-simulated, realistic data distributions and on correlated, discrete variables (see Section 4 and Appendix G). Furthermore, some of our model variants (e.g., **FDT**) strongly outperforms neural network in almost all situations.

He et al. (2021) [He et al., 2021] is another closely-related work that employ partial derivative-based kernel method to realize variable selection. [He et al., 2021] shares similarity with this work in that we both consider gradient norm as the variable importance estimator, but with drastically different focus in theoretical results and empirical investigations. Specifically, [He et al., 2021] (1) studies a frequentist variable selection approach based on thresholding (2) does not address uncertainty in the variable importance estimators, and (3) focuses on classical, non-adaptive Gaussian process kernels (e.g., linear, quadratic, RBF kernels, see Sections 4 and 6 of [He et al., 2021]) and does not consider generalization across modern ML models. In comparison, our framework (1) focuses on Bayesian estimation of variable importance, (2) explicitly incorporates model uncertainty and derives its theoretical guarantee, (3) considers and empirically investigates the generalization of the approach across wide range of modern ML models, in particular tree-ensemble models whose compatibility to kernel-based variable importance method is not obvious.

# K   Further Discussion of Limitations

Our proposed framework provides principled uncertainty quantification by performing exact Bayesian inference on the weights $\boldsymbol{\beta}$ of a feature map $\phi(\mathbf{x})$. We do not consider uncertainty in the feature map itself. This means, for example, that if the feature map is given by the last hidden layer of a neural network trained by maximizing the posterior, then our model class corresponds to the *neural linear model*. This model is different from a fully Bayesian neural network, which performs posterior inference also on the kernel hyperparameters (i.e., the hidden weights) Ober and Rasmussen [2019], Snoek et al. [2015], Thakur et al. [2021]. Likewise, the kernel induced by the featurized decision tree studied here does not consider uncertainty in the tree's partitioning process. However, as discussed in the method section (Section 2.1), this "linearity" of the model parameter does not impact the expressiveness of the GP model, since the basis functions $\phi = \{\phi_k\}_{k=1}^D$ themselves are nonlinear and are allowed to be updated as part of the learning process. At the same time, the fact that the full posterior inference is performed only with respect to $\boldsymbol{\beta}$ indeed places a limitation on the model's ability in uncertainty quantification, as the uncertainty in the model hyperparameters is not accounted for. Yet, this does not seem to be a **significant limitation in the method's empirical performance** (e.g., **FDT** outperforms **BART** in our experiments), although this point still merits further investigation in the future.

The theoretical results of the current paper assume fixed data dimension $d = O(1)$. However, this does not restrict the significance and practicality of our theoretical results. On the theoretical side, even for fixed $d$, a general framework to nonparametric Bayesian inference of feature importance is currently missing in the field. On the application side, the majority of the machine learning applications fall into the fixed dimension setting. For example, in vision tasks, we usually handle images with fixed dimensions (i.e. height, width, and number of channels); in language tasks, we handle sentences with fixed vocabulary size and maximum sentence length; and in tabular tasks, we often work with tables with fixed number of columns. Furthermore, as commented in Section 3.2, the posterior concentration of variable importance (Theorem 1) does not rely on this assumption in its proof. Therefore, posterior concentration can occur even for high-dimensional settings with $d = o(n)$. In contrast, the BvM theorem usually represents a much stronger type of convergence (i.e., convergence in the predictive CDF of the entire Bayesian predictive posterior) and usually requires a stricter set of conditions. This is especially true in our setting, where our BvM results consider the quadratic functional of the nonlinear model, rather than the predictive function of the model itself. To this end, we highlight that most of the modern BvM results in ML models are derived by assuming fixed dimension. This includes [Wang and Rocková, 2020] for Bayesian neural networks, [Burnaev et al., 2013, Yang et al., 2015] for Gaussian process models, [Rockova, 2020] for BART (i.e., Bayesian tree models), and finally [Castillo and Rousseau, 2015] for general semi-parametric models. Therefore, our setting is consistent with the modern literature in Bernstein von-Mises results for ML models. At the same time, it should be acknowledged that in recent years, there have been a few "high-dimensional" Bernstein von Mises results for specialized settings. The more recent ones include [Bontemps, 2011] for linear regression, and [Lu, 2017] for Bayesian inverse problems with nonlinear forward dynamics. However, extending these results to the *quadratic functionals* of a *general* nonlinear model is non-trivial and out-of-scope for our current work, but is an important and interesting direction of future theoretical investigations.

In our experiments, we focused on kernels based on tree ensembles, kernel methods and linear models. In the future, it would be worth expanding this framework to other model classes (e.g., MARS Friedman [1991]) and estimating the importance of interaction effects and higher-order terms. We would also like to apply this method to large-scale scientific studies (e.g., epidemiology studies based on extremely large EHR datasets) where an uncertainty-aware nonlinear variable importance estimation method is typically impossible due to challenges with scalability.