# OpenReview forum: "Towards a Unified Framework for Uncertainty-aware Nonlinear Variable Selection with Theoretical Guarantees"
_NeurIPS.cc/2022/Conference — NeurIPS 2022 Accept_

### Official Review · Reviewer_adBr · 2022-07-11

**Rating:** 7
**Confidence:** 4
**Soundness:** 3 good
**Presentation:** 3 good
**Contribution:** 3 good

**Summary:**

The paper develops a unified framework for uncertainty-aware nonlinear variable selection for machine learning models that can be represented by RKHS including linear models, neural networks, random forests, and tree ensembles. Two major theoretical results about the approach are derived, namely, posterior convergence of variable Importance and a Bernstein-von Mises Theorem verifying that posterior distribution of variance importance is statistically optimal.

The authors then apply and validate the approach on several synthetic and real datasets, and conclude that:

- there does not exist a model class that performs universally well across all data scenarios, highlighting the necessity of a unified framework
- good prediction translates to effective variable selection

**Questions:**

n/a

**Limitations:**

None noted

**Strengths And Weaknesses:**

### Strengths

- Significance:
    - The paper considers an important problem in machine learning: uncertainty-aware nonlinear variable selections across multiple classes of models. The multiple-model-class aspect is particularly of great interest to the machine learning community since in many practical applications, it is often unknown a priori which specific class of model is more appropriate for the data.
- Quality and Clarity:
    - In general, the paper is well-written and well-organized. Discussions about pre-existing literature and related works are sufficient (with some caveats detailed below).
    - The theoretical analyses of the work are rigorous and of high quality.
    - One of the stronger points of the paper is that it provides a very careful treatment of the non-trivial gradient function assumption and studies in detail the complications arising when the features are discrete or the model is non-differentiable. These issues are often neglected in the field but are unavoidable in studying a unified framework, and the paper’s efforts in this direction should be appreciated.

### Weaknesses

- Literature: One concern I have with the paper is its lack of direct discussions/comparisons with the work

     > J. Liu (2019). Variable selection with rigorous uncertainty quantification using deep Bayesian neural networks: Posterior concentration and Bernstein-von Mises phenomenon.

   cited as [45] in the manuscript. [45] is mentioned in the paper with minor references (correlated features, simpler conditions), but I believe it’s hard to ignore the similarities between the two works (the same measure of feature importance, similar approaches, emphasis and questions, same theoretical results, even the titles are somewhat similar).

  More importantly, the two papers lead to two contrasting conclusions:
    - the paper observes that good prediction translates to effective variable selection
    - [45] hinted that a powerful model alone is not sufficient to guarantee effective variable selection

    Given the connections (and the fact that the authors are aware of the work by Liu), a thorough comparison in both theoretical analyses and (perhaps) experiments would be more appropriate.
- Quality:
  - The paper conflates the two concepts of variable importance and variable selection, and this may lead to an incorrect interpretation of the results.
      - Mathematically, posterior concentration of variance important and Bernstein-von Mises phenomenon do not translate to efficient/consistent variable selection. For example, for standard settings of linear models, ordinary least square satisfies both of the above properties but is not selection consistent/efficient, thus necessitating the use of lasso-based methods. Thus Theorem 1 and 2 do not support the claim that “good prediction translates to effective variable selection” as the authors stated in Line 280.
      - In practice, to quantify the efficiency of variable selection procedures, a good measure of variable importance and a rigorous decision rule is equally important, and AUROC (which integrates over threshold) is not an appropriate measure in these comparisons. This might be the reason why the conclusions of the paper and [45] (which uses F1 scores on three different modes of thresholding) are pointing in two different directions.
  - Since the paper attempted to provide a framework that works across models, they seem to avoid directly stating the assumptions of the analyses. For example, the feature map could be differentiable, non-differentiable, or even discontinuous and there seem to be separate treatments for each case. However, this leads to a very difficult reading of the work, and it seems impossible to verify from the main text whether a general model would fit into the framework. I suggest adding a list of assumptions on the feature map, and the parameter spaces of the models in the mathematical descriptions.
  - The decisions of evaluating the methods on tabular data only, and thus remove neural network models from the comparisons are contradictory to the goal of providing a unified framework of variable selection. Some additional evaluations, even on a synthetic dataset, perhaps in comparison with Liu (2019) would strengthen the manuscript.
- Significance
    - The paper assumes that the number of significant features (d) is fixed while data (n) increases. This setting is significantly more restrictive than typical results about posterior concentrations in the field for both generalized linear models and neural network models.
    - In essence, the models considered in this work need to be compatible with an RHKS and have to be linear in terms of the parameters. As pointed out by the authors, this places a significant restriction on the application of the framework. For example, for neural network models, this means all layers of the network except the last (linear) one have to be fixed, and the same limit applies for tree-based models. I believe this should be acknowledged more directly in the title, the abstract, and the introduction.


#####Post-rebuttal comment####

The authors have addressed all of my comments with their extensive efforts with the revision.

---

> ### Author Response · Authors · 2022-08-02
> **Response to Reviewer adBr**
>
> * **“One concern I have with the paper is its lack of direct discussions/comparisons with the work Liu (2021)...”**
>
> Thanks for highlighting Liu (2021)’s work that is worth a more explicit comparison. To this end, we (1) added further description of Liu (2021) in the introduction, (2) added the neural network result to the experiments, and (3) added a more detailed comparison with Liu (2021) to the Appendix J.
>
> Briefly:
>
>   * On the theoretical side, both works prove posterior concentration and a BvM result for variable importance. However, Liu (2021) focuses narrowly on a deep Bayesian neural network model, while our work establishes a general set of conditions that are applicable to a much wider class of machine learning models (GAMs, decision trees, random-feature models, DNNs, ensembles). This is a core contribution of our work. More specifically, Liu (2021)'s result relies on specific assumptions on the hidden weights and width of a ReLU network (Section 2 and Assumption 1) and cannot be generalized in a straightforward manner to a wider class of models. In comparison, we established much more generalized conditions in terms of the Lipschitz condition of predictive function and the effective dimensionality of the model space (see Appendix B.1, and associated discussion in Section 3.2.). We also discussed in detail the implication of these assumptions for different model classes (Appendices E.1 and E.2).
>
>   * On the empirical side, Liu (2021) only studied model performance on simulated data sampled i.i.d. from fixed simple distributions, while our work investigated performance of the metric on a wide range of model classes and on non-simulated, realistic data distributions and on correlated, discrete variables (see Section 4 and Appendix G). Furthermore, some of our model variants (e.g., FDTs) strongly outperforms neural networks in almost all situations.
>
> We have added this comparison to Appendix J.
>
> * **“More importantly, the two papers lead to two contrasting conclusions…”**
>
> Thanks for highlighting this important point regarding the seemingly contrasting conclusions between the two papers. This point is worth further elaboration. These two statements are in fact complementary and both correct: Liu (2021) states that, fixing the model class, the choice of variable importance metric matters for variable selection. While our paper (specifically, Theorem 1, and associated discussion in lines 243-258 of the updated manuscript) states that, fixing the variable importance metric $\psi_j$, the choice of model classes matters for variable selection. **This point of ours is in fact also stated in Liu (2021) (see their Section 3.1, "In other words, good performance in prediction translates to good performance in learning variable importance.").** Together, these two works state that a variable selection method's performance is impacted by BOTH the choice of model class AND the choice of variable importance metrics, and the practitioner should be mindful about the choice of both. Thanks again for highlighting this, we have added additional descriptions in the Appendix (Section J) and in the Conclusion section to highlight this point.
>
> * **“The paper conflates the two concepts of variable importance and variable selection, and this may lead to an incorrect interpretation of the results.”**
>
> Thanks for highlighting this important point. Yes, the focus of our work is the estimation of variable importance. Consequently, our conclusion shall be restated as “good prediction translates to effective variable importance estimation.” We have updated the manuscript throughout to reflect this. However, as discussed in the last answer, we would like to emphasize again that **our conclusion does not conflict with the variable selection result from Liu (2021), as their work also stated exactly the same theoretical conclusion ("...good performance in prediction translates to good performance in learning variable importance.." third paragraph of Section 3.1 of that paper).**

---

> > ### Author Response · Authors · 2022-08-02
> > **Part 2: Response to Reviewer adBr**
> >
> > **“Since the paper attempted to provide a framework that works across models, they seem to avoid directly stating the assumptions of the analyses…”**
> >
> > Thanks for highlighting a point that could be stated more clearly in the paper.
> > Indeed, to ensure the generality of our framework, we focused our assumptions on the mathematical property of the fitted functions ( i.e., weak differentiability, the Lipschitz condition of the function map (condition (2) in Theorem 1) and the "effective dimensionality" of the function space (Condition (4) in Theorem 2), rather than specific conditions of the feature mapping. This choice is deliberate and grants our framework the ability to generalize across a wide range of models, which is a core contribution of our work.
> >
> > With that said, we agree with Reviewer adBr that it is beneficial to provide the reader a clear list of assumptions on feature map and parameter space in a centralized place. We have done so in Appendix B.1, and linked to it in the main text (end of Section 2.1. "Incorporating Modern ML Classes", and then beginning of Section 3.2. "Theoretical Guarantees'').
> >
> > **"remove neural network models from the comparisons are contradictory to the goal of providing a unified framework of variable selection.."**
> >
> > Thanks for this suggestion. We have added neural network results to all datasets in the Experiment Analysis section. As shown in Figure 1, under the tabular data modality, the neural network model underperforms GAM or tree ensemble models (i.e., FDT) while it outperforms RFF (previously called RFNN) models in predictive performance (which is sensible, since, respectively, neural network models are known to be less effective under tabular data and since RFF models are essentially a 1-layer neural networks with un-trainable input weights). Correspondingly, we also see that the neural network model underperforms GAM and FDT but outperforms RFF in variable importance estimation, which is consistent with our existing empirical observations O1-O3.

---

> > > ### Author Response · Authors · 2022-08-02
> > > **Part 3: Response to Reviewer adBr**
> > >
> > > **“The paper assumes that the number of significant features (d) is fixed while data (n) increases...”**
> > >
> > > Thanks for this comment. We would like to highlight that the conditions in our paper are used to satisfy both theorems, (1) posterior concentration and (2) Bernstein von-Mises phenomenon. For posterior concentration, our result can be satisfied in much less restrictive conditions (i.e., $d=o(n)$), since the the uniformity of posterior convergence of the $\Psi_j(f)$’s for $j=1,\dotsc,d$ does not rely on a fixed $d$ (we have add discussion to Section 3.2, and also a technical remark (Remark 4) to the proof of Theorem 1 in Appendix C highlighting this point).
> > >
> > > On the other hand, the Bernstein von-Mises theorem usually represents a much stronger sense of convergence (i.e., convergence in the predictive CDF of the entire Bayesian predictive posterior) and usually requires a stricter set of conditions. This is especially true in our setting, where our BvM results consider the quadratic functional of the nonlinear model, rather than the predictive model itself. To this end, we highlight that most of the modern results of Bernstein von Mises theorems in ML models are derived by assuming a fixed dimension. This includes [Wang and Rockova (2021)](https://proceedings.mlr.press/v108/wang20b/wang20b.pdf) for Bayesian neural networks, [Burnaev et al (2013)](https://www.researchgate.net/publication/260295885_The_Bernstein-von_Mises_theorem_for_regression_based_on_Gaussian_Processes) and [Yang et al (2015)](https://arxiv.org/pdf/1503.04493.pdf) for Gaussian process models, [Rockova (2020)](http://proceedings.mlr.press/v119/rockova20a/rockova20a.pdf) for BART (i.e., Bayesian tree models), and finally [Castillo and Rousseau (2015)](https://projecteuclid.org/journals/annals-of-statistics/volume-43/issue-6/A-Bernsteinvon-Mises-theorem-for-smooth-functionals-in-semiparametric-models/10.1214/15-AOS1336.full) for general semi-parametric models. Therefore, our setting is consistent with the modern literature in Bernstein von-Mises results for ML models.
> > >
> > > At the same time, we do acknowledge that in recent years, there have been a few "high-dimensional" Bernstein von Mises results for specialized settings. The more recent ones include [Bontemps (2011)](https://projecteuclid.org/journals/annals-of-statistics/volume-39/issue-5/Bernsteinvon-Mises-theorems-for-Gaussian-regression-with-increasing-number-of/10.1214/11-AOS912.full) for linear regression, and [Lu (2017)](https://arxiv.org/abs/1706.00289) for Bayesian inverse problems with nonlinear forward dynamics. However, extending these results to the quadratic functionals of a general nonlinear model is non-trivial and out-of-scope for our current work. We will highlight this explicitly in the future directions.
> > >
> > > Finally, we would like to highlight that the fixed $d$ setting does not restrict the significance and practicality of our theoretical results. On the theoretical side, even for fixed $d$, a general framework for nonparametric Bayesian inference of feature importance is currently missing in the field. On the application side, the majority of the machine learning applications fall into the fixed dimension setting. For example, in vision tasks, we usually handle images with fixed dimensions (i.e., height, width, and number of channels); in language tasks, we handle sentences with fixed vocabulary and maximum sentence length; and in tabular tasks we often work with tables with a fixed number of columns.
> > >
> > > We have added this discussion to the discussion for posterior concentration result in Section 3.2, and then in the future direction at the end of the paper.

---

> > > > ### Author Response · Authors · 2022-08-02
> > > > **Part 4: Response to Reviewer adBr**
> > > >
> > > > **“In essence, the models considered in this work need to be compatible with an RKHS and have to be linear in terms of the parameters…”**
> > > >
> > > > Thanks for this comment. We would like to clarify that although our framework focuses on quantifying uncertainty in the weights of the feature map (as discussed in the first paragraph of the limitations section), **it does not impose a significant restriction on the effectiveness of the resulting model, since the models are allowed to fit the kernel hyperparameters, for example via MAP inference (e.g., hidden weights of the neural network kernel, or space partition rules of a decision tree kernel).** In the probabilistic neural network literature, this practice of performing full Bayesian inference in the last layer (i.e., the weights of the feature map) while fitting hidden layers via MAP inference has a long history ([Salakhutdinov and Hinton, 2007](https://proceedings.neurips.cc/paper/2007/file/4b6538a44a1dfdc2b83477cd76dee98e-Paper.pdf)) and has been the key approach behind some current state-of-the-art methods in deep learning uncertainty ([Wilson et al 2016](https://proceedings.mlr.press/v51/wilson16.html), [Liu et al 2020](https://papers.nips.cc/paper/2020/file/543e83748234f7cbab21aa0ade66565f-Paper.pdf), [van Amersfoort 2021](http://bayesiandeeplearning.org/2021/papers/28.pdf)). As a result, our framework does not impose a significant restriction on the practical effectiveness of the fitted model (e.g., in terms of generalization performance). For example, in the experiment section, we showed that our method is fully compatible with a fitted random forest model, enabling an already practically effective model to provide much improved variable selection performance.
> > > >
> > > > On the other hand, we do acknowledge that our framework does not yet account for the uncertainty in hyperparameter estimation (as mentioned in the limitation section). However, as we have commented in lines 434-441, this theoretical limitation does not discount the practical effectiveness of our approach. For example, as shown in Figure 1, our method based on random forest (FDT) strongly outperforms the BART model, which is a Bayesian random forest model that accounts for uncertainty in the partitioning of the feature space.

---

> > ### Comment · Reviewer_adBr · 2022-08-03
> > **(comments addressed)**
> >
> > I want to thank the authors for a (very) detailed response to my comments. I am impressed by the author's efforts (in theory, contexts and simulations) to revise the manuscript and make it more rigorous. I'm glad to say that all of my main concerns about the paper are well addressed.
> >
> > I still have some reservations about the last point (linearity in parameters), but overall I think the contributions of the paper outweighs its limitations. I intend to revise my rating up to Weak-Accept or Accept (depending on further discussions with other reviewers).

---

> > > ### Author Response · Authors · 2022-08-09
> > > **Response to Reviewer adBr**
> > >
> > > We thank adBr for the thoughtful response. We are glad to hear that all concerns are addressed and the intended rating is now weak acceptance (6) / acceptance (7).
> > >
> > > Thanks again for raising the point about model linearity in parameters. We would like to highlight that although our framework focuses on Bayesian inference with respect to weight parameters. It does not require other parameters to be fixed to a null / random value, and allows them to be optimized (e.g., via MAP inference) as well. To further address this point, we added additional language to the manuscript for further discussion, and provided concrete algorithm examples in Appendix A.4 to show how kernel hyper-parameters and GP posteriors can be learned jointly in a coherent procedure. We hope these additional details add further clarity around this point. As before, all changes are highlighted in blue:
> > >
> > > 1. **Added Language**: In Introduction (Section 1, lines 53-56), we added language to explicitly acknowledge that the RKHS-based approach implies that the full Bayesian inference is done on the weight parameters of the basis functions, which are linear with respect to the predictive function. In the method section (Section 2.1), we also highlight that the basis functions can be updated (e.g., via MAP inference) as part of the learning procedure (lines 117-121 and lines 137-139), and provide further detail in Appendix A.4. (see below). Finally, in Limitation (lines 426-431), we elaborated the different implications of this practice to model's two aspects of performance: prediction and uncertainty quantification.
> > >
> > > 2. **Additional Algorithmic Examples**: In Appendix A.4, we supply two concrete meta-algorithms to illustrate how joint learning of kernel hyperparameters and GP posterior can be done for classic GP kernels, and for advanced kernels like neural networks and decision trees. This includes (1) the alternative inference procedure, where the procedure alternate between GP posterior computation and hyperparameter updates (e.g.,  learn the bandwidth parameter of RBF kernel by minimizing it with respect to marginalized likelihood. [Rasmussen and Williams (2006)](http://gaussianprocess.org/gpml/chapters/RW.pdf), Section 5.4.). (2) the "pretraining" procedure, where the model first learn the hyperparameters using a separate method before computing the GP posterior (e.g., learn the neural network hidden weights via multi-epoch SGD, and then compute the GP posterior in the last epoch. See [Liu et al (2020)](https://arxiv.org/pdf/2006.10108.pdf), Algorithm 1). Both of these algorithms are well-established procedures in the literature.

---

### Official Review · Reviewer_2frY · 2022-07-11

**Rating:** 6
**Confidence:** 2
**Soundness:** 3 good
**Presentation:** 3 good
**Contribution:** 3 good

**Summary:**

This paper proposes an uniﬁed framework for nonlinear variable selection that incorporates uncertainty in the prediction function and is compatible with a wide range of machine learning models (e.g., tree ensembles, kernel methods, neural networks, etc).  The authors (1) provide a principled approach for quantifying variable selection uncertainty by deriving its posterior distribution, and (2) show that the approach is generalizable even to non-differentiable models. Rigorous Bayesian nonparametric theorems are derived to guarantee the posterior consistency and asymptotic uncertainty of the proposed approach. Extensive simulations and experiments verify the effectiveness of the proposed method.

**Questions:**

+  In line 76, the data-generating function f_0 is assumed as a ﬂexible nonlinear function that resides in a reproducing kernel Hilbert space (RKHS).  How restrictive is this assumption? in other words, does this assumption cover most real data generation functions.
+  In Theorem 1, what is the specific convergence rate of $\epsilon$ mentioned in condition (1) (line 203).
+  He et al. [1] also employ the partial derivative-based kernel method to realize variable selection. It would be appreciated if the authors could explain the differences between theirs and the advantages of this paper.

Ref:
[1] He et al. SCALABLE KERNEL-BASED VARIABLE SELECTION WITH SPARSISTENCY.

**Limitations:**

All my concerns are presented in “Weakness” and “Questions”. Moreover, I do not have any concern on negative social impact.

**Strengths And Weaknesses:**

Strength:
+  The paper is well-written.

+ The proposed method is supported by theory and experiments.

Weakness:

The numerical experiments can be improved, e.g., considering larger input dimensions in the experiment.

---

> ### Author Response · Authors · 2022-08-02
> **Response to Reviewer 2frY**
>
> **“The numerical experiments can be improved, e.g., considering larger input dimensions in the experiment.”**
>
> We have added $d=200$ cases to all experiments. Overall the trends in the relative performance between models are similar to the case with $d=100$. In particular, the performance of kernel approaches (RFF and BKMR) deteriorate further in $d=200$ case. This is expected due to the classic kernel method's well-known issue of lacking the ability in learning adaptive feature representation, which consequently leads to the suffering of curse of dimensionality and unstable and suboptimal variable importance performance in high dimension ([Bach (2017)](https://jmlr.org/papers/v18/14-546.html )). On the other hand, we observe the double descent phenomenon for BAKR, a random-feature-based model ([d’Ascoli et al (2020)](https://proceedings.neurips.cc/paper/2020/hash/1fd09c5f59a8ff35d499c0ee25a1d47e-Abstract.html )). This is obvious in testing MSE figures. From these observations, we expect the trends for higher dimensions are similar, with the performances of RFF and BKMR deteriorating further and the double descent phenomenon for BAKR.
>
> **“How restrictive is this assumption [that f_0 belongs to an RKHS]? In other words, does this assumption cover most real data generation functions?”**
>
> This is a great question! The assumption that $f_0$ resides in an RKHS is not very restrictive in practice, since we place no restriction on the kernel (i.e., the model class) that induces this RKHS. For example, we could assume a standard kernel like the Matérn kernel, whose RKHS corresponds to the Sobolev spaces with a known degree of differentiability, a common target of nonparametric analysis ([Fasshauer and Ye (2011)](https://link.springer.com/article/10.1007/s00211-011-0391-2)). Or, we could assume the kernel is parameterized by a pre-trained deep neural network, allowing for rich, nonstationary structure in $f_0$ and a large approximation space per the universal approximation theorem ([Hornik et al (1989)](https://www.sciencedirect.com/science/article/abs/pii/0893608089900208)).  Since these models are empirically effective for real-world datasets, we believe the function space spanned by these model classes do cover realistic data generation mechanisms.
>
> **“In Theorem 1, what is the specific convergence rate of $\epsilon$ mentioned in condition (1)?”**
>
> Thanks for this question. $\epsilon_n$ is the posterior convergence rate of the prediction function of the machine learning model $f$ to the ground truth $f_0$. Depending on the choice of the model, the exact rate of convergence varies. See Section 2.2 (lines 164-166) for a list of references about convergence rates of well-established ML models, including random features, random forests, and neural networks. As a result, as long as we are given an ML model with a known convergence rate $\epsilon_n$ for the prediction function, we can use Theorem 1 to derive a convergence rate for its variable importance (as discussed in lines 259-262). We added additional sentences in Section 3.2 to highlight this important point.
>
> **“He et al. [1] also employ the partial derivative-based kernel method to realize variable selection. It would be appreciated if the authors could explain the differences between theirs and the advantages of this paper.”**
>
> Thanks for this question. He et al. (2018) and our work both consider the gradient norm as the variable importance estimator but with drastically different focus in theoretical results and empirical investigations. Specifically, He et al. (2018) (1) studies a frequentist variable selection approach based on thresholding (2) does not address uncertainty in the variable importance estimators, and (3) focuses on classic, non-adaptive Gaussian process kernels (e.g., linear, quadratic, RBF kernels, see Sections 4 and 6 of He et al.) and does not consider generalization across modern ML models.
>
> In comparison, our framework (1) focuses on Bayesian estimation of variable importance, (2) explicitly incorporates model uncertainty and derives its theoretical guarantees, (3) considers and empirically investigates the generalization of the approach across a wide range of modern ML models, in particular tree-ensemble models whose compatibility with kernel-based variable importance method is not obvious. We have added this comparison to the Introduction section and Appendix J.

---

> > ### Comment · Reviewer_2frY · 2022-08-08
> > **Thanks for your response!**
> >
> > I thank  the authors for addressing my questions well.   I will raise my score to acknowledge its development.

---

### Official Review · Reviewer_AEuT · 2022-07-14

**Rating:** 6
**Confidence:** 3
**Soundness:** 3 good
**Presentation:** 4 excellent
**Contribution:** 2 fair

**Summary:**

The importance of a variable in this paper is derived from the norm the corresponding partial derivative.
The authors use a Bayesian linear regression to capture the uncertainty of the model parameters.
Because of the form of feature represented method, the proposed method is independent to the model and thus can be applied to tree-based structures.

**Questions:**

- Even though the title uses the word "nonlinear", it seems that the variable selection part is still based on linear structure. If so, how could the model be extended to more complicated cases? For example, a multilayer neural network.
- The above question also reflected in the current experiments. Except for the synthetic experiments, the results on the real dataset is not very consistent. Why does the performance of RFNN is not good?
- Why does the points of GAM not show in the Figure 1?



**Limitations:**

Yes.

**Strengths And Weaknesses:**

Strength:
- The paper is well structured and written.
- The uncertainty is counted into the calculation of variable selection.
- Use the feature-based representation of Gaussian process to accelerate training and generalization on different machine learning models.
- The authors also added theoretical proofs to show the convergence and efficiency.

Weakness:
- The experimental results does not fully reflect the profoundness of the theory.
- The variable selection still relies on linear structure and could be hard to extend to more complex scenarios.
- It will be better if the legends of Figure 1 are enlarged.

---

> ### Author Response · Authors · 2022-08-02
> **Response to Reviewer AEuT**
>
> **“Even though the title uses the word "nonlinear", it seems that the variable selection part is still based on linear structure…”**
>
> Thanks for this question. We would like to highlight that our variable importance metric explicitly incorporates nonlinearity and is not restricted to a linear structure. Specifically, our variable importance metric $\psi_j(x)=||\frac{\partial}{\partial x^j} f||^2$ is defined as the partial derivative of the nonlinear function $f$ with respect to the $j$th input variable $x^j$ (Section 3.1. Equation 7), where the derivative function $\frac{\partial}{\partial x^j} f$ is propagated through the nonlinear feature map and reflects the variable importance learned under the nonlinear function $f$ .
>
> For the case of a deep neural network, the feature map is the last hidden layer of a (pre-trained) deep neural network (please see Appendix B, "(Deep) Neural Networks," for detailed mathematical description). As a result, the featurized Gaussian process model $f(x)=\beta^\top \phi(x)$ under the deep neural network kernel is identical to the original deep neural network, except the last-layer weights are estimated with full Bayesian inference (which is a rather established practice in probabilistic deep learning, see e.g., [Wilson et al 2016](https://proceedings.mlr.press/v51/wilson16.html), [Liu et al 2020](https://papers.nips.cc/paper/2020/file/543e83748234f7cbab21aa0ade66565f-Paper.pdf), or [van Amersfoort 2021](http://bayesiandeeplearning.org/2021/papers/28.pdf).
> In this example, the partial derivative metric $\frac{\partial}{\partial x^j} f(x) = \frac{\partial}{\partial x^j} \beta^\top \phi(x) = \beta^\top [ \frac{\partial}{\partial x^j}  \phi(x) ] $ incorporates the neural network gradient, which backpropogates through the entire nonlinear network, thus capturing its nonlinear structure. To make this point more explicit, in the experiment section, we have also added neural networks results to all benchmark settings.
>
> **“Except for the synthetic experiments, the results on the real dataset is not very consistent…”**
>
> Thanks for this comment. We would like to highlight that the goal of the experiment section is to validate the main conclusions of our theorems (Theorem 1: "under the gradient-norm variable importance metric, effective prediction translates to effective variable importance estimation" and Theorem 2: "gradient-norm metric is statistically efficient"). As discussed in the experiment section, (1) the ranking of each method's predictive performance is consistent with its ranking of variable selection performance using the gradient norm variable importance metric (consistent with Theorem 1) and (2) the tree-ensemble methods perform better using the gradient norm than other variable importance metrics (consistent with Theorem 2). Therefore, we believe the observations in the experiment are consistent with our theoretical conclusions.
>
> Thanks for highlighting the RFF result (previously named RFNN), which is worth additional description in the paper. Our experiment results show that in high dimension ($d=100$), RFF severely underperforms in prediction, and exhibits unstable variable selection behavior. This is not entirely surprising:  due to its lack of adaptivity in feature representation, the RFF model is known to suffer heavily from the curse of dimensionality and there performs poorly in high-dimensional generalization (i.e., in RMSE). As a result, its gradient-norm metric is less effective in capturing the true data-generating features, leading to poor variable selection results. This is consistent with our theoretical conclusion (Theorem 1). We have added corresponding language about RFFs to the experiment section (Section 4.1).
> Thanks again, please let us know if you feel there are other inconsistencies that we can address or clarify. We would be happy to continue the discussion to make the paper more clear.
>
> **“Why does the points of GAM not show in the Figure 1?”**
>
> We have added the explanation in the caption of Figure 1. Specifically, this GAM does not produce valid results due to small sample size, so the points in the cases of $n \leq d$ are not shown. We have added this explanation to the caption of Figure 1.

---

### Official Review · Reviewer_smu5 · 2022-07-14

**Rating:** 6
**Confidence:** 3
**Soundness:** 3 good
**Presentation:** 3 good
**Contribution:** 3 good

**Summary:**

This paper considers a method of evaluating the importance of variables of a (possibly) non-convex function f under a Bayesian perspective. Given a Gaussian process over a function set (including true function f) as a prior distribution, the evaluation measure for variable j is the expectation of magnitude of the partial gradient of f w.r.t. j, where the expectation is taken w.r.t. some distribution from which instances are drawn. The paper proves convergence results that the estimate of the measure converges to the true value when the sample size increases up to infinity.

**Questions:**

- Do you have any assumption on the distribution from which instances are drawn?

-Do you assume that the function is fully differentiable?

- I don't understand how to compute AUCROC. Please explain.

**Strengths And Weaknesses:**

Strength:
- Consistent approach from a Bayesian perspective


Weakness:
-  Theoretical results would be more convincing if finite sample analyses are possible.

---

> ### Author Response · Authors · 2022-08-02
> **Response to Reviewer smu5**
>
> **“Theoretical results would be more convincing if finite sample analyses are possible..”**
>
> Thanks for highlighting this point. It is indeed possible to obtain a finite-sample result.
> We stated Theorem 1 as an asymptotic result to be consistent with classical Bayesian nonparametric literature, where posterior concentration is typically stated in terms of an asymptotic rate ([Ghosal, Ghosh and van der vaart (2000)](https://www4.stat.ncsu.edu/~ghoshal/papers/bayes.pdf)).
> However, since our proof of Theorem 1 relies on the finite-sample Berstein inequality, given an ML model with a known finite-sample generalization bound $\epsilon_n$, it is straightforward to extend our proof of Theorem 1 to obtain a finite-sample error bound on the variable importance $\psi_j$. Specifically, the error bound will take the form $C \epsilon_n + n^{1/2} * 2\sqrt{2} * L * log(2/\eta)$, where the second term comes from the Berstein inequality (see proof of Theorem 1).
> We will add a detailed discussion regarding this point to Appendix C and a brief discussion to the end of Section 3.2 in the main text.
>
> **“Do you have any assumption on the distribution from which instances are drawn?”**
>
> Consistent with theoretical literature in Bayesian nonparametric regression, we assume the outcome $y$ has a Gaussian distribution surrounding the data-generating function $f_0$ (see Problem Setup, lines 79-86).  We do not make any assumptions about how the input features $x$ are generated or any additional assumptions on $f_0$ (other than what is already imposed by the arbitrary data-generating kernel $k_0$).
>
> **“Do you assume that the function is fully differentiable?”**
>
> Thanks for highlighting this point. As listed in the technical conditions (see Appendix B.1), we only assume the function is differentiable almost everywhere. This condition is much weaker than full differentiability and allows for models whose gradient is not well-defined in some places, e.g., a ReLU network whose gradient is not well-defined at 0. Also as discussed in Appendix B.1 and Appendix F, non-differentiable models like decision trees can be incorporated into our framework by applying a differentiable approximation, which in fact delivered the best empirical performance in the experiments.
>
> **“I don't understand how to compute AUCROC. Please explain.”**
>
> We treat each of the $d$ variables as an example from a binary classification problem and then compute the standard AUROC across these $d$ examples. In other words, we classify whether each variable relates to the ground truth response by thresholding its associated variable importance. The ROC curve then represents the tradeoff between the true positive rate and false positive rate (i.e., across the $d$ variables) as a function of the threshold and the AUROC integrates over all thresholds. For example, in the sklearn package, we can compute it as `roc_auc_score(y_true=if_causal_variable, y_score=variable_importance)`.

---

> > ### Comment · Reviewer_smu5 · 2022-08-08
> > **Thanks for your responses.**
> >
> > Thanks for your responses. One more minor technical comment. On AUC, I now understand what you mean. But, the figures need more information on the x-axis in Figure 1 (e.g., # of variables used with higher variables scores or whatever...).

---

> > > ### Author Response · Authors · 2022-08-09
> > > **Response to Reviewer smu5**
> > >
> > > We thank smu5 for noticing this. We have added descriptions of the x-axis and number of variables with high variable importance estimations in the caption of Figure 1. All changes are highlighted in blue.

---

### Author Response · Authors · 2022-08-02
**Summary Response**

We thank the Area Chair and all the reviewers for providing high-quality feedback for our paper. We were pleased to see that most reviewers found our paper to be well written ([AEuT](https://openreview.net/forum?id=crRhj1Y2wv&noteId=S91lBUuQRU7), [2frY](https://openreview.net/forum?id=crRhj1Y2wv&noteId=A2-PtGC3F1l), [adBr](https://openreview.net/forum?id=crRhj1Y2wv&noteId=O5eCvRo9Jw)) and that all reviewers appreciated the theoretical grounding of our results. In particular, we were pleased to see recognition of how our framework incorporates uncertainty into the variable selection process ([AEuT](https://openreview.net/forum?id=crRhj1Y2wv&noteId=S91lBUuQRU7)), handles multiple models classes, which is "of great interest to the machine learning community" ([adBr](https://openreview.net/forum?id=crRhj1Y2wv&noteId=O5eCvRo9Jw)); and provides a careful treatment of non-differentiable models and discrete features, issues which are "often neglected in the field but are unavoidable in studying a unified framework" ([adBr](https://openreview.net/forum?id=crRhj1Y2wv&noteId=O5eCvRo9Jw)).

Following suggestions of the reviewers, we have updated the manuscript with additional experimental and theoretical results, as well as additional clarifications and discussions to make the content more accessible. The new changes are made both to the main text and to the supplementary materials, and new content is highlighted in blue. We will work on adjusting the manuscript back to appropriate number of pages post-rebuttal and when preparing camera-ready.

Summary of changes:

* **General**
   * As suggested by Reviewer [adBr](https://openreview.net/forum?id=crRhj1Y2wv&noteId=O5eCvRo9Jw), we clarified the focus of the paper to variable importance estimation. We have updated the language throughout the manuscript to reflect this.

* **Theoretical Clarifications**
   * To address Reviewer [adBr](https://openreview.net/forum?id=crRhj1Y2wv&noteId=O5eCvRo9Jw)'s comment about assumptions on feature map, in Appendix B.1., we listed the full set of assumptions on the model and feature map.
   * To address Reviewers [AEuT](https://openreview.net/forum?id=crRhj1Y2wv&noteId=S91lBUuQRU7) and [adBr](https://openreview.net/forum?id=crRhj1Y2wv&noteId=O5eCvRo9Jw)'s comments about fixed dimension, we added additional discussion around Theorem 1 and a technical remark (Remark 4) in Appendix C to highlight that our posterior concentration result allows for data dimension to grow with sample size in a rate of $o(n)$.

* **New theoretical results on finite-sample error rate**:
   * As suggested by Reviewer [smu5](https://openreview.net/forum?id=crRhj1Y2wv&noteId=nzEdsXrM2YW), we derived a finite-sample result for variable importance measure in Appendix C.1., and added relevant comment in Section 3.2. (lines 259-262).

* **Detailed comparison with related work**:
    * As suggested by Reviewers [2frY](https://openreview.net/forum?id=crRhj1Y2wv&noteId=A2-PtGC3F1l) and [adBr](https://openreview.net/forum?id=crRhj1Y2wv&noteId=O5eCvRo9Jw), in introduction and then Appendix J, we added detailed comparison to related work [He et al. (2021)](http://www3.stat.sinica.edu.tw/sstest/j31n4/J31N421/J31N421.html) and [Liu (2021)](http://proceedings.mlr.press/v130/liu21g/liu21g.pdf).

* **Full set of neural network experiments**
    * As requested by Reviewers [AEuT](https://openreview.net/forum?id=crRhj1Y2wv&noteId=S91lBUuQRU7) and [adBr](https://openreview.net/forum?id=crRhj1Y2wv&noteId=O5eCvRo9Jw), in Section 4 and Appendix H, we added a full set of neural network results to all the 240 experiment settings. The new result is consistent with our existing conclusions (O1-O3), and provides additional empirical support for the role of Lipschitz condition in variable importance estimation.

* **Added algorithm summaries**:
  * To further address concern by [adBr](https://openreview.net/forum?id=crRhj1Y2wv&noteId=MO7INL2H2YK), in  Appendix A.4, we supply two concrete meta-algorithms to illustrate how joint learning of kernel hyperparameters and GP posterior can be done for classic GP kernels, and for advanced kernels like neural networks and decision trees.

In addition, we respond to all clarifications requested by the reviewers individually.

---

### Meta-Review · Area_Chair_77Hi · 2022-08-22

**Recommendation:** Accept
**Confidence:** Certain

**Metareview:**

The authors propose a new method for quantifying variable importance (and thus, for variable selection) in a totally nonparametric setting. They use a Bayesian setting (GP) and quantify the variable importance based on the L2-norm of the corresponding partial derivative. The method comes with strong theoretical guarantees (rate of contraction, BvM theorem) and numerical experiments.

The reviewers initially agreed that the method proposed was nice and that the theoretical results were interesting enough to justify publication. However, some of them also pointed out various problems: weaknesses in the experiments [Reviewers AEuT, 2frY], problems in some figures [AEuT], missing references that would require further discussion [adBr], absence of a non-asymptotic (ie, d not fixed) analysis [smu5, adBr], assumptions to be clarified [adBr]. The authors addressed all these points, fixing the minor mistakes, adding new experiments and non-asymptotic results to the paper (on a personal note, I will add that I appreciate the summary of the changes provided by the authors, that make it easier to track the modifications in the paper). The reviewers agreed the new version of the paper is much better, and ready for publication in its current state.


**Award:**

No

---

### Decision · Program_Chairs · 2022-09-14

Accept